NORDITA 2022-084
UUITP-55/22

# $p$-Forms on the Celestial Sphere

Laura Donnay[a,b1], Erfan Esmaeili[c2], Carlo Heissenberg[d,e3]

[a] *SISSA, Via Bonomea 265, 34136 Trieste, Italy*
[b] *INFN, Sezione di Trieste, Via Valerio 2, 34127 Trieste, Italy*
[c] *School of Physics, Institute for Research in Fundamental Sciences (IPM),*
*P.O.Box 19395-5531, Tehran, Iran*
[d] *Department of Physics and Astronomy, Uppsala University,*
*Box 516, 75120 Uppsala, Sweden*
[e] *Nordita, Stockholm University and KTH Royal Institute of Technology,*
*Hannes Alfvéns väg 12, 106 91 Stockholm, Sweden*

## Abstract

We construct a basis of conformal primary wavefunctions (CPWs) for $p$-form fields in any dimension, calculating their scalar products and exhibiting the change of basis between conventional plane wave and CPW mode expansions. We also perform the analysis of the associated shadow transforms. For each family of $p$-form CPWs, we observe the existence of pure gauge wavefunctions of conformal dimension $\Delta = p$, while shadow $p$-forms of this weight are only pure gauge in the critical spacetime dimension value $D = 2p + 2$. We then provide a systematic technique to obtain the large-$r$ asymptotic limit near $\mathscr{I}$ based on the method of regions, which naturally takes into account the presence of both ordinary and contact terms on the celestial sphere. In $D = 4$, this allows us to reformulate in a conformal primary language the links between scalars and dual two-forms.

[1] ldonnay@sissa.it
[2] erfanili@ipm.ir
[3] carlo.heissenberg@physics.uu.se

# 1 Introduction

Conformal primary wavefunctions (CPWs) [1,2] with conformal dimension $\Delta$ are solutions of the free-field equations of motion $T^{\mu_1\cdots\mu_p}_{\Delta,a_1\cdots a_k}(x,w)$ that transform covariantly under Lorentz transformations $\Lambda^\mu{}_\nu$, according to

$$T^{\mu_1\cdots\mu_p}_{\Delta,a_1\cdots a_k}(\Lambda x, w') = \alpha_\Lambda^{\Delta-p}(w)\frac{\partial w^{b_1}}{\partial w'^{a_1}}\cdots\frac{\partial w^{b_k}}{\partial w'^{a_k}}\Lambda^{\mu_1}{}_{\nu_1}\cdots\Lambda^{\mu_p}{}_{\nu_p}T^{\nu_1\cdots\nu_p}_{\Delta,b_1\cdots b_k}(x,w)\,, \tag{1.1}$$

where $q^\mu(w)$ with $w = (w^1,\ldots,w^{D-2})$ is a fixed section of the light-cone $q^\mu(w)q_\mu(w) = 0$ and

$$\Lambda^\mu{}_\nu q^\nu(w) = \alpha_\Lambda(w)\,q^\mu(w')\,. \tag{1.2}$$

The interest in this type of field solutions is mainly motivated by the celestial holography program, where one aims at encoding the $S$-matrix of quantum gravity in four dimensions in terms of a two-dimensional celestial conformal field theory; see [1–13] for early works and [14] for more references. In this framework, one trades 4D scattering amplitudes for 2D correlators, and symmetries of the "bulk" $S$-matrix naturally translate into Ward identities for the "boundary" description. With the aim of studying the properties of such correlators in a conformal basis, it is of course important to construct CPWs not only for electromagnetic potentials $A^\mu_{a,\Delta}$ and linearized metric fluctuations $h^{\mu\nu}_{a,\Delta}$, but also for more exotic types of fields that can be relevant in the ultimate celestial formulation of quantum gravity [15–20]. For instance, two-form fields naturally emerge in double-copy constructions, whereby one decomposes the "square" of a one-form field into irreducible components, including a graviton, a scalar and a two-form. Thus two-form primaries are expected to play a role in future investigations of celestial double copies [19,21–24].

In this work, we explicitly construct CPWs for two-forms in $D = 4$ and more generally for $p$-forms in generic spacetime dimensions $D \geq 4$. For each family of CPWs, we calculate the invariant scalar products and we identify conformal dimensions corresponding to pure gauge configurations. For $p$-forms, these pure gauge CPWs turn out to occur for $\Delta = p$ irrespectively of the spacetime dimension $D$. We also carry out explicitly the construction of the corresponding shadow transforms, obtaining for each CPW with dimension $\Delta$ a shadow partner with conformal weight $\Delta' = D - 2 - \Delta$. Moreover, in the critical dimension $D = 2 + 2p$ for each form degree $p$, shadows with $\Delta' = p$ are proportional to the original non-shadow CPW and thus are also pure gauge. Our strategy is based on two building blocks: the scalar CPWs,

$$\Phi_\Delta(x,w) = \frac{(\mp i)^\Delta \Gamma(\Delta)}{(-x\cdot q(w)\mp i0)^\Delta}\,, \tag{1.3}$$

which can be seen as Mellin transforms of conventional plane wave states [1–3,6], and the "Mellin-space polarization vectors" [19]

$$\epsilon^\mu_a(x,w) = \frac{\partial}{\partial w^a}\left(\frac{q^\mu(w)}{-x\cdot q(w)\mp i0}\right)\,, \tag{1.4}$$

which obey simple transformation rules under Lorentz transformations,

$$\epsilon_a^\mu(\Lambda x, w') = \frac{\partial w^b}{\partial w'^a} \, \Lambda^\mu{}_\nu \epsilon_a^\nu(x, w) \,. \tag{1.5}$$

These two ingredients neatly combine to produce all higher-form CPWs. Along the way, we also comment on the relation between conformal primary transformation rules and Wigner rotations and translations arising in the standard little-group construction [25, 26].

A further motivation for studying $p$-form conformal primaries is provided by dualities, which for instance relate forms of different degrees to one another depending on the spacetime dimension. When $D = 4$, in particular, the two-form field is naturally dual to a scalar degree of freedom. This fact was at the basis of the proposal to identify scalar soft theorems and their associated charges [27, 28] as manifestations of asymptotic symmetries involving their dual two-forms [29, 30]. Here we provide an explicit realization of this duality. This turns out to map scalar CPWs to two-form CPWs with the same conformal weight, with singularities associated to the constant $\Delta = 0$ on the scalar side and to the pure gauge mode $\Delta = 2$ on the two-form side. With an appropriate normalization, the canonical pairing between this $\Delta = 2$ pure gauge configuration and the $\Delta = 0$ CPW leads to an explicit finite and nonzero expression for the associated charge.

In order to investigate the behavior of the CPWs obtained in this way near null infinity, one is also led to analyze in detail the limits of scalar CPWs, and more generally of solutions of the wave equation, as the radius $r$ is sent to infinity for fixed retarded time [9, 20, 31, 32]. Due to singularities appearing in null directions and in particular on the celestial sphere, these limits need to be taken in a distributional sense. We formulate here a strategy based on the method of regions [33, 34] to handle them in a systematic fashion, treating carefully contact-term contributions. In the limit, one finds that two regions of integration are important, and the near-collinear one dominates the asymptotic expansion for $D > 4$. In general, both regions enter the leading-order expansion in $D = 4$, leading to the appearance of logarithmic terms $\log r$. However, the combination appropriate to the calculation of wave-forms is log-free, consistently with previous analyses.

Finally, we come back to the issue of two-forms and their dual scalar charges in $D = 4$, discussing their explicit expressions both in the conventional plane wave basis and in the new celestial basis. The two analyses agree and confirm that the $\Delta = 1$ mode can be held responsible for the leading soft theorem [19, 27, 35].

The paper is organized as follows. After collecting some preliminary material about polarization vectors and little-group transformations in section 2, we discuss the construction of CPWs for the various form degrees in section 3 and their shadow transforms in section 4. We discuss our method for calculating asymptotic limits in section 5 and conclude with a discussion of $D = 4$ two-forms and dual scalar fields in section 6.

**Notation.** Greek letters $\mu, \nu, \alpha, \beta, \dots$ denote $D$-dimensional spacetime indices, while Latin letters $a, b, \dots$ denote $d \equiv (D-2)$-dimensional transverse-space indices. The square brackets on $p$ indices denote the alternating sum over permutations of such indices without additional factors, for instance $A_{[\mu} B_{\nu]} = A_\mu B_\nu - A_\nu B_\mu$. We adopt the mostly-plus signature $\eta_{\mu\nu} = \text{diag}(-+\cdots+)$. In the discussion of shadow transforms, $\widetilde{\Phi}_\Delta$ denotes the shadow transform of the conformal primary with weight $\Delta$, itself with weight $d - \Delta$.

# 2 Plane waves and polarizations

In this section, we collect useful preliminary material that is later used to discuss conformal primaries. We begin by recalling some facts about the geometry of the light-cone, which we then employ in order to construct ordinary polarization vectors, making contact explicitly with the transformation rules under little-group transformations and Wigner rotations. We then employ such polarization vectors to construct two-form and $p$-form polarization tensors.

## 2.1 Projective light-cone

When dealing with massless fields, a distinguished role is played by the light-cone, defined by

$$p^2 = 0 \,. \tag{2.1}$$

To parametrize it, it can be useful to fix a reference chart of the form

$$p^\mu = \lambda \, q^\mu(w), \tag{2.2}$$

where $\lambda$ and $w^a$, with $a = 1, 2, \dots, D-2$, are real coordinates. In the next subsection we will specialize to a standard choice of $q^\mu(w)$ (2.23) that simplifies several formulas. Once a choice of $q^\mu(w)$ has been made, it induces tangent vectors

$$e_a^\mu(q) = \partial_a q^\mu \,, \tag{2.3}$$

and a metric

$$h_{ab}(q) = e_a^\mu(q) \, \eta_{\mu\nu} \, e_a^\nu(q) \tag{2.4}$$

on the $\lambda = 1$ cross section it describes.

The projective light-cone can be defined by identifying vectors that differ only by an overall rescaling:

$$p^2 = 0 \,, \qquad p^\mu \sim \lambda \, p^\mu \,. \tag{2.5}$$

This suggests considering an expression that is automatically invariant under rescalings:

$$q^\mu \to \frac{q^\mu}{u \cdot q}\,, \tag{2.6}$$

with a suitable reference vector $u^\mu$, and to consider as tangent vectors

$$\epsilon_a^\mu(q) = \partial_a \left( \frac{q^\mu}{u \cdot q} \right) = \frac{1}{u \cdot q} \left( \partial_a q^\mu - \frac{u \cdot \partial_a q}{u \cdot q} \, q^\mu \right)\,, \tag{2.7}$$

so that, by construction,

$$\epsilon_a^\mu(\lambda\, q) = \epsilon_a^\mu(q)\,. \tag{2.8}$$

The metric associated to $\epsilon_a^\mu(q)$ is given by

$$\gamma_{ab}(q) = \epsilon_a^\mu(q)\, \eta_{\mu\nu}\, \epsilon_b^\nu(q) = \frac{1}{(u \cdot q)^2}\, e_a^\mu(q)\, \eta_{\mu\nu}\, e_b^\nu(q) = \frac{h_{ab}(q)}{(u \cdot q)^2}\,, \tag{2.9}$$

since the additional pieces in (2.7) drop out by orthogonality. Again, (2.9) is manifestly invariant under local rescalings.

We shall adopt (2.7) as our preferred tangent vectors. To denote points on the projective light-cone, we shall also use interchangeably $q^\mu$ or $w$, writing for instance

$$\epsilon_a^\mu(w) = \epsilon_a^\mu(q(w)) \tag{2.10}$$

and

$$ds^2 = \gamma_{ab}(w)\, dw^a dw^b\,, \qquad \gamma_{ab}(w) = \gamma_{ab}(q(w))\,. \tag{2.11}$$

Under a generic coordinate change $w = w(w')$, one then has

$$\epsilon_a'^\mu(w') = \frac{\partial w^b}{\partial w'^a}\, \epsilon_b^\mu(w)\,, \qquad \gamma_{ab}'(w') = \frac{\partial w^c}{\partial w'^a}\, \gamma_{cd}(w)\, \frac{\partial w^d}{\partial w'^b}\,. \tag{2.12}$$

Let us now consider a Lorentz transformation $\Lambda^\mu{}_\nu \in SO(1, D-1)$, which acts via

$$q^\mu \mapsto \Lambda^\mu{}_\nu\, q^\nu\,. \tag{2.13}$$

Since $\Lambda^\mu{}_\nu q^\nu(w)$ is null, the effect of the Lorentz transformation $\Lambda$ can be also characterized by a mapping $w \mapsto w'(w)$ such that

$$\Lambda^\mu{}_\nu\, q^\nu(w) = \alpha_\Lambda(w) q^\mu(w')\,, \tag{2.14}$$

for a suitable factor $\alpha_\Lambda$. Taking a derivative of this relation with respect to $w'^a$ leads to

$$\Lambda^\mu{}_\nu \frac{\partial w^c}{\partial w'^a} e_c^\nu(w) = \alpha_\Lambda(w) e_a^\mu(w') + \frac{\partial \alpha_\Lambda}{\partial w'^a}\, q^\mu(w')\,. \tag{2.15}$$

Taking the "square" of this identity, and using $q^2 = 0 = e_a^\mu q_\mu$, we see that $\alpha_\Lambda$ acts as a conformal factor for $h_{ab}$,

$$\frac{\partial w^c}{\partial w'^a} h_{cd}(w) \frac{\partial w^d}{\partial w'^b} = \alpha_\Lambda^2(w) h_{ab}(w') . \tag{2.16}$$

The determinant of this relation gives

$$\alpha_\Lambda^{D-2}(w) = \sqrt{\frac{h(w)}{h(w')}} \det\left(\frac{\partial w^a}{\partial w'^b}\right), \tag{2.17}$$

where $h(w) = \det(h_{ab}(w))$. Under the Lorentz transformation described by (2.14), the metric also obeys

$$ds^2 = \gamma_{ab}(w) dw^a dw^b = \frac{(u \cdot \Lambda q(w))^2}{(u \cdot q(w))^2} \gamma_{ab}(w') dw'^a dw'^b , \tag{2.18}$$

as can be checked using eq. (2.16) and (2.14) contracted with $u^\mu$, so that

$$\gamma'_{ab}(w') = \frac{\gamma_{ab}(w')}{\Omega^2(w)} , \qquad \Omega(w) = \frac{u \cdot q(w)}{u \cdot \Lambda q(w)} . \tag{2.19}$$

In view of (2.14), $\epsilon_a^\mu$ behaves as follows under Lorentz transformations

$$\epsilon_a^\mu(\Lambda q) \equiv \frac{\partial}{\partial w'^a}\left(\frac{q^\mu(w')}{u \cdot q(w')}\right) = \epsilon_a^\mu(w') . \tag{2.20}$$

Consequently, it obeys the nonlinear transformation law

$$\left(\Lambda_\nu{}^\mu - \frac{q^\mu}{u \cdot q} \Lambda_\nu{}^\rho u_\rho\right) \epsilon_a^\nu(\Lambda q) = \frac{u \cdot q}{u \cdot \Lambda q} \frac{\partial w^b}{\partial w'^a} \epsilon_b^\mu(q) . \tag{2.21}$$

It is useful to make the dependence on the reference vector $u^\mu$ explicit, writing for instance $\epsilon_a^\mu(u; q)$ or $\epsilon_a^\mu(u; w)$. Then, one finds the simpler transformation rule

$$\epsilon_a^\mu(\Lambda u; w') = \frac{\partial w^b}{\partial w'^a} \Lambda^\mu{}_\nu \epsilon_b^\nu(u; w) . \tag{2.22}$$

## 2.2 Standard parametrization

A very convenient choice for the section $q^\mu(w)$ is

$$q^\mu(w) = \left(\frac{1 + |w|^2}{2}, w^a, \frac{1 - |w|^2}{2}\right) \tag{2.23}$$

i.e.

$$\lambda = p^0 + p^{D-1} , \qquad w^a = \frac{p^a}{p^0 + p^{D-1}} , \tag{2.24}$$

in (2.2). Notice that this choice differs from the one usually taken in the celestial holography literature (e.g. [2, 7]) by a factor of two. The resulting metric $h_{ab}$ with this choice is flat $h_{ab} =$

$\delta_{ab}$ and the coordinates $w^a$ cover Euclidean space $\mathbb{R}^{D-2}$. Moreover, identifying $u^\mu = -n^\mu = (-1, 0, \ldots, 0, 1)$, one has $u \cdot q(w) = 1$ for any $w^a$, so that $e_a^\mu(q) = \epsilon_a^\mu(q)$.

Under the Lorentz transformation $\Lambda^\mu{}_\nu$, via (2.14), we find

$$- n_\mu \Lambda^\mu{}_\nu\, q^\nu(w) = \alpha_\Lambda(w)\,, \tag{2.25}$$

while (2.16) and (2.17) reduce to

$$\frac{\partial w^c}{\partial w'^a} \frac{\partial w^c}{\partial w'^b} = \alpha_\Lambda^2(w)\delta_{ab} \tag{2.26}$$

and

$$\alpha_\Lambda^{D-2}(w) = \det\left(\frac{\partial w^a}{\partial w'^b}\right)\,. \tag{2.27}$$

In particular, the square root on the right-hand side of (2.17) yields 1 for this parametrization. Finally, the transformation law (2.21) simplifies to

$$\left(\Lambda_\nu{}^\mu - \frac{q^\mu}{n \cdot q} \Lambda_\nu{}^\rho\, n_\rho\right) e_a^\nu(\Lambda q) = \frac{1}{\alpha_\Lambda(w)} \frac{\partial w^b}{\partial w'^a} e_b^\mu(q)\,. \tag{2.28}$$

## 2.3 Little group

In this section we compare the polarization vectors constructed above with the ones that one usually builds in terms of little-group elements (see e.g. [36, 37]). In particular, this provides an explicit link between their transformation laws on the celestial sphere and standard Wigner rotations; see also [38, 39].

Given the reference vector

$$k^\mu = (\kappa, 0, \ldots, 0, \kappa)\,, \tag{2.29}$$

the most general Lorentz transformation which leaves $k^\mu$ invariant takes the form

$$\mathcal{R} = \begin{pmatrix} 1 + \frac{|x|^2}{2} & x^b & -\frac{|x|^2}{2} \\ (\mathcal{O}x)^a & \mathcal{O}^{ab} & -(\mathcal{O}x)^a \\ \frac{|x|^2}{2} & x^b & 1 - \frac{|x|^2}{2} \end{pmatrix}\,, \tag{2.30}$$

where $x^a$, with $a = 1, \ldots, D-2$, are real parameters and $\mathcal{O}$ is a $(D-2)$-dimensional rotation matrix. For any null vector $p^\mu$ of the form

$$p^\mu = \omega\, q^\mu(z)\,, \qquad q^\mu(z) = \frac{1}{2}\left(1 + |z|^2, 2z^a, 1 - |z|^2\right)\,, \tag{2.31}$$

one can always construct a Lorentz transformation $\mathcal{L}(p)$ such that

$$\mathcal{L}(p)^\mu{}_\nu\, k^\nu = p^\mu\,. \tag{2.32}$$

For definiteness, let us take $\mathcal{L}(p)$ as follows

$$\mathcal{L}(p) = R(z)\, B(\omega) \tag{2.33}$$

with $B(\omega)$ a boost in the direction $D-1$, which only affects the overall scale of $k^\mu$, and $R(z)$ a Lorentz transformation that aligns it in the direction specified by $z^a$,

$$B(\omega) = \begin{pmatrix} \frac{1}{2}\left(\frac{\omega}{2\kappa} + \frac{2\kappa}{\omega}\right) & 0 & \frac{1}{2}\left(\frac{\omega}{2\kappa} - \frac{2\kappa}{\omega}\right) \\ 0 & \delta^{ab} & 0 \\ \frac{1}{2}\left(\frac{\omega}{2\kappa} - \frac{2\kappa}{\omega}\right) & 0 & \frac{1}{2}\left(\frac{\omega}{2\kappa} + \frac{2\kappa}{\omega}\right) \end{pmatrix}, \qquad R(z) = \begin{pmatrix} 1 + \frac{|z|^2}{2} & z^b & \frac{|z|^2}{2} \\ z^a & \delta^{ab} & z^a \\ -\frac{|z|^2}{2} & -z^b & 1 - \frac{|z|^2}{2} \end{pmatrix}. \tag{2.34}$$

Let us now define

$$\mathcal{R}(\Lambda, p) = \mathcal{L}(p)^{-1} \Lambda^{-1} \mathcal{L}(\Lambda p). \tag{2.35}$$

This transformation belongs to the little group of $k^\mu$, since it obeys

$$\mathcal{R}(\Lambda, p)^\mu{}_\nu \, k^\nu = k^\mu \tag{2.36}$$

by construction, and therefore it must take the form (2.30) with suitable $\mathcal{O}^{ab}(\Lambda, p)$ and $x_a(\Lambda, p)$. We define the physical polarizations according to

$$\hat{e}_a = \left(0, \delta_a^b, 0\right), \qquad e_a^\mu(z) = \mathcal{L}(z)^\mu{}_\nu \hat{e}_a^\nu = \left(z^a, \delta^{ab}, -z^a\right), \tag{2.37}$$

where we can identify $e_a^\mu(z) = e_a^\mu(q) = e_a^\mu(p)$. Note that, as a result, $e_a^\mu(z) = \partial q^\mu(z)/\partial z^a$. These obey

$$\mathcal{R}(\Lambda, p)^\mu{}_\nu \hat{e}_a^\nu = \hat{e}_b^\mu \, \mathcal{O}^{ba}(\Lambda, p) + \frac{k^\mu}{\kappa} \, x_a(\Lambda, p) \tag{2.38}$$

and therefore, recalling the decomposition (2.36) and the defining property (2.32),

$$\Lambda_\nu{}^\mu e_a^\nu(\Lambda p) = e_b^\mu(p)\, \mathcal{O}^{ba} + \frac{p^\mu}{\kappa}\, x_a. \tag{2.39}$$

Since $e_a^\mu(p)$ obeys

$$n_\mu e_a^\mu(p) = 0, \qquad n^\mu = (1, 0, \ldots, 0, -1), \tag{2.40}$$

for any $p$, contracting (2.39) with $n_\mu$ we obtain

$$\frac{x_a}{\kappa}\,(n \cdot p) = n_\mu \Lambda_\nu{}^\mu e_a^\nu(\Lambda q) \tag{2.41}$$

and we can recast (2.39) in the form

$$\left(\Lambda_\nu{}^\mu - \frac{p^\mu}{n \cdot p}\, n_\rho \Lambda_\nu{}^\rho\right) e^\nu(\Lambda p) = e_b^\mu(p)\, \mathcal{O}^{ba}(\Lambda, p). \tag{2.42}$$

Comparing with (2.28), this transformation rule is of the same type as (2.42), with the Wigner rotation explicitly given by

$$\mathcal{O}^{ba} = \frac{1}{\alpha_\Lambda(w)} \frac{\partial w^b}{\partial w'^a}. \tag{2.43}$$

Note that, indeed, (2.43) identifies an orthogonal matrix thanks to (2.26).

## 2.4 Standard 4D expressions

When $D = 4$, switching to complexified coordinates and defining

$$z = z^1 + iz^2\,, \qquad \bar{z} = z^1 - iz^2\,, \tag{2.44}$$

the convenient parametrization (2.23) can be cast as follows

$$q^\mu(z, \bar{z}) = \frac{1}{2}\left(1 + z\bar{z}, z + \bar{z}, -i(z - \bar{z}), 1 - z\bar{z}\right)\,, \tag{2.45}$$

so that

$$e_z^\mu = \frac{1}{2}\left(\bar{z}, 1, -i, -\bar{z}\right)\,, \qquad e_{\bar{z}}^\mu = \frac{1}{2}\left(z, 1, i, -z\right)\,, \qquad h_{z\bar{z}} = \frac{1}{2}\,, \qquad h_{zz} = 0 = h_{\bar{z}\bar{z}}\,. \tag{2.46}$$

Moreover, since finite Lorentz transformations are parametrized by

$$z \mapsto z' = \frac{az + b}{cz + d}\,, \qquad ad - bc = 1\,, \tag{2.47}$$

the Jacobian in (2.17) takes the simple form

$$\frac{\partial z'}{\partial z} = \frac{1}{(cz + d)^2}\,, \qquad \alpha_\Lambda(z, \bar{z}) = (cz + d)(c^*\bar{z} + d^*)\,. \tag{2.48}$$

The basic transformation rule (2.22) reads

$$\epsilon_z^\mu(\Lambda u; z', \bar{z}') = (cz + d)\,\Lambda^\mu{}_\nu \epsilon_z^\nu(u; z, \bar{z})\,, \qquad \epsilon_{\bar{z}}^\mu(\Lambda u; z', \bar{z}') = (c^*\bar{z} + d^*)\,\Lambda^\mu{}_\nu \epsilon_{\bar{z}}^\nu(u; z, \bar{z})\,. \tag{2.49}$$

Other useful transformation rules are

$$\nabla^2 = (\partial_1)^2 + (\partial_2)^2 = 4\partial_z\partial_{\bar{z}}\,, \quad 2dz^1 dz^2 = dz\,d\bar{z}\,, \quad \delta^{(2)}(z^1 - z'^1, z^2 - z'^2) = 2\delta^{(2)}(z - z')\,. \tag{2.50}$$

together with

$$2dz^1 \wedge dz^2 = i\,dz \wedge d\bar{z}\,, \qquad \epsilon_{12} = -\epsilon_{21} = +1\,, \qquad \epsilon_{z\bar{z}} = -\epsilon_{\bar{z}z} = \frac{i}{2}\,. \tag{2.51}$$

## 2.5 Polarizations in momentum space

The scalar wave equation

$$\Box \Phi(x) = 0 \tag{2.52}$$

reads, in Fourier space,

$$p^2\varphi(p) = 0\,, \tag{2.53}$$

which restricts the support of $\varphi(p)$ to the light-cone $p^2 = 0$. Positive- and negative-frequency plane wave states can be thus taken as

$$\Phi(x; p) = e^{\pm ipx}, \qquad p^2 = 0, \qquad p^0 > 0. \tag{2.54}$$

The free Maxwell equations for the vector potential $A_\mu(x)$, identified up to

$$A_\mu(x) \sim A_\mu(x) + \partial_\mu \Lambda(x), \tag{2.55}$$

are given by

$$\Box A_\mu(x) - \partial_\mu \partial^\nu A_\nu(x) = 0. \tag{2.56}$$

Adopting Lorenz gauge, one can reduce the discussion to the Fierz system

$$\Box A_\mu(x) = 0, \qquad \partial^\mu A_\mu(x) = 0, \qquad \Box \Lambda(x) = 0. \tag{2.57}$$

In Fourier space, this translates into the conditions

$$p^2 a_\mu(p) = 0, \qquad p^\mu a_\mu(p) = 0, \qquad p^2 \lambda(p) = 0, \tag{2.58}$$

which restrict the supports of $a_\mu(p)$ and $\lambda(p)$ to the light-cone $p^2 = 0$ and enforce the transversality of $a_\mu(p)$. The residual gauge freedom in (2.58) is given by

$$a_\mu(p) \sim a_\mu(p) + \lambda p_\mu. \tag{2.59}$$

Thus, $D - 2$ independent polarizations for $a_\mu$ are naturally given by the tangent vectors $\epsilon_a^\mu(p)$ on the projective light-cone with $a = 1, 2, \ldots, D - 2$. We may therefore consider general plane wave states of the form

$$A_a^\mu(x; p) = \epsilon_a^\mu(p) \Phi(x; p). \tag{2.60}$$

The free two-form $B_{\mu\nu}(x)$ is subject to the gauge equivalence

$$B_{\mu\nu}(x) \sim B_{\mu\nu}(x) + \partial_\mu A_\nu(x) - \partial_\nu A_\mu(x) \tag{2.61}$$

with parameters $A_\mu(x)$ to be identified up to the gauge-for-gauge transformations

$$A_\mu(x) \sim A_\mu(x) + \partial_\mu \Lambda(x). \tag{2.62}$$

In this case the equations of motion are given by

$$\Box B_{\mu\nu}(x) + \partial_\mu \partial^\alpha B_{\nu\alpha}(x) + \partial_\nu \partial^\alpha B_{\alpha\mu}(x) = 0, \tag{2.63}$$

and can be reduced to the Fierz-like system

$$\Box B_{\mu\nu}(x) = 0, \qquad \partial^\nu B_{\mu\nu}(x) = 0, \qquad \Box A_\mu(x) = 0, \qquad \partial^\mu A_\mu(x) = 0, \qquad \Box \Lambda(x) = 0. \tag{2.64}$$

In Fourier space, all the fields have support on the light-cone and they obey

$$p^\mu b_{\mu\nu}(p) = 0 \,, \qquad p^\mu a_\mu(p) = 0 \,, \tag{2.65}$$

while gauge and gauge-for-gauge transformations translate into

$$b_{\mu\nu}(p) \sim b_{\mu\nu}(p) + p_\mu a_\nu(p) - p_\nu a_\mu(p) \,, \qquad a_\mu(p) \sim a_\mu(p) + \lambda p_\mu \,. \tag{2.66}$$

The second relation in (2.65) and the gauge-for-gauge residual freedom (2.66) can be used to identify $D-2$ independent polarizations $\epsilon_a^\mu$ for $a_\mu$ as discussed for the spin-one case. The first relation in (2.65) enforces $D-1$ constraints on $b_{\mu\nu}$ (since $p^\mu p^\nu b_{\mu\nu} = 0$ is identically true). On the other hand, $b_{\mu\nu}$ starts out with $D(D-1)/2$ independent components, in view of its antisymmetry, so that, after imposing transversality, one is left with $(D-1)(D-2)/2$ independent components. Therefore, we can parametrize $b^{\mu\nu}$ in the following way

$$b^{\mu\nu} = \sum_{a<b} \varphi_{ab} \left( \epsilon_a^\mu \epsilon_b^\nu - \epsilon_a^\nu \epsilon_b^\mu \right) + \sum_a \lambda_a (p^\mu \epsilon_a^\nu - p^\nu \epsilon_a^\mu) \,, \tag{2.67}$$

with suitable coefficients $\varphi_{ab}$ and $\lambda_a$. The second type of terms can be eliminated using the gauge freedom in eq. (2.66), arriving at

$$b^{\mu\nu} = \sum_{a<b} \varphi_{ab} \, \epsilon_{ab}^{[\mu\nu]} \,, \qquad \epsilon_{ab}^{[\mu\nu]} = \epsilon_a^\mu \epsilon_b^\nu - \epsilon_a^\nu \epsilon_b^\mu \tag{2.68}$$

which shows how $(D-2)(D-3)/2$ independent two-form polarizations can be constructed from the $D-2$ photon polarizations. Consequently, we may write

$$B_{ab}^{\mu\nu}(x;p) = \epsilon_{ab}^{[\mu\nu]}(p)\Phi(x;p) \,. \tag{2.69}$$

In the $D=4$ case, only one independent polarization is available, for instance

$$\epsilon^{[\mu\nu]} = \epsilon_1^\mu \epsilon_2^\nu - \epsilon_1^\nu \epsilon_2^\mu \,. \tag{2.70}$$

For a $p$-form, $\binom{D-2}{p}$ independent polarizations can be constructed in a similar fashion, taking antisymmetrized products of one-form polarizations

$$\epsilon_{a_1 \cdots a_p}^{[\mu_1 \cdots \mu_p]} = \epsilon_{a_1}^{[\mu_1} \cdots \epsilon_{a_p}^{\mu_p]} \,. \tag{2.71}$$

For completeness, let us also note that $D(D-3)/2$ independent transverse, traceless polarizations for the spin-two field can be taken as follows

$$\epsilon_{ab}^{(\mu\nu)} = \epsilon_a^\mu \epsilon_b^\nu + \epsilon_a^\nu \epsilon_b^\mu - \frac{2\gamma_{ab}}{D-2} \, \epsilon_c^\mu \gamma^{cd} \epsilon_d^\nu \,, \tag{2.72}$$

where $\gamma_{ab} = \epsilon_a^\mu \, \eta_{\mu\nu} \, \epsilon_b^\nu$ as in (2.9) and $\gamma^{ab}$ is its inverse. In $D=4$, it is convenient to choose a parametrization where $\gamma_{ab}$ vanishes for the two independent physical polarizations.

# 3  $p$-form conformal primary wavefuntions

In this section we construct the CPWs for $p$-form fields out of the building blocks discussed in the previous section. We then evaluate their scalar products and identify conformal weights corresponding to pure gauge configurations. We also provide explicit maps between plane wave and CPW mode expansions for the quantized field operators. For ease of presentation, we first review as a warm up the scalar (zero-form) and the vector (one-form) originally built in [2], and then move to two-form and $p$-form fields.

## 3.1  Scalar

The Mellin transform of a scalar plane wave state is given by [1–3, 6]

$$\Phi_\Delta^\pm(x; q) = \int_0^\infty \frac{d\omega}{\omega} \, \omega^\Delta \, e^{i(\pm q^\mu x_\mu + i0)\omega} = \frac{\Gamma(\Delta)}{(0 \mp iq \cdot x)^\Delta} = \frac{(\mp i)^\Delta \Gamma(\Delta)}{(\mp i0 - q \cdot x)^\Delta} \,. \tag{3.1}$$

Here, the upper (lower) sign corresponds to positive (negative) frequency plane waves and $+i0$ indicates a small positive imaginary part. To simplify the notation, we will make the $\pm$ labels on the fields such as $\Phi_\Delta^\pm$ explicit only when strictly necessary. Incidentally, let us note that the map $x \to -x$ has the only effect of flipping this sign, interchanging incoming with outgoing plane waves.

The field $\Phi_\Delta(x; q(w)) = \Phi_\Delta(x; w)$ defines a conformal primary wave function of dimension $\Delta$ since indeed, under the mapping (2.14),

$$\Phi_\Delta(\Lambda x; w') = \alpha_\Lambda^\Delta(w) \Phi_\Delta(x; w) \,, \tag{3.2}$$

with $\alpha_\Lambda(w)$ given in (2.17). In the standard $D = 4$ conventions of subsection 2.4, this reads

$$\Phi_\Delta(\Lambda x; z', \bar{z}') = (cz + d)^\Delta (c^* \bar{z} + d^*)^\Delta \Phi_\Delta(x; z, \bar{z}) \,. \tag{3.3}$$

Moreover, $\Phi_\Delta(x; w)$ satisfies the Klein–Gordon equation with respect to $x$,

$$\Box \Phi_\Delta(x; w) = 0 \,. \tag{3.4}$$

To evaluate the standard scalar product

$$(f, f') = -i \int_\Sigma d\Sigma^\mu \left( f \partial_\mu f'^* - f'^* \partial_\mu f \right) \tag{3.5}$$

between two such conformal primaries, it is convenient to use the integral representation in (3.1), which leads to

$$(\Phi_\Delta(x; q), \Phi_{\Delta'}(x, q')) = -\int_0^\infty \frac{d\omega}{\omega} \omega^\Delta \int_0^\infty \frac{d\omega'}{\omega'} \omega'^{\Delta'^*} \left( \omega q_\mu + \omega' q'_\mu \right) \int_\Sigma d\Sigma^\mu \, e^{i(\omega q \cdot x - \omega' q' \cdot x)} \,, \tag{3.6}$$

focusing for definiteness on the product of two positive-frequency states. Note that the scalar product satisfies

$$(f, f')^* = (f', f) , \qquad (f^*, f'^*) = - (f', f) , \tag{3.7}$$

so the product between negative-frequency states can be obtained by complex conjugation. Choosing the $x^0 = 0$ surface (so that $d\Sigma^\mu \equiv d^{D-1}x \, \delta_0^\mu$), eq. (3.6) yields

$$(\Phi_\Delta(x; q), \Phi_{\Delta'}(x, q')) = \int_0^\infty \frac{d\omega}{\omega} \omega^\Delta \int_0^\infty \frac{d\omega'}{\omega'} \omega'^{\Delta'^*} (\omega q^0 + \omega' q'^0) (2\pi)^{D-1} \delta^{(D-1)}(\omega q - \omega' q') , \tag{3.8}$$

where the argument of the delta function is restricted to the $D-1$ spatial components. We can use the identity

$$\int d^{D-1}p \, f(p) T(p) = \int d\Omega_{D-2}(q) \int_0^\infty \omega^{D-2} d\omega \, f(\omega q) T(\omega q) , \tag{3.9}$$

which follows from the change of variables $p^I = \omega q^I(w)$ with $I = 1, \dots, D-1$

$$d^{D-1}p = \det\left(\frac{\partial p^I}{\partial(\omega, w^a)}\right) d\omega \, d^{D-2}w = \omega^{D-2} d\omega \det\begin{pmatrix} q^I \\ \partial_a q^I \end{pmatrix} d^{D-2}w = \omega^{D-2} d\omega \, d\Omega_{D-2}(q) , \tag{3.10}$$

to conclude that the delta function can be factorized as follows

$$\delta^{(D-1)}(\omega q - \omega' q') = \frac{1}{\omega^{D-2}} \delta(\omega - \omega') \, \delta(q, q') . \tag{3.11}$$

Here $\delta(q, q')$ is the invariant delta function on the $(D-2)$-surface parametrized by the spatial components of $q(w)$, which obeys

$$\int d\Omega_{D-2}(q') \, \varphi(q') \, \delta(q, q') = \varphi(q) . \tag{3.12}$$

In terms of the standard parametrization (2.23),

$$d\Omega_{D-2}(q) = \det\begin{pmatrix} q^I \\ \partial_a q^I \end{pmatrix} d^d\vec{w} = q^0(\vec{w}) d^d\vec{w} , \qquad \delta(q, q') = \frac{1}{q^0(\vec{w})} \delta^{(d)}(\vec{w} - \vec{w}') . \tag{3.13}$$

We then have

$$(\Phi_\Delta(x; q), \Phi_{\Delta'}(x, q')) = (2\pi)^{D-1} 2q^0 \, \delta(q, q') \int_0^\infty \frac{d\omega}{\omega} \omega^{\Delta+\Delta'^* - D+2} . \tag{3.14}$$

The last integral is well-defined provided that the conformal weights take the form

$$\Delta = \frac{D-2}{2} + i\lambda , \qquad \lambda \in \mathbb{R} , \tag{3.15}$$

so that we can use

$$\int_0^\infty \frac{d\omega}{\omega} \omega^{i(\lambda-\lambda')} = \int_{-\infty}^{+\infty} dt \, e^{it(\lambda-\lambda')} = 2\pi \, \delta(\lambda - \lambda') \tag{3.16}$$

to obtain

$$(\Phi_\Delta(x;q), \Phi_{\Delta'}(x,q')) = (2\pi)^D 2q^0 \, \delta(q,q')\delta(\lambda - \lambda') = (2\pi)^D 2 \, \delta^{(d)}(\vec{w} - \vec{w}')\delta(\lambda - \lambda'). \tag{3.17}$$

This is by now a standard derivation showing that CPWs with dimensions lying on the principal series (3.15) form a basis for normalizable radiative wave packets [2]. In the Mellin transform (3.1), the conformal dimension is however in principle an arbitrary complex number $\Delta \in \mathbb{C}$. In [31], it was showed that conformal primaries with analytically continued conformal dimension can be understood as certain contour integrals on the principal series. Using the generalization of the Dirac delta function to the complex plane considered in [31], we may write

$$(\Phi_\Delta(x;q), \Phi_{\Delta'}(x;q')) = I(\Delta, q; \Delta', q'), \tag{3.18}$$

where we introduced the shorthand notation

$$I(\Delta, q; \Delta', q') \equiv (2\pi)^D 2 \, \delta^{(d)}(\vec{w} - \vec{w}')\delta(i(\Delta + \Delta'^* - D + 2)) \tag{3.19}$$

for later convenience. The precise meaning of the formal distribution $\delta(iz)$ has been discussed in [31]. Note that, while in the present case the inner product between positive- and negative-energy states is zero, in the massive case this would no longer be the case [2].

Let us also mention that the following more general family of functions, called generalized conformal primaries [19],

$$\phi_\Delta(x;q) = \frac{f(x^2)}{(x \cdot q)^\Delta}, \tag{3.20}$$

with $f$ a real function, also obey the transformation rule (3.2). In this family, the Klein–Gordon equation selects only two independent solutions (up to normalization):

$$\phi_{1,\Delta} = \frac{1}{(-q \cdot x)^\Delta}, \qquad \phi_{2,d-\Delta} = \frac{(x^2)^{\frac{d}{2}-\Delta}}{(-q \cdot x)^{d-\Delta}}. \tag{3.21}$$

Indeed, $\phi_{1,\Delta}$ is proportional to the scalar CPW (3.1), while as we shall see below $\phi_{2,\Delta}$ is proportional to its shadow transform [2].

Let us now compare the the standard plane wave mode expansion of a scalar field operator $\Phi(x)$ with its conformal basis counterpart. The former reads

$$\Phi(x) = \int \left[e^{ip \cdot x} a(p) + e^{-ip \cdot x} a^\dagger(p)\right] 2\pi\delta(p^2)\theta(p^0) \frac{d^D p}{(2\pi)^D}, \tag{3.22}$$

with the usual commutation relations for creation and annihilation operators

$$[a(p), a^\dagger(p')] = 2|\mathbf{p}|(2\pi)^{D-1}\delta^{(D-1)}(\mathbf{p} - \mathbf{p}')\,, \tag{3.23}$$

or equivalently (with the parametrization (2.31); more details on this step are given in Sect. 5.2)

$$\Phi(x) = \frac{1}{(2\pi)^{d+1}} \int_{\mathbb{R}^d} d^d\vec{w} \int_0^\infty \omega^d \frac{d\omega}{2\omega} \left[ e^{i\omega q(\vec{w})\cdot x} a(\omega\, q(\vec{w})) + e^{-i\omega q(\vec{w})\cdot x} a^\dagger(\omega\, q(\vec{w})) \right]. \tag{3.24}$$

The inverse Mellin transform gives

$$\int_{c-i\infty}^{c+i\infty} \frac{\omega^{-\Delta}\,\Gamma(\Delta)}{(0 \mp iq\cdot x)^\Delta} \frac{d\Delta}{i2\pi} = e^{\pm i\omega q\cdot x}\,, \tag{3.25}$$

as can be seen closing the contour to the left of the line $(c - i\infty, c + i\infty)$, with positive $c$, which runs parallel to the imaginary axis. Using (3.25) in (3.24) then yields (see [9,31] for analogous mode expansions for operators with spin)

$$\Phi(x) = \frac{1}{2(2\pi)^{d+1}} \int_{\mathbb{R}^d} d^d\vec{w} \int_{c-i\infty}^{c+i\infty} \frac{d\Delta}{i2\pi} \left[ \Phi_\Delta^+(x;\vec{w})a_{d-\Delta}(\vec{w}) + \Phi_\Delta^-(x;\vec{w})a_{d-\Delta}^\dagger(\vec{w}) \right], \tag{3.26}$$

where we defined as in [40]

$$a_\Delta(\vec{w}) \equiv \int_0^\infty \omega^{\Delta-1} a(\omega\, q(\vec{w}))\, d\omega\,, \qquad a_\Delta^\dagger(\vec{w}) \equiv \int_0^\infty \omega^{\Delta-1} a^\dagger(\omega\, q(\vec{w}))\, d\omega\,. \tag{3.27}$$

Consistently with (3.18) these obey [31]

$$[a_\Delta(\vec{w}), a_{\Delta'}^\dagger(\vec{w}')] = (2\pi)^D 2q^0\, \delta(q, q')\delta(i(\Delta + \Delta'^* - D + 2)) \tag{3.28}$$

and, using the standard scalar product, one also finds

$$\left(\Phi(x), \Phi_\Delta^+(x;\vec{w})\right) = a_\Delta(\vec{w})\,, \qquad \left(\Phi(x), \Phi_\Delta^-(x;\vec{w})\right) = a_\Delta^\dagger(\vec{w}) \tag{3.29}$$

so that

$$a_\Delta(\vec{w})\Phi(x)|0\rangle = \Phi_\Delta^-(x;\vec{w})|0\rangle\,, \qquad a_\Delta^\dagger(\vec{w})\Phi(x)|0\rangle = \Phi_\Delta^+(x;\vec{w})|0\rangle\,. \tag{3.30}$$

Since the standard Fourier mode decomposition for asymptotic field operators is most commonly employed in scattering amplitude calculations, the map (3.27) between creation/annihilation operators for plane waves and CPWs is particularly useful in comparing energetically soft and conformally soft theorems [9–11, 35, 41, 42].

Let us consider the emission of a massless scalar particle on top of a given scattering event. For this process, if in the energetically soft limit $\omega \to 0$ the soft theorem dictates a behavior like $1/\omega$ for the emission amplitude, then the conformally soft limit is $\Delta \to 1$. A quick way to see

this is the following [43]. Suppose $f(\omega) \sim c_n \, \omega^n$ for some $n$ as $\omega \to 0^+$. Then, introducing an upper cutoff $\Lambda$ in the intermediate steps,

$$f_\Delta = \int_0^\infty \omega^\Delta f(\omega) \frac{d\omega}{\omega} = c_n \int_0^\Lambda \omega^{\Delta+n} \frac{d\omega}{\omega} + \int_\Lambda^\infty \omega^\Delta f(\omega) \frac{d\omega}{\omega} = c_n \frac{\Lambda^{\Delta+n}}{\Delta + n} + \text{regular} \qquad (3.31)$$

and thus

$$\text{Res} f_\Delta \Big|_{\Delta=-n} = c_n \,. \qquad (3.32)$$

More precisely, we can start from the energetically soft theorem written as

$$\langle \text{out}|a(\omega q(\vec{w}))\mathcal{S}|\text{in}\rangle \sim \omega^\alpha \sum_n \frac{g_n}{p_n \cdot q(\vec{w})} \langle \text{out}|\mathcal{S}|\text{in}\rangle \qquad (3.33)$$

as $\omega \to 0^+$. We have kept the exponent $\alpha$ arbitrary in order to encompass different possible soft behaviors. Then

$$\langle \text{out}|a_\Delta(\vec{w})\mathcal{S}|\text{in}\rangle \sim \int_0^\Lambda \omega^{\alpha+\Delta} \frac{d\omega}{\omega} \sum_n \frac{g_n}{p_n \cdot q(\vec{w})} \langle \text{out}|\mathcal{S}|\text{in}\rangle + \cdots \qquad (3.34)$$

so that

$$\langle \text{out}|a_\Delta(\vec{w})\mathcal{S}|\text{in}\rangle \sim \frac{\Lambda^{\alpha+\Delta}}{\alpha + \Delta} \sum_n \frac{g_n}{p_n \cdot q(\vec{w})} \langle \text{out}|\mathcal{S}|\text{in}\rangle + \cdots \qquad (3.35)$$

and

$$\text{Res}\langle \text{out}|a_\Delta(\vec{w})\mathcal{S}|\text{in}\rangle \Big|_{\Delta=-\alpha} \sim \sum_n \frac{g_n}{p_n \cdot q(\vec{w})} \langle \text{out}|\mathcal{S}|\text{in}\rangle \,. \qquad (3.36)$$

For instance, for the leading soft theorem, $\alpha = -1$ and the relevant weight is thus $\Delta = 1$, independently of the spacetime dimensions, while the sub$^n$-leading soft theorems single out the conformal dimension $\Delta = 0, -1, -2, \ldots$ (see e.g. [35, 44, 45]).

## 3.2 One-form

We now consider the Mellin-space counterpart of eq. (2.60), with polarization vectors given by (2.7), identifying the reference vector with the observation point $u = -x$, or more precisely writing

$$\epsilon_a^\mu(x; q(w)) = \frac{\partial}{\partial w^a} \left( \frac{q^\mu(w)}{\mp i0 - x \cdot q(w)} \right) = -\frac{\partial}{\partial w^a} \frac{\partial}{\partial x_\mu} \log(\pm i0 + x \cdot q(w)) \,, \qquad (3.37)$$

which satisfies

$$q_\mu \epsilon_a^\mu(x; q) = 0 \,, \qquad x_\mu \epsilon_a^\mu(x; q) = 0 \,, \qquad \partial_\mu \epsilon_a^\mu(x; q) = 0 \,, \qquad \partial^\mu \epsilon_a^\nu(x; q) - \partial^\nu \epsilon_a^\mu(x; q) = 0 \,. \quad (3.38)$$

One then defines [2]

$$A_{a,\Delta}^\mu(x; q) = \epsilon_a^\mu(x; q) \, (-q \cdot x) \, \Phi_\Delta(x; q) = \epsilon_a^\mu(x; q) \frac{(\mp i)^\Delta \Gamma(\Delta)}{(\mp i0 - q \cdot x)^{\Delta-1}} \,, \qquad (3.39)$$

or more explicitly

$$A^{\mu}_{a,\Delta}(x;q) = (\mp i)^{\Delta}\Gamma(\Delta)\left[\frac{\partial_a q^{\mu}}{(\mp i0 - q\cdot x)^{\Delta}} + \frac{x\cdot\partial_a q}{(\mp i0 - q\cdot x)^{\Delta+1}}\,q^{\mu}\right].\tag{3.40}$$

In view of the basic transformation rules (2.22) and (3.2), the field (3.39) behaves as a spin-one conformal primary under Lorentz transformations:

$$A^{\mu}_{a,\Delta}(\Lambda x;w') = \alpha_{\Lambda}^{\Delta-1}(w)\frac{\partial w^b}{\partial w'^a}\,\Lambda^{\mu}{}_{\nu}A^{\nu}_{b,\Delta}(x;w)\,.\tag{3.41}$$

In the standard $D=4$ conventions of subsection 2.4, the Jacobian is simply (2.48) and $\alpha_{\Lambda}$ is given by (2.48), so that we have from (3.41)

$$A^{\mu}_{z,\Delta}(\Lambda x;z',\bar{z}') = (cz+d)^{\Delta+1}(c^*\bar{z}+d^*)^{\Delta-1}\,\Lambda^{\mu}{}_{\nu}A^{\nu}_{z,\Delta}(x;z,\bar{z})\tag{3.42}$$

and similarly for the $\bar{z}$ component. Moreover, in view of the last equation in (3.38), the field strength

$$F^{\mu\nu}_{a,\Delta} = \partial^{\mu}A^{\nu}_{a,\Delta} - \partial^{\nu}A^{\mu}_{a,\Delta}\tag{3.43}$$

is given by

$$F^{\mu\nu}_{a,\Delta} = (\Delta-1)(\mp i)^{\Delta}\Gamma(\Delta)\frac{q^{\mu}\epsilon^{\nu}_a - q^{\nu}\epsilon^{\mu}_a}{(\mp i0 - q\cdot x)^{\Delta}}\,,\tag{3.44}$$

and the $\Delta=1$ mode

$$A^{\mu}_{a,\Delta=1}(x;q) = \mp i\epsilon^{\mu}_a(x;q)\tag{3.45}$$

gives rise to a pure gauge configuration. This is also clearly displayed by writing (3.40) in the form

$$A^{\mu}_{a,\Delta} = \left(1 - \frac{1}{\Delta}\right)V^{\mu}_{a,\Delta} + \partial^{\mu}\left(\frac{1}{\Delta}\,x_{\nu}V^{\nu}_{a,\Delta}\right),\tag{3.46}$$

where one isolated a "representative" [2]

$$V^{\mu}_{a,\Delta} = (\mp i)^{\Delta}\Gamma(\Delta)\,\frac{\partial_a q^{\mu}}{(\mp i0 - q\cdot x)^{\Delta}}\,.\tag{3.47}$$

Finally, we note that $A^{\mu}_{a,\Delta}(x;q)$ solves the Maxwell equations thanks to the properties (3.38).

To calculate scalar products, it is useful to first recast (3.39) in the equivalent form [31]

$$A^{\mu}_{a,\Delta} = \left(\partial_a q^{\mu} + \frac{1}{\Delta}\,q^{\mu}\partial_a\right)\Phi_{\Delta}(x;q) = \left(\partial_a q^{\mu} + \frac{1}{\Delta}\,q^{\mu}\partial_a\right)\int_0^{\infty}\frac{d\omega}{\omega}\,\omega^{\Delta}\,e^{\pm i\omega q^{\mu}x_{\mu}-0\omega}\,,\tag{3.48}$$

so that

$$F^{\mu\nu}_{a,\Delta} = \pm i\left(1 - \frac{1}{\Delta}\right)(q^{\mu}\partial_a q^{\nu} - q^{\nu}\partial_a q^{\mu})\int_0^{\infty}\frac{d\omega}{\omega}\,\omega^{\Delta+1}\,e^{\pm i\omega q^{\mu}x_{\mu}-0\omega}\,.\tag{3.49}$$

In this case the scalar product reads

$$(A, A') = -i \int d\Sigma^{\mu} \left( A^{\nu} F'^{*}_{\mu\nu} - A'^{*\nu} F_{\mu\nu} \right) \tag{3.50}$$

and we turn to evaluate it focusing for simplicity on positive-frequency wavefunctions. Choosing the $x^0 = 0$ surface and substituting the above expressions, one finds

$$(A_{a,\Delta}(x; q), A_{a',\Delta'}(x; q')) = -(2\pi)^D \delta(i(\Delta + \Delta'^{*} - D + 2))$$
$$\times \left[ \left( 1 - \frac{1}{\Delta'^{*}} \right) \left( \partial_a q^{\nu} + \frac{1}{\Delta} q^{\nu} \partial_a \right) \delta(q, q') \left( q'_0 \partial_{a'} q'_{\nu} - q'_{\nu} \partial_{a'} q'_0 \right) \right.$$
$$\left. + \left( 1 - \frac{1}{\Delta} \right) \left( \partial_{a'} q'^{\nu} + \frac{1}{\Delta'^{*}} q'^{\nu} \partial_{a'} \right) \delta(q, q') \left( q_0 \partial_a q_{\nu} - q_{\nu} \partial_a q_0 \right) \right]. \tag{3.51}$$

We now use $q^2 = 0 = q \cdot \partial_a q$, together with $\partial_a q \cdot \partial_b q = h_{ab}$, and we note that

$$q^{\nu} \partial_a \delta(q, q') \left( q'^0 \partial_{a'} q'_{\nu} - q'_{\nu} \partial_{a'} q'^0 \right) = -h_{aa'}(q) q^0 \delta(q, q'), \tag{3.52}$$

as can be checked integrating by parts with respect to $q$. We then find

$$(A_{a,\Delta}(x; q), A_{a',\Delta'}(x; q')) = h_{aa'}(q) \left( 1 - \frac{1}{\Delta} \right) \left( 1 - \frac{1}{\Delta'^{*}} \right) I(\Delta, q; \Delta', q'), \tag{3.53}$$

with $I(\Delta, q; \Delta', q')$ as in (3.19). In fact, the same result for the inner product would obtain dropping the pure gauge piece in eq. (3.46). In four dimensions, where the delta function enforces $\Delta'^{*} = 2 - \Delta$, this can be simplified to

$$(A_{a,\Delta}(x; q), A_{a',\Delta'}(x; q')) = (2\pi)^4 2q^0 h_{aa'}(q) \frac{(\Delta - 1)^2}{\Delta(\Delta - 2)} \delta(q, q') \delta(i(\Delta + \Delta'^{*} - 2)). \tag{3.54}$$

In order to compare with (A.9) of [31], we need to divide both sides by

$$(-i)^{\Delta} \Gamma(\Delta)(+i)^{\Delta'^{*}} \Gamma(\Delta'^{*}) = e^{-i\pi\Delta}(\Delta - 1) \frac{\pi}{\sin(\pi\Delta)} \tag{3.55}$$

to match the overall normalization, obtaining

$$(2\pi)^4 2q^0 h_{aa'}(q) e^{i\pi\Delta} \frac{\sin(\pi\Delta)}{\pi\Delta} \frac{\Delta - 1}{\Delta - 2} \delta(q, q') \delta(i(\Delta + \Delta'^{*} - 2)), \tag{3.56}$$

and take into account that $h_{aa'} = \delta_{aa'}$ in the parametrization (2.23). The two formulas agree (see also (3.13)).

Let us conclude by mentioning the possibility to construct a more general family of primary fields. Suppose that in addition to $x^{\mu}$, we have another $D$-vector $Z^{\mu}$ at our disposal. Then we can write down a conformal primary

$$A^{\mu}_{a,\Delta}(x, Z; q) = \left( \partial_a q^{\mu} - \frac{q^{\mu} Z \cdot \partial_a q}{q \cdot Z} \right) \Phi_{\Delta}(x, p). \tag{3.57}$$

This is a solution to field equations in Lorenz gauge but it does not obey the radial gauge $x_\mu A^\mu_{a,\Delta} = 0$. Rather, it satisfies $Z_\mu A^\mu_{a,\Delta} = 0$. In this regard, it can be compared to the generalized conformal primaries of [19]. The field strength for (3.57) is

$$F^{\mu\nu}_{a,\Delta}(x, Z; q) = (q^\mu e^\nu_a - q^\nu e^\mu_a) \frac{\Delta}{-q \cdot x}, \Phi_\Delta(x, p) \tag{3.58}$$

which is non-vanishing except for $\Delta = 0$. We may in principle use this to construct a pure gauge two form at $\Delta = 1$. We shall return on this point in the next subsection.

In a similar spirit, we could consider the gradient of a conformal primary scalar wavefunction, $\partial_\mu \Phi_\Delta(x; q)$, which also transforms as a primary vector and obeys the Lorenz gauge condition, but does not obey the "radial" gauge condition. However, it is easy to see, retracing the steps leading to (3.53), that such wavefunctions are orthogonal to any $A^\mu_{a,\Delta}(x; q)$.

As for the scalar case, we conclude this section by working out the map between creation/annihilation operators for plane waves and for CPWs. We start from the textbook Fourier-mode expansion of the vector field operator in Lorenz gauge, $\partial_\mu \boldsymbol{A}^\mu = 0$,

$$\boldsymbol{A}^\mu(x) = \frac{1}{(2\pi)^{d+1}} \int_{\mathbb{R}^d} d^d\vec{w} \int_0^\infty \omega^d \frac{d\omega}{2\omega} \left[ e^{i\omega q(\vec{w}) \cdot x} a^\mu(\omega\, q(\vec{w})) + e^{-i\omega q(\vec{w}) \cdot x} a^{\mu\dagger}(\omega\, q(\vec{w})) \right]. \tag{3.59}$$

Proceeding like we did for the scalar, we change integration variables and arrive at

$$\boldsymbol{A}^\mu(x) = \frac{1}{2(2\pi)^{d+1}} \int_{\mathbb{R}^d} d^d\vec{w} \int_{c-i\infty}^{c+i\infty} \frac{d\Delta}{i2\pi} \left[ \Phi^+_\Delta(x; \vec{w}) a^\mu_{d-\Delta}(\vec{w}) + \Phi^-_\Delta(x; \vec{w}) a^{\mu\dagger}_{d-\Delta}(\vec{w}) \right], \tag{3.60}$$

where

$$a^\mu_\Delta(\vec{w}) \equiv \int_0^\infty \omega^{\Delta-1} a^\mu(\omega\, q(\vec{w}))\, d\omega\,, \qquad a^{\mu\dagger}_\Delta(\vec{w}) \equiv \int_0^\infty \omega^{\Delta-1} a^{\mu\dagger}(\omega\, q(\vec{w}))\, d\omega \tag{3.61}$$

are the "naive" Mellin transforms of plane wave creation and annihilation operators, in complete analogy with (3.27). Our objective is to compare (3.60) to the expansion of the one-form field in terms of CPWs, according to

$$A^\mu(x) = \frac{1}{2(2\pi)^{d+1}} \int_{\mathbb{R}^d} d^d\vec{w} \int_{c-i\infty}^{c+i\infty} \frac{d\Delta}{i2\pi} \left[ A^{+\mu}_{a,\Delta}(x; \vec{w}) a_{a,d-\Delta}(\vec{w}) + A^{-\mu}_{a,\Delta}(x; \vec{w}) a^\dagger_{a,d-\Delta}(\vec{w}) \right], \tag{3.62}$$

where the CPWs are given by (3.39), which obeys both Lorenz $\partial_\mu A^\mu = 0$ and "radial" gauge $x_\mu A^\mu = 0$. To this end, we can use

$$A^\mu_{a,\Delta}(x; \vec{w}) = \left[ \partial_a q^\mu + q^\mu \frac{1}{\Delta} \partial_a \right] \Phi_\Delta(x; \vec{w}) \tag{3.63}$$

as in (3.48), equate (3.60) with (3.62) up to pure-gauge terms and take the invariant scalar product of both sides with $\Phi_{\Delta'^*}(x; q')$. Integrating over $x$ using (3.19), we find ("$\sim$" stands for equality up to pure-gauge terms)

$$a^\mu_{\Delta'}(\vec{w}') \sim \int d^d\vec{w} \left[ \left( \partial_a q^\mu + \frac{q^\mu}{d-\Delta'} \partial_a \right) \delta^{(d)}(\vec{w} - \vec{w}') \right] a_{a,\Delta'}(\vec{w})\,. \tag{3.64}$$

Performing the integral over the angles and dropping the "prime" superscripts we get

$$a_\Delta^\mu(\vec{w}) \sim \partial_a q^\mu \left(1 - \frac{1}{d-\Delta}\right) a_{a,\Delta}(\vec{w}) - \frac{q^\mu(\vec{w})}{d-\Delta} \partial^a a_{a,\Delta}(\vec{w}) \,. \tag{3.65}$$

The latter term is manifestly pure-gauge. Contracting both sides with $q_\mu(\vec{w})$ and $\partial_a q_\mu(\vec{w})$ we see that (using $h_{ab} = \delta_{ab}$)

$$q_\mu(\vec{w}) a_\Delta^\mu(\vec{w}) = 0 \,, \qquad \partial_a q_\mu(\vec{w}) a_\Delta^\mu(\vec{w}) = \left(1 - \frac{1}{d-\Delta}\right) a_{a,\Delta}(\vec{w}). \tag{3.66}$$

The latter relation can be solved for $a_{a,\Delta}$ provided $d - \Delta \neq 1$ and we obtain

$$a_{a,\Delta}(\vec{w}) = \frac{\partial_a q_\mu(\vec{w}) a_\Delta^\mu(\vec{w})}{\left(1 - \dfrac{1}{d-\Delta}\right)} \,. \tag{3.67}$$

This provides the sought-for relation between the ladder operators for conformal primary states (on the left) and Mellin transforms of ladder operators for plane wave states (on the right).

Instead of the ladder operators $a_{a,\Delta}$, it is customary to work with closely-related quantities called celestial primary operators, defined as [31]

$$\mathcal{O}_{a,\Delta}(\vec{w}) \equiv \left(A_{a,\Delta}^+(x;\vec{w}), A(x)\right). \tag{3.68}$$

Using (3.62) in (3.68) and recalling the explicit form for the inner product (3.53), we see that $\mathcal{O}_{a,\Delta}$ and $a_{a,\Delta}$ are proportional to each other,

$$\mathcal{O}_{a,\Delta}(\vec{w}) = \left(1 - \frac{1}{\Delta}\right)\left(1 - \frac{1}{\Delta'^*}\right) a_{a,d-\Delta'^*}(\vec{w})\Big|_{\Delta+\Delta'^*=d} = \left(1 - \frac{1}{\Delta}\right)\left(1 - \frac{1}{d-\Delta}\right) a_{a,\Delta}(\vec{w}) \,, \tag{3.69}$$

so that by (3.67)

$$\mathcal{O}_{a,\Delta}(\vec{w}) = \left(1 - \frac{1}{\Delta}\right) a_\Delta^\nu(\vec{w}) \partial_a q_\nu(\vec{w}) \,. \tag{3.70}$$

This equation provides the counterpart of (3.67), i.e. it links the celestial primary operator to the Mellin transform of the momentum-space ladder operators.

The definition (3.68) suggests an alternative way to arrive at (3.70). Since the scalar product is gauge invariant, we can insert in (3.68) the expansion (3.60) for $\boldsymbol{A}^\mu(x)$, (here $a_{d-\Delta'}$ is the one-form annihilation operator)

$$\mathcal{O}_{a,\Delta}(\vec{w}) = \frac{1}{2(2\pi)^{d+1}} \int_{\mathbb{R}^d} d^d\vec{w}' \int_{c-i\infty}^{c+i\infty} \frac{d\Delta'^*}{i2\pi} \left(A_{a,\Delta}^+(x,w), \Phi_{\Delta'}^+(x;\vec{w}') a_{d-\Delta'}(\vec{w}')\right). \tag{3.71}$$

The inner product $(\cdot, \cdot)$ appearing in the integrand between CPWs can be computed retracing the above steps leading to (3.53),

$$
\begin{aligned}
(\cdot, \cdot) = &-(2\pi)^D \delta(i(\Delta + \Delta'^* - d)) \\
&\times \left[ \left( \partial_a q^\nu + \frac{1}{\Delta} q^\nu \partial_a \right) \delta(q, q') \left( q_0' a_{\nu, d-\Delta'^*} - q_\nu' a_{0, d-\Delta'^*} \right) (\vec{w}') \right. \\
&\left. + \left( 1 - \frac{1}{\Delta} \right) a_{d-\Delta'^*}^\nu (\vec{w}') \delta(q, q') \left( q_0 \partial_a q_\nu - q_\nu \partial_a q_0 \right) \right].
\end{aligned}
\tag{3.72}
$$

The Lorenz gauge implies $q_\mu a_\Delta^\mu = 0$, so this simplifies to

$$
(\cdot, \cdot) = -2(2\pi)^D \delta(i(\Delta + \Delta'^* - d)) \left( 1 - \frac{1}{\Delta} \right) a_{d-\Delta'^*}^\nu (\vec{w}') \delta(q, q') q_0 \partial_a q_\nu .
\tag{3.73}
$$

Finally, recalling that $q^0 \delta(q, q') = \delta^{(d)}(\vec{w} - \vec{w}')$,

$$
\mathcal{O}_{a, \Delta}(\vec{w}) = \frac{2(2\pi)^D}{2(2\pi)^{d+1}} \int_{c-i\infty}^{c+i\infty} \frac{d\Delta'^*}{i 2\pi} \delta(i(\Delta + \Delta'^* - d)) \left( 1 - \frac{1}{\Delta} \right) a_{d-\Delta'^*}^\nu (\vec{w}) \partial_a q_\nu(\vec{w}) ,
\tag{3.74}
$$

which reduces to (3.70). In conclusion, thanks to (3.70), in order to calculate amplitudes involving conformal primary states (celestial amplitudes), one need not worry too much about the "pure gauge" difference between a true CPW and the naive Mellin transforms of a momentum-space wavefunction, e.g. the second term on the right-hand side of (3.63). One can simply calculate amplitudes involving ordinary momentum-space states projected on the $a$th polarization, and take the Mellin transform as in (3.61). The only thing to keep track of is the factor in the right-hand side of (3.70), which one can easily insert afterwards, as is usually done in these kind of calculations (see e.g. [13]).

## 3.3 Two-form

In order to discuss the the two-form case, let us begin by taking a closer look at the polarizations (2.68), denoting as above

$$
\epsilon_{ab}^{[\mu\nu]}(u; w) = \epsilon_{ab}^{[\mu\nu]}(u; q(w))
\tag{3.75}
$$

interchangeably. By eq. (2.22), these obey

$$
\epsilon_{ab}^{[\mu\nu]}(\Lambda u; w') = \Lambda^\mu{}_\rho \Lambda^\nu{}_\sigma \frac{\partial w^c}{\partial w'^a} \frac{\partial w^d}{\partial w'^b} \epsilon_{cd}^{[\rho\sigma]}(u; w)
\tag{3.76}
$$

for any Lorentz transformation $\Lambda$. When $D = 4$, a short calculation shows that,

$$
\epsilon^{[\mu\nu]}(\Lambda u; w') = \Lambda^\mu{}_\rho \Lambda^\nu{}_\sigma \epsilon^{[\rho\sigma]}(u; w) \det\left( \frac{\partial w}{\partial w'} \right),
\tag{3.77}
$$

where $\epsilon^{[\mu\nu]} = \epsilon^{[\mu\nu]}_{12}$ is the only independent polarization (2.70).

We can now consider the Mellin-space analog of (2.69), defining

$$B^{\mu\nu}_{ab,\Delta}(x;q) = \epsilon^{[\mu\nu]}_{ab}(x;q)(-q \cdot x)^2 \Phi_\Delta(x,q) \tag{3.78}$$

or more explicitly

$$B^{\mu\nu}_{ab,\Delta}(x;q) = (\mp i)^\Delta \Gamma(\Delta) \left[ \frac{\partial_a q^{[\mu} \partial_b q^{\nu]}}{(\mp i0 - q \cdot x)^\Delta} + \frac{q^{[\mu}(x \cdot \partial_a q)\,\partial_b q^{\nu]} + \partial_a q^{[\mu}(x \cdot \partial_b q)\,q^{\nu]}}{(\mp i0 - q \cdot x)^{\Delta+1}} \right], \tag{3.79}$$

which, by (3.2) and (3.77), transforms according to

$$B^{\mu\nu}_{ab,\Delta}(\Lambda x; w') = \alpha_\Lambda^{\Delta-2}(w) \Lambda^\mu{}_\sigma \Lambda^\nu{}_\rho \frac{\partial w^c}{\partial w'^a} \frac{\partial w^d}{\partial w'^b} B^{\rho\sigma}_{cd,\Delta}(x;w) \tag{3.80}$$

under Lorentz transformations. When $D = 4$, this simplifies to

$$B^{\mu\nu}_\Delta(\Lambda x; w') = \alpha_\Lambda^{\Delta-2}(w) \Lambda^\mu{}_\sigma \Lambda^\nu{}_\rho B^{\rho\sigma}_\Delta(x;w) \det\left(\frac{\partial w}{\partial w'}\right) \tag{3.81}$$

or, using eq. (2.17),

$$B^{\mu\nu}_\Delta(\Lambda x; w') = \alpha_\Lambda^\Delta(w) \sqrt{\frac{h(w')}{h(w)}} \, \Lambda^\mu{}_\sigma \Lambda^\nu{}_\rho B^{\rho\sigma}_\Delta(x;w)\,, \tag{3.82}$$

where the Jacobian determinant has canceled out. Adopting for instance the choice (2.23), for which $h(w) = h(w') = 1$, we note that (3.82) has the same conformal transformation law as the scalar (3.2). In the standard conventions of subsection 2.4, we have

$$B^{\mu\nu}_\Delta(\Lambda x; z', \bar{z}') = (cz + d)^\Delta (c^* \bar{z} + d^*)^\Delta \Lambda^\mu{}_\sigma \Lambda^\nu{}_\rho B^{\rho\sigma}_\Delta(x; z, \bar{z})\,, \tag{3.83}$$

which matches (3.3).

To calculate the field strength

$$H^{\mu\nu\rho}_{ab,\Delta} = \partial^\mu B^{\nu\rho}_{ab,\Delta} + \partial^\nu B^{\rho\mu}_{ab,\Delta} + \partial^\rho B^{\mu\nu}_{ab,\Delta}\,, \tag{3.84}$$

we note that many terms drop out thanks to (3.38), so that the end result is simply

$$H^{\mu\nu\rho}_\Delta = (\Delta - 2)(\mp i)^\Delta \Gamma(\Delta) \frac{q^\mu \epsilon^{[\nu\rho]} + q^\nu \epsilon^{[\rho\mu]} + q^\rho \epsilon^{[\mu\nu]}}{(\mp i0 - q \cdot x)^{\Delta-1}}\,. \tag{3.85}$$

Therefore, we see that the $\Delta = 2$ mode

$$B^{\mu\nu}_{a_1 a_2, \Delta=2}(x;q) = -\epsilon^{[\mu}_{a_1}(x;q)\epsilon^{\nu]}_{a_2}(x;q) \tag{3.86}$$

corresponds to a pure gauge configuration. In analogy with the one-form case, we can make this more manifest by isolating a representative two-form plus a pure gauge configuration according to

$$B_{ab,\Delta}^{\mu\nu} = \left(1 - \frac{2}{\Delta}\right) V_{ab,\Delta}^{\mu\nu} + \partial^{[\mu} \left(\frac{1}{\Delta} x \cdot V_{ab,\Delta}^{\nu]}\right), \tag{3.87}$$

where

$$V_{ab,\Delta}^{\mu\nu} = (\mp i)^{\Delta} \Gamma(\Delta) \frac{\partial_a q^{\mu} \partial_b q^{\nu} - \partial_a q^{\nu} \partial_b q^{\mu}}{(\mp i0 - q \cdot x)^{\Delta}}, \qquad x \cdot V_{ab,\Delta}^{\nu} = x_{\alpha} V_{ab,\Delta}^{\alpha\nu}. \tag{3.88}$$

The equations of motion are once again satisfied. In fact, all tensors that we have been considering throughout are built out of terms the form

$$T^{\mu\cdots\nu\alpha\cdots\beta} = q^{\mu} \cdots q^{\nu} \partial_a q^{\alpha} \cdots \partial_b q^{\beta} \, f(x \cdot q) \tag{3.89}$$

with $f$ a scalar function. Therefore they are divergence-free and obey $\Box T^{\mu\cdots\nu\alpha\cdots\beta} = 0$.

To evaluate the scalar products, for generic $D$, it is convenient to recast $B_{ab,\Delta}^{\mu\nu}$ in the form

$$B_{ab,\Delta}^{\mu\nu}(x;q) = \left(e_{ab}^{[\mu\nu]} + \frac{1}{\Delta} \left(q^{[\mu} \partial_b q^{\nu]} \partial_a + q^{[\nu} \partial_a q^{\mu]} \partial_b\right)\right) \int_0^{\infty} \frac{d\omega}{\omega} \omega^{\Delta} e^{\pm i\omega q \cdot x - 0\omega}, \tag{3.90}$$

with

$$e_{ab}^{[\mu\nu]} = \partial_a q^{\mu} \partial_b q^{\nu} - \partial_a q^{\nu} \partial_b q^{\mu}, \tag{3.91}$$

and the field strength $H_{\Delta}^{ab,\mu\nu\rho}$ in the form

$$H_{ab,\Delta}^{\mu\nu\rho}(x;q) = \pm i \left(1 - \frac{2}{\Delta}\right) \left(q^{\mu} e_{ab}^{[\nu\rho]} + q^{\nu} e_{ab}^{[\rho\mu]} + q^{\rho} e_{ab}^{[\mu\nu]}\right) \int_0^{\infty} \frac{d\omega}{\omega} \omega^{\Delta+1} e^{\pm i\omega q \cdot x - 0\omega}. \tag{3.92}$$

We then substitute into

$$(B, B') = -\frac{i}{2!} \int d\Sigma^{\mu} \left(B^{\rho\sigma} H'^{*}_{\mu\rho\sigma} - B'^{*\rho\sigma} H_{\mu\rho\sigma}\right), \tag{3.93}$$

and follow very similar steps compared to the scalar and one-form cases. We are thus led to

$$(B_{ab,\Delta}(x;q), B_{a'b',\Delta'}(x;q')) = -\frac{1}{2!} (2\pi)^D \delta(i(\Delta + \Delta'^* - D + 2)) \tag{3.94}$$

$$\left[\left(1 - \frac{2}{\Delta'^*}\right)\left(e_{ab}^{[\rho\sigma]} + \frac{1}{\Delta}\left(q^{[\rho} \partial_b q^{\sigma]} \partial_a + q^{[\sigma} \partial_a q^{\rho]} \partial_b\right)\right) \delta(q,q') \left(q'_0 e'_{a'b'[\rho\sigma]} + q'_\rho e'_{a'b'[\sigma 0]} + q'_\sigma e'_{a'b'[0\rho]}\right)\right.$$

$$\left. + \left(1 - \frac{2}{\Delta}\right)\left(e'^{[\rho\sigma]}_{a'b'} + \frac{1}{\Delta'^*}\left(q'^{[\rho} \partial'_{b'} q'^{\sigma]} \partial'_{a'} + q'^{[\sigma} \partial'_{a'} q'^{\rho]} \partial'_{b'}\right)\right) \delta(q,q') \left(q_0 e_{ab[\rho\sigma]} + q_\rho e_{ab[\sigma 0]} + q_\sigma e_{ab[0\rho]}\right)\right]$$

and thus, using integration by parts to simplify the action of the derivative on the delta function,

$$(B_{ab,\Delta}(x;q), B_{a'b',\Delta'}(x;q')) = I(\Delta, q; \Delta', q')\left(1 - \frac{2}{\Delta}\right)\left(1 - \frac{2}{\Delta'^*}\right)(e_{ab}, e_{a'b'}), \tag{3.95}$$

with $I(\Delta, q; \Delta', q')$ as in (3.19) and

$$(e_{ab}, e_{a'b'}) = \frac{1}{2!}\, e_{ab}^{[\mu\nu]}\, e_{a'b'[\mu\nu]} = \det\begin{pmatrix} h_{aa'} & h_{ab'} \\ h_{ba'} & h_{bb'} \end{pmatrix}. \tag{3.96}$$

As for the one-form inner product, the result (3.95) is insensitive to the pure gauge part in the decomposition (3.87). When $D = 4$, one can also use the delta function $\delta(i(\Delta + \Delta'^* - 2))$ to simplify the scalar product as follows,

$$(B_\Delta(x; q), B_{\Delta'}(x; q')) = I(\Delta, q; \Delta', q') \det(h), \tag{3.97}$$

which is equal to the inner product for scalar CPWs when we choose parametrizations such as (2.23) where $\det(h) = 1$.

In principle, additional pure gauge "two-form CPWs" could be built by acting with an exterior derivative on one-form CPWs, i.e. considering $B_{a,\Delta}^{\mu\nu} = \partial^{[\mu} A_{a,\Delta}^{\nu]}$ with arbitrary $\Delta$. These two-forms obey Lorenz gauge, but not radial gauge. However, the scalar product $(F_{a,\Delta}(x; q), B_{\Delta'}(x; q'))$, with $F_{a,\Delta}^{\mu\nu}$ as in (3.49), is identically zero because

$$(q^\rho e_a^\sigma - q^\sigma e_a^\rho)\,(q_\mu e_{\rho\sigma} + q_\rho e_{\sigma\mu} + q_\sigma e_{\mu\rho}) = 0. \tag{3.98}$$

The generalized one-form primaries introduced at the end of the previous subsection (3.57) have the same field strength as the standard ones (up to a constant relative factor). Therefore, the property (3.98) holds for them as well. In conclusion, both types of pure gauge two-form CPWs built out of them are always orthogonal to the CPWs (3.78).

We now turn to the map between creation/annihilation operators for plane wave and CPW states. This can be worked out following the same logic employed for the one-form in the previous subsection and using the same technical tools employed in the rest of the present section.

$$\boldsymbol{B}_{\mu\nu}(x) = \frac{1}{(2\pi)^{d+1}} \int_{\mathbb{R}^d} d^d\vec{w} \int_0^\infty \omega^d \frac{d\omega}{2\omega} \left[ e^{i\omega q(\vec{w})\cdot x} a_{\mu\nu}(\omega\, q(\vec{w})) + e^{-i\omega q(\vec{w})\cdot x} a_{\mu\nu}^\dagger(\omega\, q(\vec{w})) \right], \quad (3.99)$$

and

$$B^{\mu\nu}(x) = \frac{1}{4(2\pi)^{d+1}} \int_{\mathbb{R}^d} d^d\vec{w} \int_{c-i\infty}^{c+i\infty} \frac{d\Delta}{i2\pi} \left[ B_{ab,\Delta}^{+\mu\nu}(x; \vec{w}) a_{ab,d-\Delta}(\vec{w}) + B_{ab,\Delta}^{-\mu\nu}(x; \vec{w}) a_{ab,d-\Delta}^\dagger(\vec{w}) \right], \tag{3.100}$$

Using (3.90) we find

$$a_\Delta^{\mu\nu}(\vec{w}) \sim \frac{1}{2}\partial_a q^{[\mu}\partial_b q^{\nu]}\left(1 - \frac{2}{d-\Delta}\right) a_{ab,\Delta}(\vec{w}) - \frac{q^{[\mu}\partial_b q^{\nu]}}{d-\Delta}\partial_a a_{ab,\Delta}(\vec{w}), \tag{3.101}$$

so that

$$a_{ab,\Delta}(\vec{w}) = \frac{e_{ab,[\mu\nu]}(\vec{w})\, a_\Delta^{\mu\nu}(\vec{w})}{2\left(1 - \frac{2}{d-\Delta}\right)} \tag{3.102}$$

and similarly

$$\mathcal{O}_{ab,\Delta}(\vec{w}) = \left(B_{ab,\Delta}^{+}(x;\vec{w}), B(x)\right) = \frac{1}{2}\left(1 - \frac{2}{\Delta}\right) e_{ab,[\mu\nu]}(\vec{w})\, a_{\Delta}^{\mu\nu}(\vec{w})\,. \tag{3.103}$$

This extends the dictionary to convert insertions of momentum-space states into the corresponding ones of conformal-primary states to the two-form case.

## 3.4 Higher forms

The extension of the previous construction to anti-symmetric tensors of rank $p$ can be achieved naturally defining polarizations

$$\epsilon_{a_1\cdots a_p}^{[\mu_1\cdots\mu_p]}(x;q) = \epsilon_{a_1}^{[\mu_1}(x;q)\cdots\epsilon_{a_p}^{\mu_p]}(x;q) \tag{3.104}$$

and conformal primaries

$$B_{a_1\cdots a_p,\Delta}^{\mu_1\cdots\mu_p}(x;q) = \epsilon_{a_1\cdots a_p}^{[\mu_1\cdots\mu_p]}(x;q)(-q\cdot x)^p \Phi_\Delta(x;q), \tag{3.105}$$

or more explicitly

$$B_{a_1\cdots a_p,\Delta}^{\mu_1\cdots\mu_p}(x;q) = (\mp i)^\Delta \Gamma(\Delta)\left[\frac{e_{a_1}^{[\mu_1}\cdots e_{a_p}^{\mu_p]}}{(\mp i0 - q\cdot x)^\Delta} + \sum_{k=1}^{p}\frac{x\cdot e_{a_k}\, q^{[\mu_k} e_{a_1}^{\mu_1}\cdots\widehat{e_{a_k}^{\mu_k}}\cdots e_{a_p}^{\mu_p]}}{(\mp i0 - q\cdot x)^{\Delta+1}}\right] \tag{3.106}$$

where in the second term, the "hat" notation means that $e_{a_k}^{\mu_k}$ is omitted from the product. As for the two-form, these fields trivially obey the equations of motion as noted below (3.89). An equivalent expression is obtained by isolating a $p$-form "Mellin representative" and a pure gauge part

$$B_{a_1\cdots a_p,\Delta}^{\mu_1\cdots\mu_p}(x;q) = \left(1 - \frac{p}{\Delta}\right) V_{a_1\cdots a_p,\Delta}^{\mu_1\cdots\mu_p} + \partial^{[\mu_1}\left(\frac{1}{(p-1)!}\frac{1}{\Delta}x\cdot V_{a_1\cdots a_p,\Delta}^{\mu_2\cdots\mu_p]}\right) \tag{3.107}$$

where

$$V_{a_1\cdots a_p,\Delta}^{\mu_1\cdots\mu_p} = (\mp i)^\Delta \Gamma(\Delta)\frac{e_{a_1}^{[\mu_1}\cdots e_{a_p}^{\mu_p]}}{(\mp i0 - q\cdot x)^\Delta}\,, \qquad x\cdot V_{a_1\cdots a_p,\Delta}^{\mu_2\cdots\mu_p} = x_\alpha V_{a_1\cdots a_p,\Delta}^{\alpha\mu_2\cdots\mu_p}\,. \tag{3.108}$$

The corresponding field strength is easy to evaluate by remembering eqs. (3.38) and reads

$$H_{a_1\cdots a_p,\Delta}^{\alpha\mu_1\cdots\mu_p}(x;q) = (\Delta - p)(\mp i)^\Delta \Gamma(\Delta)\frac{q^{[\alpha} e_{a_1}^{\mu_1}\cdots e_{a_p}^{\mu_p]}}{(\mp i0 - q\cdot x)^{\Delta+1}}\,. \tag{3.109}$$

Thus, as one can also observe from (3.107), $p$-form CPWs of conformal dimension $\Delta = p$ are pure gauge; see Table 1 below. This holds independently of the dimension $d$ of the celestial sphere, in contrast with the case of shadow $p$-forms, as we will see in section 4.3.

In order to evaluate scalar products, it is again convenient to trade the $x$-dependence in the polarization for a derivative

$$B^{\mu_1\cdots\mu_p}_{a_1\cdots a_p,\Delta}(x;w) = \left[e^{[\mu_1\cdots\mu_p]}_{a_1\cdots a_p} + \frac{1}{\Delta}\sum_{k=1}^{p} q^{[\mu_k}e^{\mu_1}_{a_1}\cdots\widehat{e^{\mu_k}_{a_k}}\cdots e^{\mu_p]}_{a_p}\partial_{a_k}\right]\int_0^\infty \frac{d\omega}{\omega}\,\omega^\Delta e^{\pm i\omega q\cdot x - 0\omega} \tag{3.110}$$

and similarly for the field strength

$$H^{\alpha\mu_1\cdots\mu_p}_{a_1\cdots a_p,\Delta}(x;w) = \pm i\left(1 - \frac{p}{\Delta}\right)q^{[\alpha}e^{\mu_1}_{a_1}\cdots e^{\mu_p]}_{a_p}\int_0^\infty \frac{d\omega}{\omega}\,\omega^{\Delta+1} e^{\pm i\omega q\cdot x - 0\omega}\,. \tag{3.111}$$

This highlights that $\Delta = p$ mode

$$B^{\mu\nu}_{a_1\cdots a_p,\Delta=p}(x;q) = (\mp i)^p\Gamma(p)\epsilon^{[\mu_1}_{a_1}(x;q)\cdots\epsilon^{\mu_p]}_{a_p}(x;q) \tag{3.112}$$

is a pure-gauge wavefunction. The inner product

$$(B,B') = -\frac{i}{p!}\int d\Sigma^\mu(B^{\rho_1\cdots\rho_p}H'^*_{\mu\rho_1\cdots\rho_p} - B'^{*\rho_1\cdots\rho_p}H_{\mu\rho_1\cdots\rho_p}) \tag{3.113}$$

thus takes the form

$$(B_{a_1\cdots a_p,\Delta}(x;q), B_{a_1'\cdots a_p',\Delta'}(x;q')) = -\frac{1}{p!}(2\pi)^D\,\delta(i(\Delta + \Delta'^* - D + 2))$$
$$\left[\left(1 - \frac{p}{\Delta'^*}\right)\left(e^{[\rho_1\cdots\rho_p]}_{a_1\cdots a_p} + \frac{1}{\Delta}\sum_{k=1}^{p} q^{[\rho_k}e^{\rho_1}_{a_1}\cdots\widehat{e^{\rho_k}_{a_k}}\cdots e^{\rho_p]}_{a_p}\partial_{a_k}\delta(q,q')q'_{[0|}e'_{a_1'\cdots a_p'|\rho_1\cdots\rho_p]}\right)\right.$$
$$\left. + \left(1 - \frac{p}{\Delta}\right)\left(e'^{[\rho_1\cdots\rho_p]}_{a_1'\cdots a_p'} + \frac{1}{\Delta'^*}\sum_{k=1}^{p} q'^{[\rho_k}e'^{\rho_1}_{a_1'}\cdots\widehat{e'^{\rho_k}_{a_k'}}\cdots e'^{\rho_p]}_{a_p'}\partial'_{a_k'}\delta(q,q')q_{[0|}e_{a_1\cdots a_p|\rho_1\cdots\rho_p]}\right)\right] \tag{3.114}$$

Derivatives acting on the delta function can be simplified integrating by parts and this leads to

$$(B_{a_1\cdots a_p,\Delta}(x;q), B_{a_1'\cdots a_p',\Delta'}(x;q')) = \left(1 - \frac{p}{\Delta}\right)\left(1 - \frac{p}{\Delta'^*}\right)I(\Delta,q;\Delta',q')\,(e_{a_1\cdots a_p}, e_{a_1'\cdots a_p'})\,, \tag{3.115}$$

where $I(\Delta,q;\Delta',q')$ is defined in (3.19) and

$$(e_{a_1\cdots a_p}, e_{a_1'\cdots a_p'}) = \frac{1}{p!}e^{[\mu_1\cdots\mu_p]}_{a_1\cdots a_p}e_{a_1'\cdots a_p'[\mu_1\cdots\mu_p]}\,. \tag{3.116}$$

Eq. (3.115) provides the generalization of (3.95) to generic form degree, and, in analogy with one- and two-forms, the same result would obtain dropping the pure-gauge term in the decomposition (3.107). The result can be simplified by noting that, letting $e^a_\mu = h^{ab}\eta_{\mu\nu}e^\nu_b$,

$$(e^{[\mu_1}_{b_1}\cdots e^{\mu_p]}_{b_p})(e^{a_1}_{\mu_1}\cdots e^{a_p}_{\mu_p}) = \delta^{a_1}_{[b_1}\cdots\delta^{a_p}_{b_p]}\,. \tag{3.117}$$

We conclude by remarking that, when $D = 2p + 2$, the scalar product (3.115) can be written in the form

$$\left(B_{a_1 \cdots a_p, \Delta}(x; q), B_{a'_1 \cdots a'_p, \Delta'}(x; q')\right) = \frac{(\Delta - p)^2}{\Delta(\Delta - 2p)} I(\Delta, q; \Delta', q') \left(e_{a_1 \cdots a_p}, e_{a'_1 \cdots a'_p}\right) \qquad (3.118)$$

which highlights the pure gauge mode $\Delta = p$, in analogy with the expression in eq. (3.54) for the one-form. Note that $D = 2p + 2$ can be also regarded as a "critical dimension" for the $p$-form theory [46, 47], since it is the dimension for which the asymptotic behaviors of radiative and Coulombic solutions coincide [48].

Comparing plane wave and CP decompositions of the operators

$$\boldsymbol{B}_{\mu_1 \cdots \mu_p}(x) = \frac{1}{(2\pi)^{d+1}} \int_{\mathbb{R}^d} d^d \vec{w} \int_0^\infty \omega^d \frac{d\omega}{2\omega} \left[ e^{i\omega q(\vec{w}) \cdot x} a_{\mu_1 \cdots \mu_p}(\omega \, q(\vec{w})) + e^{-i\omega q(\vec{w}) \cdot x} a^\dagger_{\mu_1 \cdots \mu_p}(\omega \, q(\vec{w})) \right],$$

$$(3.119)$$

and

$$\begin{aligned} B^{\mu_1 \cdots \mu_p}(x) = \frac{1}{2p!(2\pi)^{d+1}} \int_{\mathbb{R}^d} d^d \vec{w} \int_{c-i\infty}^{c+i\infty} \frac{d\Delta}{i2\pi} \Big[ & B^{+\mu_1 \cdots \mu_p}_{a_1 \cdots a_p, \Delta}(x; \vec{w}) a_{a_1 \cdots a_p, d-\Delta}(\vec{w}) \\ & + B^{-\mu_1 \cdots \mu_p}_{a_1 \cdots a_p, \Delta}(x; \vec{w}) a^\dagger_{a_1 \cdots a_p, d-\Delta}(\vec{w}) \Big], \end{aligned} \qquad (3.120)$$

we deduce

$$\mathcal{O}_{a_1 a_2 \cdots a_p, \Delta}(\vec{w}) = \left(B^+_{a_1 a_2 \cdots a_p, \Delta}(x; \vec{w}), B(x)\right) = \frac{1}{p!} \left(1 - \frac{p}{\Delta}\right) e_{a_1 a_2 \cdots a_p, [\mu_1 \mu_2 \cdots \mu_p]}(\vec{w}) \, a_\Delta^{\mu_1 \mu_2 \cdots \mu_p}(\vec{w}),$$

$$(3.121)$$

which completes our dictionary to convert insertions of momentum-space states into the corresponding ones of conformal-primary states.

## 3.5   Hodge duality and self-duality

We want to show that, using the parameterization (2.23), if $B_{a_1 \cdots a_p, \Delta}$ is a $p$-form conformal primary in $D = d + 2$ dimensions with field strength $H_{a_1 \cdots a_p, \Delta} = dB_{a_1 \cdots a_p, \Delta}$ then

$$\frac{1}{\Delta - p} * H_{a_1 \cdots a_p, \Delta} = -\frac{1}{(d - p)!(\Delta - d + p)} \epsilon_{a_1 \cdots a_p}{}^{b_1 \cdots b_{d-p}} H_{b_1 \cdots b_{d-p}, \Delta}. \qquad (3.122)$$

That is to say, Hodge duality maps a $p$-form conformal primary to a $(d - p)$-form with dualized polarization.

To this end, let us first decompose the anti-symmetric tensor according to

$$\epsilon^{\alpha \mu_1 \cdots \mu_d \beta} = n^{[\alpha} e_1^{\mu_1} \cdots e_d^{\mu_d} q^{\beta]} = \frac{1}{d!} n^{[\alpha} e_{b_1}^{\mu_1} \cdots e_{b_d}^{\mu_d} q^{\beta]} \epsilon^{b_1 \cdots b_d}, \qquad n^\mu = (1, 0, \cdots, 0, -1), \qquad (3.123)$$

which can be proved by

$$e_a^\mu = n^\mu w_a + \delta_a^\mu, \qquad q \cdot n = -1. \tag{3.124}$$

Alternatively, we may first check (3.123) in the $w^a = 0$ case, and then apply the transformation $R$ given in (2.34) to obtain the general case. Then, it is sufficient to note that

$$\det \begin{pmatrix} 1 & 0 & -1 \\ 0 & \delta_a^b & 0 \\ \frac{1}{2} & 0 & \frac{1}{2} \end{pmatrix} = 1. \tag{3.125}$$

By eq. (3.109), in order to retrieve the duality (3.122), it is sufficient to show that

$$\frac{1}{(p+1)!} \epsilon_{\alpha \mu_1 \cdots \mu_d \beta} \, q^{[\alpha} e_{a_1}^{\mu_1} \cdots e_{a_p}^{\mu_p]} = -\frac{1}{(d-p)!} \epsilon_{a_1 \cdots a_p b_{p+1} \cdots b_d} \, q_{[\beta} e_{\mu_{p+1}}^{b_{p+1}} \cdots e_{\mu_d]}^{b_d} . \tag{3.126}$$

This can be seen using (3.123) according to the following steps:

$$\begin{aligned}
\frac{1}{(p+1)!} \epsilon_{\alpha \mu_1 \cdots \mu_d \beta} \, q^{[\alpha} e_{a_1}^{\mu_1} \cdots e_{a_p}^{\mu_p]} &= \frac{1}{d!} \left( n_{[\alpha} e_{\mu_1}^{b_1} \cdots e_{\mu_d}^{b_d} q_{\beta]} \, \epsilon_{b_1 \cdots b_d} \right) \left( q^\alpha e_{a_1}^{\mu_1} \cdots e_{a_p}^{\mu_p} \right) \\
&= -\frac{1}{d!} \left( e_{[\mu_1}^{b_1} \cdots e_{\mu_d}^{b_d} q_{\beta]} \, \epsilon_{b_1 \cdots b_d} \right) \left( e_{a_1}^{\mu_1} \cdots e_{a_p}^{\mu_p} \right) \\
&= -\frac{1}{d!} \delta_{a_1}^{[b_1} \cdots \delta_{a_p}^{b_p} e_{[\mu_{p+1}}^{b_{p+1}} \cdots e_{\mu_d]}^{b_d]} q_{\beta]} \, \epsilon_{b_1 \cdots b_d} \frac{1}{(d-p)!} \\
&= -\frac{1}{(d-p)!} \epsilon_{a_1 \cdots a_p b_{p+1} \cdots b_d} e_{[\mu_{p+1}}^{b_{p+1}} \cdots e_{\mu_d}^{b_d} q_{\beta]} .
\end{aligned} \tag{3.127}$$

In the third line, the factor $1/(d-p)!$ compensates for double anti-symmetrization in $\mu_{p+1} \cdots \mu_d$.

Let us apply this duality to a few examples, especially, to critical dimensions where self-dual forms exist [49]. For one-forms in four dimensions we have

$$\frac{1}{2} \epsilon_{\mu\nu}{}^{\alpha\beta} F_{a,\Delta}^{\mu\nu} = -\epsilon_a{}^b F_{b,\Delta}^{\alpha\beta} . \tag{3.128}$$

In complex coordinates of subsection 2.4, we have $\epsilon_{z\bar{z}} = i/2$ and $\epsilon^{\bar{z}}{}_{\bar{z}} = \epsilon_z{}^z = i$. As a result,

$$\frac{1}{2} \epsilon_{\mu\nu}{}^{\alpha\beta} F_{z,\Delta}^{\mu\nu} = -i F_{z,\Delta}^{\alpha\beta} \tag{3.129}$$

is an anti-self-dual form, while

$$\frac{1}{2} \epsilon_{\mu\nu}{}^{\alpha\beta} F_{\bar{z},\Delta}^{\mu\nu} = i F_{\bar{z},\Delta}^{\alpha\beta} \tag{3.130}$$

is self-dual. Another example we have already encountered is that of a two-form in four dimensions and in real coordinates $(z^1, z^2)$,

$$\frac{1}{3!(\Delta-2)} \epsilon_{\mu\nu\rho\sigma} H_\Delta^{\mu\nu\rho} = -\frac{1}{\Delta} \partial_\sigma \Phi_\Delta , \tag{3.131}$$

where $H_\Delta^{\mu\nu} = H_{12,\Delta}^{\mu\nu}$ as in (6.12).

A two-form in six dimensions has 6 degrees of freedom. Defining two pairs of complex coordinates $w, \bar{w}, z, \bar{z}$, they are $\{w\bar{w}, z\bar{z}, w\bar{z}, \bar{w}z, \bar{w}\bar{z}, wz\}$. From $\epsilon_{w\bar{w}z\bar{z}} = -4$ we have for instance

$$\frac{1}{3!}\epsilon_{\mu\nu\rho}{}^{\sigma\alpha\beta}H_{wz,\Delta}^{\mu\nu\rho} = -H_{wz,\Delta}^{\sigma\alpha\beta}. \tag{3.132}$$

$\{wz, \bar{w}\bar{z},\}$ are anti-self-dual while $\{\bar{w}z, \bar{z}w\}$ are self-dual. Finally, $H_{w\bar{w}} \leftrightarrow H_{z\bar{z}}$, hence $H_{w\bar{w}} \pm H_{z\bar{z}}$ is self-dual with positive sign and anti-self-dual with negative sign. Similarly, $p$-form CPWs in $D = 2p + 2$ dimensions are mapped to $p$-form CPWs by the duality, so that one can organize them in $\frac{1}{2}\binom{2p}{p}$ self-dual and $\frac{1}{2}\binom{2p}{p}$ anti-self-dual degrees of freedom.

# 4 Shadow transforms

In this section we construct the shadow transforms of the $p$-form conformal primaries obtained in the previous section. To this end, we also describe their uplifts obtained via the ambient-space construction, whereby polarization indices $a$, $b$ are promoted to spacetime indices up to suitable projections. From now on, we shall systematically employ the convenient parametrization (2.23) and the shorthand notation

$$d = D - 2. \tag{4.1}$$

We also highlight (Euclidean) $d$-vectors using an arrow, writing for instance $q(\vec{w})$.

## 4.1 Scalar shadows

We start by revisiting the construction of [2] for the shadow scalar conformal primary wave-functions. We use the following definition of the shadow transform (see [50, 51] for early works and [2, 52–55] for a more recent literature)

$$\widetilde{\Phi}_\Delta(x; \vec{w}) = \int \frac{d^d\vec{w}'}{|\vec{w} - \vec{w}'|^{2(d-\Delta)}} \, \Phi_\Delta(x; \vec{w}'). \tag{4.2}$$

This non-local integral transform maps the scalar CPW $\Phi_\Delta$ (3.1) to an operator $\widetilde{\Phi}_\Delta$ of conformal dimension $d - \Delta$. Notice that, with the present normalization, the shadow transform does not square to unity, but instead leads to the following proportionality factor:

$$\widetilde{\widetilde{\Phi}}_\Delta = \frac{\pi^d \Gamma(\frac{d}{2} - \Delta)\Gamma(\Delta - \frac{d}{2})}{\Gamma(\Delta)\Gamma(d - \Delta)} \Phi_\Delta. \tag{4.3}$$

Letting for brevity $q = q(\vec{w})$, $q' = q(\vec{w}')$ and using the identity

$$-2q \cdot q' = |\vec{w} - \vec{w}'|^2, \tag{4.4}$$

the shadow (4.2) reads

$$\widetilde{\Phi}_\Delta(x; q) = \int \frac{d^d \vec{w}'}{(-2q \cdot q')^{d-\Delta}} \, \Phi_\Delta(x; q') \,, \tag{4.5}$$

whose calculation involves the integral

$$S_\Delta(x; q) \equiv \int \frac{d^d \vec{w}'}{|\vec{w} - \vec{w}'|^{2(d-\Delta)}(-2q' \cdot x)^\Delta} \,. \tag{4.6}$$

Since

$$-2q' \cdot x = x^+ |\vec{w}'|^2 - 2\vec{x} \cdot \vec{w}' + x^-, \qquad x^\pm = x^0 \pm x^{d+1}, \tag{4.7}$$

this can be recast in the form

$$S_\Delta(x; q) = \frac{1}{(x^+)^\Delta} \int \frac{d^d \vec{w}'}{(|\vec{w} - \vec{w}'|^2)^{d-\Delta}(|\vec{w}'|^2 - 2\frac{\vec{x}}{x^+} \cdot \vec{w}' + \frac{x^-}{x^+})^\Delta} \,. \tag{4.8}$$

This can be simplified introducing Schwinger parameters and performing the Gaussian integral over $\vec{w}'$, which gives

$$S_\Delta(x; q) = \frac{1}{(x^+)^\Delta} \int_{\mathbb{R}^2_+} \frac{dt_1 dt_2}{t_1 t_2} \frac{t_1^{d-\Delta} t_2^\Delta}{\Gamma(d - \Delta)\Gamma(\Delta)} \frac{\pi^{\frac{d}{2}}}{(t_1 + t_2)^{\frac{d}{2}}} e^{-\frac{1}{t_1 + t_2}\left(t_1 t_2 \frac{(-2q \cdot x)}{x^+} + t_2^2 \frac{(-x^2)}{(x^+)^2}\right)} \,. \tag{4.9}$$

Changing variables according to $t_1 = \lambda$, $t_2 = \lambda t$ and carrying out the integration over $\lambda$ then yields

$$S_\Delta(x; q) = \frac{(-x^2)^{-\frac{d}{2}}}{(x^+)^{\Delta-d}} \frac{\pi^{\frac{d}{2}} \Gamma(\frac{d}{2})}{\Gamma(d - \Delta)\Gamma(\Delta)} \int_0^\infty \frac{dt}{t} \frac{t^{\Delta - \frac{d}{2}}}{\left(t + \frac{(-2q \cdot x)}{(-x^2)} x^+\right)^{\frac{d}{2}}} \,. \tag{4.10}$$

The last integral can be reduced to the Euler Beta function by suitably rescaling $t$ and then letting $t = \frac{1-s}{s}$, which finally leads to

$$S_\Delta(x; q) = \frac{(-x^2)^{\frac{d}{2}-\Delta}}{(-2q \cdot x)^{d-\Delta}} \frac{\pi^{\frac{d}{2}} \Gamma(\Delta - \frac{d}{2})}{\Gamma(\Delta)} \,. \tag{4.11}$$

Using this basic result and the definition (3.1), provided $x^+ > 0$, we obtain

$$\widetilde{\Phi}_\Delta(x; \vec{w}) = \pi^{\frac{d}{2}}(-i0 - x^2)^{\frac{d}{2}-\Delta} \frac{(\mp 2i)^\Delta \Gamma(\Delta - \frac{d}{2})}{(\mp i0 - 2q(\vec{w}) \cdot x)^{d-\Delta}} \,, \tag{4.12}$$

recovering (up to the different choice for the normalization factor) expressions given in [2, 52]. One can check that $\widetilde{\Phi}_\Delta$ is indeed a solution of the equation of motion and behaves as a CPW with weight $d - \Delta$. The more general case where $x^+$ can be positive or negative is instead captured by

$$\widetilde{\Phi}_\Delta(x; \vec{w}) = \frac{\pi^{\frac{d}{2}}(-i0x^+ - x^2)^{\frac{d}{2}-\Delta}}{[\mathrm{sgn}(x^+)]^d} \frac{(\mp 2i)^\Delta \Gamma(\Delta - \frac{d}{2})}{(\mp i0 - 2q(\vec{w}) \cdot x)^{d-\Delta}} \,. \tag{4.13}$$

This is crucial in order to ensure that the antipodal mapping $x \to -x$ leaves $\widetilde{\Phi}_\Delta$ unmodified up to reversing the $\mp i0$ prescription (i.e. interchanging incoming with outgoing states), as apparent from (3.1) and (4.2). Note that $x^+ = -n \cdot x$ where $n^\mu = (1, 0, \ldots, 0, -1)$ is the vector characterizing our preferred choice (2.23) of slicing of the light cone, $-n \cdot q = 1$.

To compute scalar products involving shadow transforms, one can proceed as follows. Let us first turn to

$$\left(\Phi_\Delta(x; \vec{w}), \widetilde{\Phi}_{\Delta'}(x; \vec{w}')\right) = \int \frac{d^d \vec{z}'}{|\vec{z}' - \vec{w}'|^{2(d-\Delta'^*)}} \left(\Phi_\Delta(x; \vec{w}), \Phi_{\Delta'}(x; \vec{z}')\right). \tag{4.14}$$

Using (3.18),

$$\left(\Phi_\Delta(x; \vec{w}), \Phi_{\Delta'}(x; \vec{w}')\right) = 2(2\pi)^D \, \delta^{(d)}(\vec{w} - \vec{w}')\delta(i(\Delta + \Delta'^* - d)) \tag{4.15}$$

when specialized to (2.23), using in particular (3.13), we immediately get

$$\left(\Phi_\Delta(x; \vec{w}), \widetilde{\Phi}_{\Delta'}(x, \vec{w}')\right) = \frac{2(2\pi)^D}{|\vec{w} - \vec{w}'|^{2\Delta}} \, \delta(i(\Delta + \Delta'^* - d)). \tag{4.16}$$

A similar approach can be adopted to calculate

$$\left(\widetilde{\Phi}_\Delta(x; \vec{w}), \widetilde{\Phi}_{\Delta'}(x; \vec{w}')\right) = \int \frac{d^d \vec{z}}{|\vec{z} - \vec{w}|^{2(d-\Delta)}} \int \frac{d^d \vec{z}'}{|\vec{z}' - \vec{w}'|^{2(d-\Delta'^*)}} \left(\Phi_\Delta(x; \vec{z}), \Phi_{\Delta'}(x; \vec{z}')\right), \tag{4.17}$$

obtaining

$$\left(\widetilde{\Phi}_\Delta(x; \vec{w}), \widetilde{\Phi}_{\Delta'}(x; \vec{w}')\right) = 2(2\pi)^D \delta(i(\Delta + \Delta'^* - d)) \int \frac{d^d \vec{z}}{|\vec{z}|^{2(d-\Delta)}|\vec{z} + \vec{w} - \vec{w}'|^{2\Delta}}. \tag{4.18}$$

Starting from

$$\int \frac{d^d \vec{z}}{|\vec{z}|^{2\alpha}|\vec{z} + \vec{w} - \vec{w}'|^{2\beta}} = \frac{\Gamma(\alpha + \beta - \frac{d}{2})}{|\vec{w} - \vec{w}'|^{2(\alpha+\beta-\frac{d}{2})}} \frac{\Gamma(\frac{d}{2} - \alpha)\Gamma(\frac{d}{2} - \beta)}{\Gamma(\alpha)\Gamma(\beta)} \frac{\pi^{\frac{d}{2}}}{\Gamma(d - \alpha - \beta)}. \tag{4.19}$$

setting $\alpha = d - \Delta - \frac{\lambda}{2}$, $\beta = \Delta$, and considering the $\lambda \to 0$ limit, we find

$$\int \frac{d^d \vec{z}}{|\vec{z}|^{2(d-\Delta)-\lambda}|\vec{z} + \vec{w} - \vec{w}'|^{2\Delta}} \sim \frac{\Gamma(\frac{d}{2})\Gamma(\Delta - \frac{d}{2})\Gamma(\frac{d}{2} - \Delta)}{2\Gamma(d - \Delta)\Gamma(\Delta)} \pi^{\frac{d}{2}} \lambda \, |\vec{w} - \vec{w}'|^{\lambda - d}. \tag{4.20}$$

Now, using (A.7), which implies to leading order

$$\lim_{\lambda \to 0} \lambda |\vec{w} - \vec{w}'|^{\lambda - d} = \frac{2\pi^{\frac{d}{2}}}{\Gamma(\frac{d}{2})} \delta^{(d)}(\vec{w} - \vec{w}'), \tag{4.21}$$

one finally has [2]

$$\left(\widetilde{\Phi}_\Delta(x; q), \widetilde{\Phi}_{\Delta'}(x; q')\right) = \frac{\pi^d \Gamma(\Delta - \frac{d}{2})\Gamma(\Delta'^* - \frac{d}{2})}{\Gamma(\Delta)\Gamma(\Delta'^*)} \left(\Phi_\Delta(x; q), \Phi_{\Delta'}(x; q')\right), \tag{4.22}$$

in terms of the scalar inner product (3.18), (3.19). In conclusion, the shadow transform preserves the inner products as expected, up to a factor arising from our choice of normalization in the shadow definition (4.2).

## 4.2 Embedding formalism

For objects carrying nontrivial polarizations, such as $A^\mu_{a,\Delta}$ and $B^{\mu\nu}_{ab,\Delta}$ with $a, b = 1, 2, \ldots, d$, performing the shadow transform involves first building uplifts $\mathcal{A}^\mu_{\alpha,\Delta}$ or $\mathcal{B}^{\mu\nu}_{\alpha\beta,\Delta}$, where the indices $a, b$ are promoted to $\alpha, \beta = 1, 2, \ldots, D$ in such a way that the resulting tensors obey [52]

$$q^\alpha \mathcal{A}^\mu_{\alpha,\Delta}(x; q) = 0, \qquad q^\alpha \mathcal{B}^{\mu\nu}_{\alpha\beta,\Delta}(x; q) = 0, \tag{4.23}$$

and are defined up to terms of the type $q_\alpha(\cdots)^\mu$ or $q_{[\alpha}(\cdots)^{\mu\nu}_{\beta]}$. These uplifts can be obtained systematically replacing the basic polarization $\epsilon^\mu_a(x; q)$ in (3.37) by

$$\varepsilon^\mu_\alpha(x; q) = \frac{\delta^\mu_\alpha}{\mp i0 - q \cdot x} + \frac{x_\alpha q^\mu}{(\mp i0 - q \cdot x)^2}, \tag{4.24}$$

which can be also written formally as follows

$$\varepsilon^\mu_\alpha(x; q) = \frac{\partial}{\partial q^\alpha}\left(\frac{q^\mu}{\mp i0 - x \cdot q}\right) = -\frac{\partial}{\partial q^\alpha}\frac{\partial}{\partial x_\mu}\log(\pm i0 + x \cdot q), \tag{4.25}$$

and reduces to $\epsilon^\mu_a$ when projected along $e^\alpha_a = \partial_a q^\alpha$, i.e.

$$e^\alpha_a \varepsilon^\mu_\alpha = \epsilon^\mu_a. \tag{4.26}$$

Note that $\varepsilon^\mu_\alpha$ also satisfies $x_\alpha \varepsilon^\alpha_\mu(x; q) = 0$. We can thus choose

$$\mathcal{A}^\mu_{\alpha,\Delta}(x; q) = (\mp i)^\Delta \Gamma(\Delta)\left[\frac{\delta^\mu_\alpha}{(\mp i0 - q \cdot x)^\Delta} + \frac{x_\alpha q^\mu}{(\mp i0 - q \cdot x)^{\Delta+1}}\right], \tag{4.27}$$

$$\mathcal{B}^{\mu\nu}_{\alpha\beta,\Delta}(x; q) = (\mp i)^\Delta \Gamma(\Delta)\left[\frac{\delta^\mu_\alpha \delta^\nu_\beta}{(\mp i0 - q \cdot x)^\Delta} + \frac{x_\alpha q^\mu \delta^\nu_\beta + x_\beta q^\nu \delta^\mu_\alpha}{(\mp i0 - q \cdot x)^{\Delta+1}} - \mu \leftrightarrow \nu\right], \tag{4.28}$$

which we can rewrite in the following way (here all derivatives are with respect to $x$)

$$\mathcal{A}^\mu_{\alpha,\Delta}(x; q) = \left(\delta^\mu_\alpha + \frac{1}{\Delta}x_\alpha \partial^\mu\right)\Phi_\Delta(x; q), \tag{4.29}$$

$$\mathcal{B}^{\mu\nu}_{\alpha\beta,\Delta}(x; q) = \left[\delta^\mu_\alpha \delta^\nu_\beta - \delta^\nu_\alpha \delta^\mu_\beta + \frac{1}{\Delta}\left(\delta^\mu_\alpha x_\beta \partial^\nu + \delta^\nu_\beta x_\alpha \partial^\mu - \delta^\nu_\alpha x_\beta \partial^\mu - \delta^\mu_\beta x_\alpha \partial^\nu\right)\right]\Phi_\Delta(x; q) \tag{4.30}$$

in terms of the scalar conformal primary.

## 4.3 $p$-form shadows

Given the uplifted fields discussed in the previous subsection, the shadow transforms are defined by first taking

$$\widetilde{\mathcal{A}}^\mu_{\alpha,\Delta}(x; q) = \int d^d \vec{w}' \frac{-2q \cdot q' \delta^\rho_\alpha + 2q^\rho q'_\alpha}{(-2q \cdot q')^{d-\Delta+1}} \mathcal{A}^\mu_{\beta,\Delta}(x; q'), \tag{4.31}$$

$$\widetilde{\mathcal{B}}^{\mu\nu}_{\alpha\beta,\Delta}(x; q) = \int d^d \vec{w}' \frac{\left[-2q \cdot q' \delta^\rho_\alpha + 2q^\rho q'_\alpha\right]\left[-2q \cdot q' \delta^\sigma_\beta + 2q^\sigma q'_\beta\right]}{(-2q \cdot q')^{d-\Delta+2}} \mathcal{B}^{\mu\nu}_{\rho\sigma,\Delta}(x; q') \tag{4.32}$$

and then projecting along $e_a^\alpha(x;q)$, $e_b^\beta(x;q)$ to finally obtain $\widetilde{A}_{a,\Delta}^\mu(x;q)$ and $\widetilde{B}_\Delta^{\mu\nu}(x;q)$. In fact, (4.29), (4.30) can be also cast in the compact form

$$\widetilde{\mathcal{A}}_{\alpha,\Delta}^\mu(x;q) = \int d^d\vec{w}\,' \frac{\mathcal{A}_{\alpha,d-\Delta}^\rho(q';2q)}{(\mp i)^{d-\Delta}\Gamma(d-\Delta)}\,\mathcal{A}_{\rho,\Delta}^\mu(x;q')\,, \tag{4.33}$$

$$\widetilde{\mathcal{B}}_{\alpha\beta,\Delta}^{\mu\nu}(x;q) = \frac{1}{2!}\int d^d\vec{w}\,' \frac{\mathcal{B}_{\alpha\beta,d-\Delta}^{\rho\sigma}(q';2q)}{(\mp i)^{d-\Delta}\Gamma(d-\Delta)}\mathcal{B}_{\rho\sigma,\Delta}^{\mu\nu}(x;q')\,. \tag{4.34}$$

Again, it is convenient to first rewrite the kernels in terms of derivatives,

$$\widetilde{\mathcal{A}}_{\alpha,\Delta}^\mu(x;q) = \left(\delta_\alpha^\beta + \frac{1}{d-\Delta}q^\beta\frac{\partial}{\partial q^\alpha}\right)\int\frac{d^d\vec{w}\,'}{(-2q\cdot q')^{d-\Delta}}\,\mathcal{A}_{\beta,\Delta}^\mu(x;q')\,, \tag{4.35}$$

$$\widetilde{\mathcal{B}}_{\alpha\beta,\Delta}^{\mu\nu}(x;q) = \left[\delta_\alpha^\rho\delta_\beta^\sigma + \frac{1}{d-\Delta}\left(\delta_\alpha^\rho q^\sigma\frac{\partial}{\partial q^\beta} + \delta_\beta^\sigma q^\rho\frac{\partial}{\partial q^\alpha}\right)\right]\int\frac{d^d\vec{w}\,'}{(-2q\cdot q')^{d-\Delta}}\,\mathcal{B}_{\rho\sigma,\Delta}^{\mu\nu}(x;q')\,. \tag{4.36}$$

Using (4.29), (4.30), the previous equations can be expressed as derivatives of the scalar shadow,

$$\widetilde{\mathcal{A}}_{\alpha,\Delta}^\mu(x;q) = \left(\delta_\alpha^\beta + \frac{1}{d-\Delta}q^\beta\frac{\partial}{\partial q^\alpha}\right)\left(\delta_\beta^\mu + \frac{1}{\Delta}x_\beta\partial^\mu\right)\widetilde{\Phi}_\Delta(x;q)\,, \tag{4.37}$$

$$\begin{aligned}\widetilde{\mathcal{B}}_{\alpha\beta,\Delta}^{\mu\nu}(x;q) &= \left[\delta_\alpha^\rho\delta_\beta^\sigma + \frac{1}{d-\Delta}\left(\delta_\alpha^\rho q^\sigma\frac{\partial}{\partial q^\beta} + \delta_\beta^\sigma q^\rho\frac{\partial}{\partial q^\alpha}\right)\right]\\ &\times\left[\delta_\rho^\mu\delta_\sigma^\nu - \delta_\rho^\nu\delta_\sigma^\mu + \frac{1}{\Delta}\left(\delta_\rho^\mu x_\sigma\partial^\nu + \delta_\sigma^\nu x_\rho\partial^\mu - \delta_\rho^\nu x_\sigma\partial^\mu - \delta_\sigma^\mu x_\rho\partial^\nu\right)\right]\widetilde{\Phi}_\Delta(x;q)\,.\end{aligned} \tag{4.38}$$

One should note that, strictly speaking, taking derivatives with respect to $q^\alpha$ is not allowed, due to the constraint $q^2 = 0$. To be more precise, for each such derivative, one ought to first consider expressions of the type

$$\frac{\partial}{\partial k^\alpha}F(k,x)\,, \qquad \text{with} \quad F(k,x) = \int\frac{d^d\vec{w}\,'}{(-2k\cdot q(\vec{w}\,'))^{d-\Delta}(-2x\cdot q(\vec{w}\,'))^\Delta}\,, \tag{4.39}$$

where both $k^\mu$ and $x^\mu$ are unconstrained $D$-vectors, and only evaluate the result at $k^\mu = q^\mu(\vec{w})$ after taking the derivative. However, we find that this only introduces mismatches that project to zero at the end of the calculations, and are thus immaterial for our present purposes. To see this explicitly, let us first note that, following steps very similar to those applied in the calculation of $\widetilde{\Phi}_\Delta(x;q)$, the integral $F(k,x)$ can be cast in the form

$$F(k,x) = \frac{\pi^{\frac{d}{2}}\Gamma\left(\frac{d}{2}\right)}{\Gamma(d-\Delta)\Gamma(\Delta)}\frac{1}{(-x^2)^{\frac{\Delta}{2}}}\,G(-k^2,-k\cdot\hat{x})\,, \tag{4.40}$$

where

$$\hat{x}^\mu = \frac{x^\mu}{\sqrt{-x^2}}\,, \qquad G(-k^2,-k\cdot\hat{x}) = \int_0^\infty\frac{dt}{t}\frac{t^{d-\Delta}}{[(-k^2)t^2 + 2t(-k\cdot\hat{x}) + 1]^{\frac{d}{2}}}\,. \tag{4.41}$$

Performing the change of variables

$$\lambda = k^+, \qquad w^a = \frac{k^a}{k^+}, \qquad \rho = -\frac{k^2}{(k^+)^2}, \tag{4.42}$$

with $k^\pm = k^0 \pm k^{D-1}$, according to which

$$k^\mu = \lambda \left[ q^\mu(\vec{w}) + \frac{\rho}{2} m^\mu \right], \qquad m = (1, 0, \ldots, 0, -1), \tag{4.43}$$

or more explicitly,

$$k^+ = \lambda, \qquad k^a = \lambda w^a, \qquad k^- = \lambda \left( |\vec{w}|^2 + \rho \right), \tag{4.44}$$

one obtains

$$\frac{\partial}{\partial k^+} = \frac{\partial}{\partial \lambda} - \frac{1}{\lambda} w^a \frac{\partial}{\partial w^a} + \frac{1}{\lambda} \left( |\vec{w}|^2 - \rho \right) \frac{\partial}{\partial \rho}, \tag{4.45}$$

$$\frac{\partial}{\partial k^a} = \frac{1}{\lambda} \left( \frac{\partial}{\partial w^a} - 2w^a \frac{\partial}{\partial \rho} \right), \tag{4.46}$$

$$\frac{\partial}{\partial k^-} = \frac{1}{\lambda} \frac{\partial}{\partial \rho}. \tag{4.47}$$

Using these derivatives, one can check that, denoting by $G_1$ and $G_2$ the partial derivatives of $G$ with respect to its two arguments,

$$\left. \frac{\partial G(-k^2, -k \cdot \hat{x})}{\partial k^\alpha} \right|_{\lambda=1, \, \rho=0} = -\hat{x}_\alpha G_2(0, -q \cdot \hat{x}) - 2q_\alpha G_1(0, -q \cdot \hat{x}). \tag{4.48}$$

This differs by the "naive derivative"

$$\frac{\partial G(0, -q \cdot \hat{x})}{\partial q^\alpha} = -\hat{x}_\alpha G_2(0, -q \cdot \hat{x}) \tag{4.49}$$

just by a term proportional to $q_\alpha$. Terms of this type project to zero by construction after going back from the embedding space to the physical polarizations, so that we can safely drop them throughout our calculations. We only need to deal with first derivatives with respect to $q^\alpha$, so this analysis is exhaustive for the present purposes.

It turns out convenient to introduce [19]

$$m_\alpha^\mu = \delta_\alpha^\mu + \frac{x_\alpha q^\mu}{-q \cdot x} = (-q \cdot x) \, \varepsilon_\alpha^\mu. \tag{4.50}$$

Using (4.37) and the explicit expression (4.12), we see that, in the embedding space, the shadow vector primary is directly related to $m_\alpha^\beta$,

$$\widetilde{\mathcal{A}}_{\alpha,\Delta}^\mu(x; q) = \left( \delta_\beta^\mu + \frac{1}{\Delta} x_\beta \partial^\mu \right) m_\alpha^\beta \, \widetilde{\Phi}_\Delta(x; q). \tag{4.51}$$

From

$$x_\beta \, m_\alpha^\beta = 0 \tag{4.52}$$

we see that no contribution comes from the action of the derivative on $\widetilde{\Phi}_\Delta$. In addition,

$$\left(\delta_\beta^\mu + \frac{1}{\Delta} x_\beta \partial^\mu\right) m_\alpha^\beta = \left(1 - \frac{1}{\Delta}\right) m_\alpha^\mu , \tag{4.53}$$

and thus,

$$\widetilde{\mathcal{A}}_{\alpha,\Delta}^\mu = \left(1 - \frac{1}{\Delta}\right) m_\alpha^\mu \, \widetilde{\Phi}_\Delta(x; q). \tag{4.54}$$

For the two-form, we may similarly write (4.38) in the form

$$\widetilde{\mathcal{B}}_{\alpha\beta,\Delta}^{\mu\nu}(x; q) = \frac{1}{2}\left(\delta_\rho^{[\mu}\delta_\sigma^{\nu]} + \frac{1}{\Delta}\delta_\rho^{[\mu} x_\sigma \partial^{\nu]} + \frac{1}{\Delta}\delta_\sigma^{[\nu} x_\rho \partial^{\mu]}\right) m_\alpha^{[\rho} m_\beta^{\sigma]} \widetilde{\Phi}_\Delta . \tag{4.55}$$

To evaluate this expression, one may note that, by (4.52), $x_\sigma \partial^\nu$ gives a non-vanishing contribution only when it acts on $m_\mu^\sigma$ with upper index $\sigma$, so that

$$x_\sigma \partial^\nu m_\mu^\sigma = -m_\mu^\nu . \tag{4.56}$$

Therefore, the final result is simply

$$\widetilde{\mathcal{B}}_{\alpha\beta,\Delta}^{\mu\nu}(q, x) = \left(1 - \frac{2}{\Delta}\right) m_\alpha^{[\mu} m_\beta^{\nu]} \widetilde{\Phi}_\Delta . \tag{4.57}$$

In summary,

$$\widetilde{\mathcal{A}}_{\alpha,\Delta}^\mu = \left(1 - \frac{1}{\Delta}\right) (-q \cdot x) \varepsilon_\alpha^\mu \, \widetilde{\Phi}_\Delta , \qquad \widetilde{\mathcal{B}}_{\alpha\beta,\Delta}^{\mu\nu} = \left(1 - \frac{2}{\Delta}\right) (-q \cdot x)^2 \left[\varepsilon_\alpha^\mu \varepsilon_\beta^\nu - \varepsilon_\alpha^\nu \varepsilon_\beta^\mu\right] \widetilde{\Phi}_\Delta . \tag{4.58}$$

The projection on the tangent space to the light-cone has the only effect of turning $\varepsilon_\alpha^\mu$ into $\epsilon_a^\mu$ via (4.26), so that the shadow transforms read

$$\widetilde{A}_{a,\Delta}^\mu = \left(1 - \frac{1}{\Delta}\right) (-q \cdot x) \epsilon_a^\mu \, \widetilde{\Phi}_\Delta , \qquad \widetilde{B}_{ab,\Delta}^{\mu\nu} = \left(1 - \frac{2}{\Delta}\right) (-q \cdot x)^2 \epsilon_{ab}^{[\mu\nu]} \, \widetilde{\Phi}_\Delta . \tag{4.59}$$

As for the scalar case, the factors of $(1 - \frac{1}{\Delta})$ and $(1 - \frac{2}{\Delta})$ can be reabsorbed into different choices of normalization in the definition of the shadow transform for fields with nontrivial tensor structure (see e.g. [2, 19] for one-form expressions).

For a generic $p$-form CPW, we define the shadow transform by

$$\widetilde{\mathcal{B}}_{\alpha_1\cdots\alpha_p,\Delta}^{\mu_1\cdots\mu_p}(x; q) = \frac{1}{p!} \int d^d\vec{w}' \frac{\mathcal{B}_{\alpha_1\cdots\alpha_p,d-\Delta}^{\rho_1\cdots\rho_p}(q'; 2q)}{(\mp i)^{d-\Delta}\Gamma(d-\Delta)} \mathcal{B}_{\rho_1\cdots\rho_p,\Delta}^{\mu_1\cdots\mu_p}(x; q') . \tag{4.60}$$

After evaluating the integral we need to compute

$$\widetilde{\mathcal{B}}^{\mu_1\cdots\mu_p}_{\alpha_1\cdots\alpha_p,\Delta} = \frac{1}{p!}\left(\delta^{[\mu_1}_{\rho_1}\cdots\delta^{\mu_p]}_{\rho_p} + \frac{1}{\Delta}\sum_{k=1}^{p}\delta^{[\mu_1}_{\rho_1}\cdots\widehat{\delta^{\mu_k}_{\rho_k}}\cdots\delta^{\mu_p}_{\rho_p}\,x_{\rho_k}\partial^{\mu_k]}\right)m^{[\rho_1}_{\alpha_1}\cdots m^{\rho_p]}_{\alpha_p}\widetilde{\Phi}_\Delta\,, \qquad (4.61)$$

where $\widehat{\delta^{\mu_k}_{\rho_k}}$ means that the factor is omitted from the product. From (4.52) and (4.56) we conclude that

$$\widetilde{\mathcal{B}}^{\mu_1\cdots\mu_p}_{\alpha_1\cdots\alpha_p,\Delta}(x;q) = \left(1 - \frac{p}{\Delta}\right)m^{[\mu_1}_{\alpha_1}\cdots m^{\mu_p]}_{\alpha_p}\widetilde{\Phi}_\Delta\,, \qquad (4.62)$$

and projecting back from the embedding space,

$$\widetilde{B}^{\mu_1\cdots\mu_p}_{a_1\cdots a_p,\Delta}(x;q) = \left(1 - \frac{p}{\Delta}\right)(-q\cdot x)^p\epsilon^{[\mu_1}_{a_1}\cdots\epsilon^{\mu_p]}_{a_p}\widetilde{\Phi}_\Delta\,. \qquad (4.63)$$

The corresponding shadow field strength is thus

$$\widetilde{H}^{\alpha\mu_1\cdots\mu_p}_{a_1\cdots a_p,\Delta}(x;q) = \left(1 - \frac{p}{\Delta}\right)r^{[\alpha}\epsilon^{\mu_1}_{a_1}\cdots\epsilon^{\mu_p]}_{a_p}\widetilde{\Phi}_\Delta\,, \qquad r^\alpha \equiv \frac{(d/2 - \Delta)x^\alpha}{x^2} + \frac{(d - \Delta - p)q^\alpha}{\mp i0 - q\cdot x}\,, \quad (4.64)$$

which is to be compared with (3.109).

In the critical dimension $d = 2p$, we get

$$r^\alpha = (p - \Delta)\left(\frac{x^\alpha}{x^2} + \frac{q^\alpha}{\mp i0 - q\cdot x}\right)\,, \qquad (4.65)$$

so that the shadow of $\Delta = p$ is pure gauge. This is actually not surprising because, in $D = 2 + 2p$, working out the explicit expressions for the shadow field (using in particular (4.12)), one has

$$\widetilde{B}^{\mu_1\cdots\mu_p}_{a_1\cdots a_p,\Delta}(x;q) = \frac{1}{\Delta}(-q\cdot x)^p\epsilon^{[\mu_1}_{a_1}\cdots\epsilon^{\mu_p]}_{a_p}\pi^p(-x^2)^{p-\Delta}\frac{(\mp 2i)^\Delta\Gamma(\Delta - p + 1)}{(-2q\cdot x)^{2p-\Delta}} \qquad (4.66)$$

and setting $\Delta = p$ leads to

$$\widetilde{B}^{\mu_1\cdots\mu_p}_{a_1\cdots a_p,\Delta=p}(x;q) = \frac{(\mp i\pi)^p}{p}\epsilon^{[\mu_1}_{a_1}\cdots\epsilon^{\mu_p]}_{a_p} \qquad (4.67)$$

which is proportional to the pure-gauge wavefunction already obtained in (3.112). In other words, the pure-gauge shadow CPW coincides with the ordinary pure gauge CPW (up to an overall factor); see Table 1.

Let us now turn to the calculation of scalar products involving shadow transforms, starting from the one-form case. To compute the inner product between $A_{a,\Delta}$ and $\widetilde{A}_{b,\Delta}$, we can start from the definition (4.33). The uplifted inner product for one-form primaries can be calculated using the same technique as for (3.53), where $\partial_a$ can be replaced by formal derivatives $\partial/\partial q^\alpha$. As discussed above, this only introduces ambiguities proportional to $q_\alpha$, which can be systematically dropped. Similarly, (3.52) translates to

$$q^\nu\partial_\alpha\delta(q,q')(q'^0\eta_{\alpha'\nu} - q'_\nu\delta^0_{\alpha'}) = -\eta_{\alpha\alpha'}q^0\delta(q,q') + \cdots\,, \qquad (4.68)$$

| | $B^{\mu_1\cdots\mu_p}_{a_1\cdots a_p,\Delta}$ | $\widetilde{B}^{\mu_1\cdots\mu_p}_{a_1\cdots a_p,\Delta}$ |
|---|---|---|
| $d \neq 2p$ | | $\times$ |
| | $\Delta = p$ | |
| $d = 2p$ | | $\Delta = p$ |

**Table 1:** *Occurrences of "pure gauge" p-forms in $D = d+2$ spacetime dimensions. While p-form CPWs of dimension $\Delta = p$ are pure gauge in any d, their shadow is pure gauge only in the critical dimension $d = 2p$. In that case, the expression of the shadow pure-gauge mode coincides (up to a factor) with the non-shadow one, given in (3.112).*

where the omitted terms are proportional to $q_\alpha$ or $q_{\alpha'}$. Proceeding in this way, one obtains

$$\left(\mathcal{A}_{\alpha,\Delta}(x;q), \mathcal{A}_{\alpha',\Delta'}(x;q')\right) = \eta_{\alpha\alpha'}\left(1 - \frac{1}{\Delta}\right)\left(1 - \frac{1}{\Delta'^*}\right)I(\Delta, q; \Delta', q'). \tag{4.69}$$

In turn, this leads to the shadow product

$$\left(\mathcal{A}_{\alpha,\Delta}(x;q), \widetilde{\mathcal{A}}_{\beta,\Delta'}(x;q')\right) = \int d^d\vec{z}\,\mathcal{A}^\rho_{\beta,d-\Delta'^*}(2q',\vec{z})\left(\mathcal{A}_{\alpha,\Delta}(x;q), \mathcal{A}_{\rho,\Delta'}(x;\vec{z})\right)$$

$$= \eta_{\alpha\rho}\left(1 - \frac{1}{\Delta}\right)\left(1 - \frac{1}{\Delta'^*}\right)\mathcal{A}^\rho_{\beta,d-\Delta'^*}(2q',q)\,2(2\pi)^D\,\delta(i(\Delta + \Delta'^* - d)), \tag{4.70}$$

and contracting with $e^\alpha_a(q)e^\beta_b(q')$ this reduces to

$$\left(A_{a,\Delta}(x;q), \widetilde{A}_{b,\Delta'}(x;q')\right) = h_{ab}\left(1 - \frac{1}{\Delta}\right)\left(1 - \frac{1}{\Delta'^*}\right)\frac{2(2\pi)^D}{|\vec{w} - \vec{w}'|^{2\Delta}}\,\delta(i(\Delta + \Delta'^* - d)). \tag{4.71}$$

The inner product for two shadow one-form primaries can be computed analogously. One has

$$\left(\widetilde{\mathcal{A}}_{\alpha,\Delta}(x;q), \widetilde{\mathcal{A}}_{\beta,\Delta'}(x;q')\right) = \int d^d\vec{s}\,\mathcal{A}^\rho_{\alpha,d-\Delta}(2q,\vec{s})\int d^d\vec{z}\,\mathcal{A}^\sigma_{\beta,d-\Delta'^*}(2q',\vec{z})\left(\mathcal{A}_{\rho,\Delta}(x;\vec{s}), \mathcal{A}_{\sigma,\Delta'}(x;\vec{z})\right), \tag{4.72}$$

and therefore, using (4.69) and dropping terms proportional to $q_\alpha$ or $q_\beta$,

$$\left(\widetilde{\mathcal{A}}_{\alpha,\Delta}(x;q), \widetilde{\mathcal{A}}_{\beta,\Delta'}(x;q')\right) = \left(1 - \tfrac{1}{\Delta}\right)\left(1 - \tfrac{1}{\Delta'^*}\right)\delta(i(\Delta + \Delta'^* - d))\int\frac{d^d\vec{s}\,\eta_{\alpha\beta}\,2(2\pi)^D}{|\vec{w} - \vec{s}|^{d-\Delta}|\vec{w}' - \vec{s}|^{d-\Delta'^*}}. \tag{4.73}$$

Projecting along $e^\alpha_a$ and $e^\beta_b$, and recognizing the same expression appearing in the scalar case (4.18), we thus obtain

$$\left(\widetilde{A}_{\alpha,\Delta}(x;q), \widetilde{A}_{\beta,\Delta'}(x;q')\right) = \eta_{\alpha\beta}\,e^\alpha_a\,e^\beta_b\left(1 - \tfrac{1}{\Delta}\right)\left(1 - \tfrac{1}{\Delta'^*}\right)\left(\widetilde{\Phi}_\Delta(x;q), \widetilde{\Phi}_{\Delta'}(x;q')\right). \tag{4.74}$$

A very similar discussion extends to forms with generic degree $p$ and leads to

$$(B_{a_1\cdots a_p,\Delta}(x;q), \widetilde{B}_{a'_1\cdots a'_p,\Delta'}(x;q')) = (e_{a_1\cdots a_p}, e_{a'_1\cdots a'_p}) \left(1 - \tfrac{p}{\Delta}\right) \left(1 - \tfrac{p}{\Delta'^*}\right) \frac{2(2\pi)^D \delta(i(\Delta + \Delta'^* - d))}{|\vec{w} - \vec{w}'|^{2\Delta}},$$
(4.75)

$$\left(\widetilde{B}_{a_1\cdots a_p,\Delta}(x;q), \widetilde{B}_{a'_1\cdots a'_p,\Delta'}(x;q')\right) = (e_{a_1\cdots a_p}, e_{a'_1\cdots a'_p}) \left(1 - \tfrac{p}{\Delta}\right) \left(1 - \tfrac{p}{\Delta'^*}\right) \left(\widetilde{\Phi}_\Delta(x;q), \widetilde{\Phi}_{\Delta'}(x;q')\right),$$
(4.76)

with $(e_{a_1\cdots a_p}, e_{a'_1\cdots a'_p})$ as in (3.116).

# 5   Singular asymptotics

In order to discuss the role that conformal primary wavefunctions play in the context of soft theorems and more broadly in celestial holography, it is crucial to have a detailed understanding of their asymptotic expansion when approaching the conformal boundary of flat spacetime, null infinity $\mathscr{I}$. In order to do so, one has to deal with the problem of calculating the limit as $r \to \infty$, for fixed retarded time $u$ and observation angles, of conformal primary wavefunctions [9,20,31,32]. As we have seen in previous sections, scalar CPWs form the basic ingredient also for $p$-form ones. The delicate issue arises from the presence in denominators of the type $1/(\mp i0 - 2q\cdot x)^\Delta$ of terms where the $r \to \infty$ limit can be compensated by the collinear limit, in which the observation point and the null momentum can be almost parallel. It has been pointed out in [32] that conformally soft limits $\Delta \to \mathbb{Z}$ [9] do not commute with the large-$r$ expansion. Indeed, the stationary phase space approximation usually taken in the soft theorem-asymptotic symmetry literature [56,57] is only strictly valid for finite energy wavefunctions, while generic conformally soft operators, for which $\mathrm{Re}(\Delta) \neq 1$, do not correspond to radiative, finite energy modes. One way to handle this issue, advocated in [32], is to take the conformally soft limit last: this prescription allows to analytically continue the Mellin transform of a radiative amplitude to conformal dimensions lying outside the principal series. The alternative road is to take the opposite order of limits, namely taking first the conformally soft limit of CPWs and then expand them in large-$r$. This leads to overleading wavefunctions at $\mathscr{I}$ (that one would have typically excluded from the phase space), whose inner product with radiative wavefunctions is divergent and thus needs to undergo a renormalization procedure [31,32,58].

In this section, we present a systematic treatment of the asymptotic expansion of CPWs which is based on the method of regions [33,34]. In order to take into account the full range of available directions, we will treat the angular dependence in the distributional sense. In this way, contributions due to the collinear regions turn out to be regular, but give rise to contact terms (i.e. delta functions and their derivatives) on the celestial sphere. Moreover, the limits

$\Delta \to 1, 2, \ldots$ involve nontrivial cancellations of singularities between collinear and non-collinear contributions, which lead in general to the appearance of $\log r$ terms in the asymptotic expansion, exhibiting a "resonance" phenomenon already noted e.g. in [59].

We conclude by presenting a similar analysis for general solutions of the scalar wave equations, expressed in terms of their Fourier representation, highlighting also in that case the presence of two types of series in the asymptotic expansion and their interplay. As an application, we consider the field generated by an idealized scattering event taking place at the origin of the spacetime. This allows us to show how in the physically relevant combination of positive- and negative-frequency modes the $\log r$ terms drop out and one retrieves the standard memory effect [60].

## 5.1  Asymptotics of scalar CPWs

We now want to discuss the limit of the scalar CPW of conformal dimension $\Delta$ in $d+2$ spacetime dimensions

$$\Phi_\Delta^\pm(x; q) = (\mp 2i)^\Delta \Gamma(\Delta)\, f_\Delta^\pm(x; q)\,, \qquad f_\Delta^\pm(x; q) = \frac{1}{(\mp i0 - 2q \cdot x)^\Delta}\,, \tag{5.1}$$

near future null infinity $\mathscr{I}^+$ (analogous expressions can be obtained for past null infinity $\mathscr{I}^-$). Cartesian coordinates $x^\mu$ relate to Bondi coordinates $(u, r, \vec{z})$ via

$$x^\mu = u\, t^\mu + \frac{2r}{1 + |\vec{z}|^2}\, q^\mu(\vec{z})\,, \tag{5.2}$$

with $t^\mu = (1, 0, \ldots, 0)$ and employing the standard parametrization (2.23) for $q^\mu(\vec{z})$. $\mathscr{I}^+$ is reached in the limit $r \to \infty$, while $u, \vec{z}$ are kept fixed. For later convenience, we define

$$\rho^2 = \frac{2r}{1 + |\vec{z}|^2}\,, \qquad m^2 = (\mp i0 + u)(1 + |\vec{w}|^2)\,, \tag{5.3}$$

so that

$$\mp i0 - 2q(\vec{w}) \cdot x = m^2 + \rho^2 |\vec{w} - \vec{z}|^2 \tag{5.4}$$

and the $\pm i0$ prescription is absorbed into the definition of $m^2$. We want to analyze the asymptotic expansion of the quantity [59]

$$f_\Delta = \frac{1}{(m^2 + \rho^2 |\vec{w} - \vec{z}|^2)^\Delta}\,, \tag{5.5}$$

regarded as a distribution in $\vec{z}$, as $\rho \to \infty$, using the so-called method of regions [33, 34][4]. To this end, we need to consider the integral

$$I_\Delta = \int \frac{\varphi(\vec{z})}{(m^2 + \rho^2 |\vec{w} - \vec{z}|^2)^\Delta}\, d^d\vec{z} \tag{5.6}$$

---

[4]We thank S. Pasterski and A. Puhm for discussions on this expansion.

for a generic test function $\varphi$. As $\rho \to \infty$, we need to distinguish two regions in the integration domain. The first region is characterized by the scaling $|\vec{w} - \vec{z}| \sim \mathcal{O}(1)$, so that $m \ll \rho|\vec{w} - \vec{z}|$, while the second one is characterized by the scaling $|\vec{w} - \vec{z}| \sim \mathcal{O}(\frac{m}{\rho})$, so that $|\vec{w} - \vec{z}| \ll 1$. We can separate the integral accordingly as

$$I_\Delta \sim I_\Delta^{(r)} + I_\Delta^{(c)}, \tag{5.7}$$

where

$$I_\Delta^{(r)} \sim \int \frac{\varphi(\vec{z})}{(\rho^2|\vec{w} - \vec{z}|^2)^\Delta} \left[ 1 - \frac{m^2 \Delta}{\rho^2|\vec{w} - \vec{z}|^2} + \mathcal{O}\left(\frac{1}{\rho^4}\right) \right] d^d\vec{z}, \tag{5.8}$$

and

$$I_\Delta^{(c)} \sim \frac{m^{d-2\Delta}}{\rho^d} \int \frac{d^d\vec{x}}{(1 + |\vec{x}|^2)^\Delta} \left[ \varphi(\vec{w}) + \frac{m}{\rho} x^a \partial_a \varphi(\vec{w}) + \frac{m^2}{2\rho^2} x^{a_1} x^{a_2} \partial_{a_1} \partial_{a_2} \varphi(\vec{w}) + \mathcal{O}\left(\frac{1}{\rho^3}\right) \right]. \tag{5.9}$$

In the first integral, we expanded the denominator for $m \ll \rho|\vec{w} - \vec{z}|$, while in last integral we introduced the variable $\vec{x}$ via

$$\vec{z} = \vec{w} + \frac{m}{\rho} \vec{x} \tag{5.10}$$

and expanded $\varphi$ for $\rho \gg m|\vec{x}|$. Moreover, both in (5.8) and (5.9), we extended the integration region back to the whole space, applying the method of regions. The leftover integration in (5.9) can be dealt with using

$$\int \frac{x^{a_1} \cdots x^{a_{2n}}}{(1 + |\vec{x}|^2)^\Delta} d^d\vec{x} = \frac{\pi^{\frac{d}{2}}}{2^{2n}\Gamma(\Delta)} \partial_b^{a_1} \cdots \partial_b^{a_{2n}} \left[ \frac{\Gamma(\Delta - \frac{d}{2} - 2n)}{(1 - |\vec{b}|^2)^{\Delta - \frac{d}{2} - 2n}} \right]_{\vec{b}=0}, \tag{5.11}$$

so that in particular

$$\int \frac{1}{(1 + |\vec{x}|^2)^\Delta} d^d\vec{x} = \frac{\pi^{\frac{d}{2}}\Gamma(\Delta - \frac{d}{2})}{\Gamma(\Delta)}, \qquad \int \frac{x^{a_1} x^{a_2}}{(1 + |\vec{x}|^2)^\Delta} d^d\vec{x} = \frac{\pi^{\frac{d}{2}}\Gamma(\Delta - \frac{d}{2} - 1)}{2\Gamma(\Delta)} \delta^{a_1 a_2}. \tag{5.12}$$

As a result, (5.8) and (5.9) read

$$I_\Delta^{(r)} \sim \int \left( \frac{1}{(\rho^2|\vec{w} - \vec{z}|^2)^\Delta} - \frac{m^2 \Delta}{(\rho^2|\vec{w} - \vec{z}|^2)^{\Delta+1}} \right) \varphi(\vec{z}) \, d^d\vec{z} + \mathcal{O}\left(\frac{1}{\rho^{2(\Delta+2)}}\right), \tag{5.13}$$

and

$$I_\Delta^{(c)} \sim \frac{\pi^{\frac{d}{2}}}{\Gamma(\Delta)} \left[ \frac{\Gamma(\Delta - \frac{d}{2})}{\rho^d(m^2)^{\Delta-\frac{d}{2}}} \varphi(\vec{w}) + \frac{\Gamma(\Delta - \frac{d}{2} - 1)}{4\rho^{d+2}(m^2)^{\Delta-\frac{d}{2}-1}} \nabla^2 \varphi(\vec{w}) \right] + \mathcal{O}\left(\frac{1}{\rho^{d+4}}\right). \tag{5.14}$$

These two expansions translate into the following double series for the distribution (5.5) itself,

$$
\begin{aligned}
f_\Delta \sim &\frac{1}{(\rho^2|\vec{w} - \vec{z}|^2)^\Delta} - \frac{m^2 \Delta}{(\rho^2|\vec{w} - \vec{z}|^2)^{\Delta+1}} + \mathcal{O}\left(\frac{1}{\rho^{2(\Delta+2)}}\right) \\
&+ \frac{\pi^{\frac{d}{2}}}{\Gamma(\Delta)} \left[ \frac{\Gamma(\Delta - \frac{d}{2})}{\rho^d(m^2)^{\Delta-\frac{d}{2}}} \delta^{(d)}(\vec{w} - \vec{z}) + \frac{\Gamma(\Delta - \frac{d}{2} - 1)}{4\rho^{d+2}(m^2)^{\Delta-\frac{d}{2}-1}} \nabla^2 \delta^{(d)}(\vec{w} - \vec{z}) \right] + \mathcal{O}\left(\frac{1}{\rho^{d+4}}\right).
\end{aligned} \tag{5.15}
$$

This expansion is valid for generic complex $\Delta$, and it is straightforward to obtain higher orders in both series. However, care must be exerted for $\Delta = \frac{d}{2}, \frac{d}{2} + 1, \cdots$ where terms in the second line can diverge. Such divergences are canceled by corresponding "resonant" terms in the first line, as can be seen applying (A.7) and (A.15). For instance, letting $\Delta = \frac{d-\lambda}{2}$, retaining the first nontrivial terms, we have

$$f_{\frac{d-\lambda}{2}} \sim \frac{1}{(\rho|\vec{w} - \vec{z}|)^{d-\lambda}} + \mathcal{O}\left(\frac{1}{\rho^{d+2-\lambda}}\right) + \frac{\pi^{\frac{d}{2}}}{\Gamma(\frac{d-\lambda}{2})} \frac{\Gamma(-\frac{\lambda}{2})}{\rho^d(m^2)^{-\frac{\lambda}{2}}} \delta^{(d)}(\vec{w} - \vec{z}) + \mathcal{O}\left(\frac{1}{\rho^{d+2}}\right) \qquad (5.16)$$

and sending $\lambda \to 0$ the $\frac{1}{\lambda}$ singularities cancel between the two terms, thanks to (A.7), leaving behind[5]

$$f_{\frac{d}{2}} \sim \frac{1}{(\rho|\vec{w} - \vec{z}|)^d} + \frac{\pi^{\frac{d}{2}}}{\Gamma(\frac{d}{2})\rho^d} \left[\log\frac{\rho^2}{m^2} - \psi\left(\tfrac{d}{2}\right) - \gamma_E\right] \delta^{(d)}(\vec{w} - \vec{z}) + \mathcal{O}\left(\frac{\log\rho}{\rho^{d+2}}\right), \qquad (5.17)$$

with $\psi$ the digamma function. In a similar fashion, one can obtain the next terms in the expansion retaining one more order and using (A.15), obtaining

$$
\begin{aligned}
f_{\frac{d}{2}} \sim{}& \frac{1}{(\rho|\vec{w} - \vec{z}|)^d} + \frac{\pi^{\frac{d}{2}}}{\Gamma(\frac{d}{2})\rho^d} \left[\log\frac{\rho^2}{m^2} - \psi\left(\tfrac{d}{2}\right) - \gamma_E\right] \delta^{(d)}(\vec{w} - \vec{z}) \\
&- \frac{m^2 d}{2(\rho|\vec{w} - \vec{z}|)^{d+2}} - \frac{\pi^{\frac{d}{2}} m^2}{4\Gamma(\frac{d}{2})\rho^{d+2}} \left[\log\frac{\rho^2 e^2}{m^2} - \psi\left(\tfrac{d}{2}\right) - \gamma_E\right] \nabla^2\delta^{(2)}(\vec{w} - \vec{z}) + \mathcal{O}\left(\frac{\log\rho}{\rho^{d+4}}\right).
\end{aligned}
\qquad (5.18)
$$

Letting instead $\Delta = \frac{d+2-\lambda}{2}$, we find

$$
\begin{aligned}
f_{\frac{d+2-\lambda}{2}} \sim{}& \frac{1}{(\rho|\vec{w} - \vec{z}|)^{d+2-\lambda}} + \mathcal{O}\left(\frac{1}{\rho^{d+4-\lambda}}\right) \\
&+ \frac{\pi^{\frac{d}{2}}}{\Gamma(\frac{d+2-\lambda}{2})} \left[\frac{\Gamma(1-\frac{\lambda}{2})}{\rho^d(m^2)^{1-\frac{\lambda}{2}}} \delta^{(d)}(\vec{w} - \vec{z}) + \frac{\Gamma(-\frac{\lambda}{2})}{4\rho^{d+2}(m^2)^{-\frac{\lambda}{2}}} \nabla^2\delta^{(d)}(\vec{w} - \vec{z})\right] + \mathcal{O}\left(\frac{1}{\rho^{d+4}}\right),
\end{aligned}
\qquad (5.19)
$$

and sending $\lambda \to 0$ yields, after using (A.15), yields the finite expression

$$
\begin{aligned}
f_{\frac{d+2}{2}} \sim{}& \frac{\pi^{\frac{d}{2}}}{\Gamma(\frac{d}{2} + 1)} \frac{1}{\rho^d m^2} \delta^{(d)}(\vec{w} - \vec{z}) + \frac{1}{(\rho|\vec{w} - \vec{z}|)^{d+2}} \\
&+ \frac{\pi^{\frac{d}{2}}}{\Gamma\left(\frac{d}{2} + 1\right)} \frac{1}{4\rho^{d+2}} \left[\log\frac{\rho^2 e}{m^2} - \psi\left(\tfrac{d}{2}\right) - \gamma_E\right] \nabla^2\delta^{(d)}(\vec{w} - \vec{z}) + \mathcal{O}\left(\frac{\log\rho}{\rho^{d+4}}\right),
\end{aligned}
\qquad (5.20)
$$

after using $\psi(z + 1) = \frac{1}{z} + \psi(z)$. To highlight the same cancellation as $\Delta \to \frac{d+4}{2}$, we would need to go further subleading in the expansion of $I_{\Delta}^{(c)}$. Instead, retaining only the first few leading terms, the $I^{(r)}$ series stays subleading and the limit can be taken naively, obtaining

$$f_{\frac{d+4}{2}} \sim \frac{\pi^{\frac{d}{2}}}{\Gamma(\frac{d}{2} + 2)} \left[\frac{1}{\rho^d m^4} \delta^{(d)}(\vec{w} - \vec{z}) + \frac{1}{4\rho^{d+2}m^2} \nabla^2\delta^{(d)}(\vec{w} - \vec{z})\right] + \mathcal{O}\left(\frac{\log\rho}{\rho^{d+4}}\right). \qquad (5.21)$$

---

[5]For simplicity, we omit the $\epsilon$ in $|\vec{w} - \vec{z}|_\epsilon^{-d}$ and $|\vec{w} - \vec{z}|_\epsilon^{-d-2}$ (see appendix A).

We note that the expansion we obtained are consistent with the identity

$$\partial_{m^2} f_\Delta = -\Delta\, f_{\Delta+1}\,. \tag{5.22}$$

It is useful to write down explicitly (5.17), (5.20) and (5.21) for the specialized case of a $d = 2$ celestial sphere. Recalling $\psi(1) = -\gamma_E$, we obtain

$$
\begin{aligned}
f_1 &\sim \frac{1}{(\rho|\vec{w} - \vec{z}|)^2} + \frac{\pi}{\rho^2} \log \frac{\rho^2}{m^2}\, \delta^{(2)}(\vec{w} - \vec{z}) - \frac{m^2}{(\rho|\vec{w} - \vec{z}|)^4} \\
&\quad - \frac{\pi m^2}{4\rho^4} \log \frac{\rho^2 e^2}{m^2}\, \nabla^2 \delta^{(2)}(\vec{w} - \vec{z}) + \mathcal{O}\left(\frac{\log \rho}{\rho^6}\right),
\end{aligned} \tag{5.23a}
$$

$$
f_2 \sim \frac{\pi}{\rho^2 m^2}\, \delta^{(2)}(\vec{w} - \vec{z}) + \frac{1}{(\rho|\vec{w} - \vec{z}|)^4} + \frac{\pi}{4\rho^4} \log \frac{\rho^2 e}{m^2}\, \nabla^2 \delta^{(2)}(\vec{w} - \vec{z}) + \mathcal{O}\left(\frac{\log \rho}{\rho^6}\right), \tag{5.23b}
$$

$$
f_3 \sim \frac{\pi}{2\rho^2 m^4}\, \delta^{(2)}(\vec{w} - \vec{z}) + \frac{\pi}{8\rho^4 m^2}\, \nabla^2 \delta^{(2)}(\vec{w} - \vec{z}) + \mathcal{O}\left(\frac{\log \rho}{\rho^6}\right). \tag{5.23c}
$$

Going back to the original variables using (5.3), one finds that in $d = 2$ the asymptotic expansion near $\mathscr{I}^+$ (5.2) of

$$f_\Delta = \frac{1}{(-2q(\vec{w}) \cdot x)^\Delta} = \frac{1}{\left[u(1 + |\vec{w}|^2) + \frac{2r}{1+|\vec{z}|^2}|\vec{w} - \vec{z}|^2\right]^\Delta} \tag{5.24}$$

for $\Delta = 1, 2, 3$ is given by

$$
\begin{aligned}
f_1 &\sim \frac{(1 + |\vec{z}|^2)}{2r|\vec{w} - \vec{z}|^2} + \frac{\pi(1 + |\vec{z}|^2)}{2r} \log\left[\frac{2r}{u(1 + |\vec{w}|^2)^2}\right] \delta^{(2)}(\vec{w} - \vec{z}) - \frac{u(1 + |\vec{w}|^2)(1 + |\vec{z}|^2)^2}{4r^2|\vec{w} - \vec{z}|^4} \\
&\quad - \frac{\pi u(1 + |\vec{w}|^2)(1 + |\vec{z}|^2)^2}{16r^2} \log\left[\frac{2e^2\, r}{u(1 + |\vec{w}|^2)(1 + |\vec{z}|^2)}\right] \nabla^2 \delta^{(2)}(\vec{w} - \vec{z}) + \mathcal{O}\left(\frac{\log r}{r^3}\right),
\end{aligned} \tag{5.25}
$$

$$
\begin{aligned}
f_2 &\sim \frac{\pi}{2ur}\, \delta^{(2)}(\vec{w} - \vec{z}) + \frac{(1 + |\vec{z}|^2)^2}{4r^2|\vec{w} - \vec{z}|^4} \\
&\quad + \frac{\pi(1 + |\vec{z}|^2)^2}{16r^2} \log\left[\frac{2e\, r}{u(1 + |\vec{w}|^2)(1 + |\vec{z}|^2)}\right] \nabla^2 \delta^{(2)}(\vec{w} - \vec{z}) + \mathcal{O}\left(\frac{\log r}{r^3}\right),
\end{aligned} \tag{5.26}
$$

$$
f_3 \sim \frac{\pi \delta^{(2)}(\vec{w} - \vec{z})}{4u^2 r(1 + |\vec{w}|^2)} + \frac{\pi(1 + |\vec{z}|^2)^2}{32u r^2(1 + |\vec{w}|^2)}\, \nabla^2 \delta^{(2)}(\vec{w} - \vec{z}) + \mathcal{O}\left(\frac{\log r}{r^3}\right) \tag{5.27}
$$

where the $\mp i0$ prescription $u \to \mp i0 + u$ is left implicit for brevity. These expansions are consistent with the identity

$$\partial_u f_\Delta = -\Delta\,(1 + |\vec{w}|^2) f_{\Delta+1}\,. \tag{5.28}$$

In appendix (B), we also provide an explicit cross-check that these expressions satisfy $\Box f_\Delta = 0$. Alternatively, one could write these expansions in complex coordinates, recalling from (2.50) that $\delta^{(2)}(\vec{w} - \vec{z}) = 2\delta^{(2)}(w - z)$ and $\nabla^2 = 4\partial_z \partial_{\bar{z}}$.

## 5.2 Asymptotics of solutions of the wave equation

In this subsection we analyze the asymptotics of the solutions of

$$\Box \Phi(x) = 0 . \tag{5.29}$$

Going to Fourier space, the most general solution of this equation can be written as follows

$$\Phi(x) = \int e^{ip \cdot x} f(p) \, \delta(p^2) \, d^D p , \tag{5.30}$$

where $f$ is arbitrary. This can be split into positive- and negative-frequency parts according to $\Phi(x) = \Phi_+(x) + \Phi_-(x)$, where

$$\Phi_\pm(x) = \int e^{\pm ip \cdot x} f(p) \, \theta(p^0) \, \delta(p^2) \, d^D p , \tag{5.31}$$

each of which separately satisfies the wave equation. This splitting is Lorentz-invariant and we can first focus on the positive-frequency part for definiteness.

Similarly to (5.2), we can change integration variables via

$$p^\mu = \mu \, t^\mu + \omega q^\mu(\vec{w}) , \tag{5.32}$$

whose Jacobian determinant is simply

$$\left| \det \frac{\partial p^\mu}{\partial(\mu, \vec{w}, \omega)} \right| = |\omega|^d \, q^0(\vec{w}) . \tag{5.33}$$

Then,

$$p^2 = -\mu^2 - 2\mu\omega q^0(\vec{w}) , \qquad \theta(p^0)\delta(p^2) = \frac{\theta(\omega) \, \delta(\mu)}{2\omega q^0(\vec{w})} . \tag{5.34}$$

Therefore,

$$\Phi_+(x) = \int_0^\infty \omega^d \frac{d\omega}{2\omega} \int_{\mathbb{R}^d} d^d\vec{w} \, e^{i\omega q(\vec{w}) \cdot x} f(\omega q(\vec{w})) , \tag{5.35}$$

or more explicitly

$$\Phi_+(x) = \int_0^\infty \omega^d \frac{d\omega}{2\omega} \int_{\mathbb{R}^d} d^d\vec{w} \, e^{-i\omega q^0(\vec{w})u - \frac{i\omega r |\vec{w} - \vec{z}|^2}{1 + |\vec{z}|^2}} f(\omega q(\vec{w})) . \tag{5.36}$$

At this stage we want to take the limit $r \to \infty$ and note that there are two regions that grant an $\mathcal{O}(1)$ scaling for the exponent

$$-\frac{i\omega r |\vec{w} - \vec{z}|^2}{1 + |\vec{z}|^2} . \tag{5.37}$$

A first possibility is to consider the near-collinear scaling $\vec{w} = \vec{z} + \frac{1}{\sqrt{r}} \vec{s}$, with $\vec{s}$ formally of $\mathcal{O}(1)$, for which to leading order

$$\Phi_+^{(c)}(x) \sim \frac{1}{r^{\frac{d}{2}}} \int_0^\infty \omega^d \frac{d\omega}{2\omega} e^{-i\omega q^0(\vec{z})u} f(\omega q(\vec{z})) \int_{\mathbb{R}^d} d^d\vec{s} \, e^{-\frac{i\omega |s|^2}{1 + |\vec{z}|^2}} . \tag{5.38}$$

Performing the resulting Gaussian $\vec{s}$-integral, we find

$$\Phi_+^{(c)}(x) \sim \int_0^\infty \left(\frac{\pi(1+|\vec{z}|^2)}{ir}\,\omega\right)^{\frac{d}{2}} e^{-i\omega q^0(\vec{z})u} f(\omega q(\vec{z})) \frac{d\omega}{2\omega}\,. \tag{5.39}$$

Rescaling $\omega$ this can be also cast in the form

$$\Phi_+^{(c)}(x) \sim \int_0^\infty \left(\frac{2\pi\omega}{ir}\right)^{\frac{d}{2}} e^{-i\omega u} f\left(\omega\frac{q(\vec{z})}{q^0(\vec{z})}\right) \frac{d\omega}{2\omega}\,. \tag{5.40}$$

A second possibility is to consider the scaling $\omega \sim \mathcal{O}(r^{-1})$, with generic $|\vec{w}-\vec{z}|^2$. To leading order, considering a general scaling

$$f(\omega q(\vec{w})) \sim \omega^{\alpha-d} f_\alpha(\vec{w}) \qquad \text{as } \omega \to 0\,, \tag{5.41}$$

this leads to

$$\Phi_+^{(r)}(x) \sim \int_{\mathbb{R}^d} d^d\vec{w}\, f_\alpha(\vec{w}) \int_0^\infty \omega^\alpha\, e^{-\frac{i\omega r|\vec{w}-\vec{z}|^2}{1+|\vec{z}|^2}} \frac{d\omega}{2\omega}\,. \tag{5.42}$$

The $\omega$ integral can be performed obtaining

$$\Phi_+^{(r)}(x) \sim \frac{\Gamma(\alpha)}{(ir)^\alpha}(1+|\vec{z}|^2)^\alpha \int_{\mathbb{R}^d} \frac{f_\alpha(\vec{w})}{2|\vec{w}-\vec{z}|^{2\alpha}}\, d^d\vec{w}\,. \tag{5.43}$$

As a cross-check, we can verify that effecting the choice

$$f(\omega\, q(\vec{w})) = 2\omega^{\Delta-d}\, \delta^{(d)}(\vec{w}-\vec{w}_0) \tag{5.44}$$

in eq. (5.35) reproduces the scalar conformal primary $\Phi_\Delta$, and that correspondingly eqs. (5.43), (5.39) reproduce the leading terms in the two lines of (5.15). In conclusion, including the corresponding analysis for negative-frequency modes as well, to leading order in each region

$$\Phi_\pm(x) \sim \frac{\Gamma(\alpha)}{(\pm ir)^\alpha}(1+|\vec{z}|^2)^\alpha \int_{\mathbb{R}^d} \frac{f_\alpha(\vec{w})}{2|\vec{w}-\vec{z}|^{2\alpha}}\, d^d\vec{w} + \int_0^\infty \left(\frac{2\pi\omega}{\pm ir}\right)^{\frac{d}{2}} e^{\mp i\omega u} f\left(\omega\frac{q(\vec{z})}{q^0(\vec{z})}\right) \frac{d\omega}{2\omega}\,. \tag{5.45}$$

The two terms are of the same order when $\alpha \sim d/2$.

As an application, let us consider the field

$$\Phi_F(x) = \int \left[e^{ip\cdot x}F(p) + e^{-ip\cdot x}F^*(p)\right] \theta(p^0)\, 2\pi\delta(p^2) \frac{d^D p}{(2\pi)^D}\,, \qquad F(p) = \sum_n \frac{g_n}{p_n\cdot p} \tag{5.46}$$

where $p_n$ are the hard momenta of a background scattering process dressed with soft scalar emissions, and $g_n$ denotes the coupling between the $n$th hard state and the scalar itself. Since $F(\omega q) = \omega^{-1}F(q)$, for positive $\omega$, the scaling (5.41) corresponds to $\alpha = D-3 = d-1$ and

$F_{d-1}(\vec{w}) = F(q(\vec{w}))$. Then, the expansion (5.45), which we consider for $D \simeq 4$ (i.e. $d \simeq 2$) so that it provides the leading-order terms in the $1/r$-expansion, gives

$$\Phi_F(x) \sim \frac{\Gamma(d-1)(1+|\vec{z}|^2)^{d-1}}{(2\pi)^{d+1}r^{d-1}} \int_{\mathbb{R}^d} \frac{i^{1-d} + (-i)^{1-d}}{2|\vec{w} - \vec{z}|^{2(d-1)}} F(q(\vec{w}))\, d^d\vec{w}$$
$$+ 2\,\mathrm{Re} \int_0^{+\infty} \left(\frac{\omega}{2i\pi r}\right)^{\frac{d}{2}} e^{-i\omega u} \frac{d\omega}{4\pi\omega^2}\, F\left(\frac{q(\vec{z})}{q^0(\vec{z})}\right), \tag{5.47}$$

where we have used $F^*(p) = F(p)$ in the first line and $F(\omega q/q^0) = \omega^{-1} F(q/q^0)$ in the second line. Let us also define

$$N(\vec{z}) = \frac{q(\vec{z})}{q^0(\vec{z})} = (1, \hat{x}(\vec{z})). \tag{5.48}$$

At this point we can set $D = 4 - 2\epsilon$ and perform the integral in the second line, which leads to

$$\Phi_F(x) \sim \frac{\Gamma(1-2\epsilon)(1+|\vec{z}|^2)^{1-2\epsilon}}{(2\pi)^{3-2\epsilon}r^{1-2\epsilon}} \int_{\mathbb{R}^d} \frac{\sin(\pi\epsilon)}{|\vec{w} - \vec{z}|^{2-4\epsilon}} F(q(\vec{w}))\, d^{2-2\epsilon}\vec{w}$$
$$+ 2\,\mathrm{Re}\, \frac{1}{(2i\pi r)^{1-\epsilon}} \frac{\Gamma(-\epsilon)}{4\pi(0 + iu)^{-\epsilon}}\, F(N(\vec{z})). \tag{5.49}$$

In order to take the limit $\epsilon \to 0$ we can use

$$\lim_{\epsilon \to 0} \frac{\sin(\pi\epsilon)}{|\vec{w} - \vec{z}|^{2-4\epsilon}} = \lim_{\epsilon \to 0} \frac{\pi\epsilon}{2|\vec{w} - \vec{z}|^{2-4\epsilon}} = \frac{\pi^2}{2} \delta^{(2)}(\vec{w} - \vec{z}), \tag{5.50}$$

$$\frac{1}{(2i\pi r)^{1-\epsilon}} \frac{\Gamma(-\epsilon)}{4\pi(0 + iu)^{-\epsilon}} = -\frac{1}{\epsilon} \frac{1}{8i\pi^2 r} - \frac{\log r + \log(0 + iu)}{8i\pi^2 r} + \mathcal{O}(\epsilon). \tag{5.51}$$

Note that the fact that we are focusing on the real combination in (5.46) is crucial to grant two simplifications. First, the limit (5.50), which follows from (A.6), involves the $\sin(\pi\epsilon)$ and this compensates the singularity of $1/|\vec{z} - \vec{w}|$ in $D = 4$. Second, while the expansion (5.51) contains singular $1/\epsilon$ and logarithmic $(\log r)/r$ terms, they both drop out in the real part. Using

$$-\frac{\log(0 + iu)}{i\pi} + \frac{\log(0 - iu)}{i\pi} = \mathrm{sgn}(u) = \begin{cases} +1 & (u > 0) \\ -1 & (u < 0), \end{cases} \tag{5.52}$$

one is led to

$$\Phi_F(x) = \frac{F(N)}{8\pi r} - \frac{F(N)}{8\pi r} \mathrm{sgn}(u) = \theta(-u) \frac{F(N)}{4\pi r}, \tag{5.53}$$

$\theta$ denoting the Heaviside step function $\theta(x) = +1$ if $x > 0$ and $\theta(x) = 0$ if $x < 0$. We can make this more explicit defining $\eta_n = +1$ if $n$ is an outgoing state and $\eta_n = -1$ if $n$ is an incoming state, and $p_n = \eta_n(E_n, \mathbf{k}_n)$, so that

$$\Phi_F(x) \sim -\frac{\theta(-u)}{4\pi r} \left[ \sum_{\text{out}} \frac{g_n}{E_n - \mathbf{k}_n \cdot \hat{x}} - \sum_{\text{in}} \frac{g_n}{E_n - \mathbf{k}_n \cdot \hat{x}} \right], \tag{5.54}$$

where $q(\vec{z})/q^0(\vec{z}) = N = (1, \hat{x})$. In this way, we retrieve the standard memory effect [60],

$$\Phi_F(x)\Big|_{u>0} - \Phi_F(x)\Big|_{u<0} \sim \frac{1}{4\pi r} \left[ \sum_{\text{out}} \frac{g_n}{E_n - \mathbf{k}_n \cdot \hat{x}} - \sum_{\text{in}} \frac{g_n}{E_n - \mathbf{k}_n \cdot \hat{x}} \right]. \tag{5.55}$$

In $D = 4$, we can consider a more physical regularization, given by the $-i0$ prescription [61–64]

$$\Phi_F(x) = \int \left[ e^{ip\cdot x} F(p) + e^{-ip\cdot x} F^*(p) \right] \theta(p^0) \, 2\pi\delta(p^2) \frac{d^4 p}{(2\pi)^4} , \qquad F(p) = \sum_n \frac{g_n}{p_n \cdot p - i0} . \tag{5.56}$$

In this case, one sees that only the near-collinear region contributes and one is led to

$$\Phi_F(x) \sim \frac{1}{4\pi r} \int_{-\infty}^{+\infty} \frac{d\omega}{2i\pi} e^{-i\omega u} F(\omega N) , \tag{5.57}$$

so that thanks to

$$\int_{-\infty}^{+\infty} \frac{d\omega}{i2\pi} \frac{e^{-i\omega u}}{-\eta_n \omega - i0} = \int_{-\infty}^{+\infty} \frac{d\omega}{i2\pi} \frac{e^{i\omega\eta_n u}}{\omega - i0} = \theta(\eta_n u) , \tag{5.58}$$

one finds

$$\Phi_F(x) \sim \frac{1}{4\pi r} \sum_n \frac{g_n \, \theta(\eta_n u)}{E_n - \mathbf{k}_n \cdot \hat{x}} , \tag{5.59}$$

or more explicitly

$$\Phi_F(x) \sim \frac{1}{4\pi r} \left[ \theta(u) \sum_{\text{out}} \frac{g_n}{E_n - \mathbf{k}_n \cdot \hat{x}} + \theta(-u) \sum_{\text{in}} \frac{g_n}{E_n - \mathbf{k}_n \cdot \hat{x}} \right], \tag{5.60}$$

which we recognize as the appropriate asymptotics of the retarded solution [65, 66]. Of course, (5.60) also reproduces the standard memory effect (5.55), as it only differs by the solution we had found by taking the $D \to 4$ limit by a $u$-independent term.

# 6 Two-form and scalar celestial primaries in 4D

In this section, we analyze more in detail certain properties of two-form primaries in four spacetime dimensions and discuss their duality to scalar degrees of freedom. This allows us to discuss in a concrete setup the connection between two-form asymptotic charges and scalar soft theorems.

## 6.1 Hodge duality between scalar and two-form primaries

We will mostly work in the standard four-dimensional conventions detailed in subsection 2.4. We begin by recalling the scalar conformal primary wave function (CPW) (3.1)

$$\Phi_\Delta(x; w, \bar{w}) = \frac{(\mp i)^\Delta \Gamma(\Delta)}{(\mp i0 - q \cdot x)^\Delta} , \tag{6.1}$$

which satisfies the Klein-Gordon equation $\Box \Phi_\Delta = 0$ and where the parametrization for the null vector $q^\mu(w, \bar{w})$ is taken as in (2.45). Under Lorentz transformations $\Lambda$,

$$\Lambda^\mu{}_\nu q^\mu(w, \bar{w}) = (cw + d)(c^*\bar{w} + d^*)q^\mu(w', \bar{w}'), \qquad w' = \frac{aw + b}{cw + d} \tag{6.2}$$

and the conformal primary transforms as

$$\Phi_\Delta(\Lambda x; w', \bar{w}') = (cz + d)^\Delta (c^*\bar{z} + d^*)^\Delta \Phi_\Delta(x; w, \bar{w}). \tag{6.3}$$

On the other hand, the two-form CPW (3.79) reads (dropping the $a = w\bar{w}$ index)

$$B_\Delta^{\mu\nu}(x; w, \bar{w}) = (\mp i)^\Delta \Gamma(\Delta)\left[\frac{\partial_w q^{[\mu}\partial_{\bar{w}} q^{\nu]}}{(\mp i0 - q \cdot x)^\Delta} + \frac{q^{[\mu}(x \cdot \partial_w q)\partial_{\bar{w}} q^{\nu]} + \partial_w q^{[\mu}(x \cdot \partial_{\bar{w}} q)q^{\nu]}}{(\mp i0 - q \cdot x)^{\Delta+1}}\right] \tag{6.4}$$

and under Lorentz transformations

$$B_\Delta^{\mu\nu}(\Lambda x, w', \bar{w}') = (cz + d)^\Delta (c^*\bar{z} + d^*)^\Delta \Lambda^\mu{}_\rho \Lambda^\nu{}_\rho B_\Delta^{\rho\sigma}(x, w, \bar{w}). \tag{6.5}$$

Hodge duality provides an on-shell link between a scalar $\Phi$ and the field strength $H^{\mu\nu\rho}$ of the two-form $B^{\mu\nu}$,

$$H^{\mu\nu\rho} = \partial^\mu B^{\nu\rho} + \partial^\nu B^{\rho\mu} + \partial^\nu B^{\rho\mu}, \tag{6.6}$$

according to

$$\frac{1}{3!}\epsilon^{\mu\nu\rho\sigma}H_{\nu\rho\sigma} = \partial^\mu \Phi, \qquad H_{\mu\nu\rho} = \epsilon_{\mu\nu\rho\sigma}\partial^\sigma \Phi. \tag{6.7}$$

The duality interchanges the equations of motion and Bianchi identities:

$$\Box \Phi = 0 \leftrightarrow \epsilon^{\mu\nu\rho\sigma}\partial_\mu H_{\nu\rho\sigma} = 0, \qquad \epsilon^{\mu\nu\rho\sigma}\partial_\rho\partial_\sigma \Phi = 0 \leftrightarrow \partial^\mu H_{\mu\alpha\beta} = 0. \tag{6.8}$$

The duality can be explored at the level of conformal primaries. Using the explicit parameterization (2.23), and $\varepsilon_{0123} = +1$, a short calculation allows one to check that

$$\varepsilon_{\mu\nu\rho\sigma}e_1^\nu e_2^\rho q^\sigma = -q_\mu \tag{6.9}$$

and therefore

$$\varepsilon_{\mu\nu\rho\sigma}\epsilon_1^\nu \epsilon_2^\rho q^\sigma = -\frac{q_\mu}{(-q \cdot x)^2}, \qquad \varepsilon_{\mu\nu\rho\sigma}\epsilon^{[\nu\rho]} q^\sigma = -\frac{2q_\mu}{(-q \cdot x)^2}. \tag{6.10}$$

As a result, the field strength (3.85) obeys

$$\frac{1}{3!}\varepsilon_{\mu\nu\rho\sigma}H_\Delta^{\nu\rho\sigma} = -(\Delta - 2)\frac{(\mp i)^\Delta \Gamma(\Delta)}{(\mp i0 - q \cdot x)^{\Delta+1}} q_\mu \tag{6.11}$$

or, equivalently,

$$\frac{1}{3!}\varepsilon_{\mu\nu\rho\sigma}H_\Delta^{\nu\rho\sigma} = -\left(1 - \frac{2}{\Delta}\right)\partial_\mu \Phi_\Delta. \tag{6.12}$$

Therefore, while for generic $\Delta$ this scalar/two-form duality determines $\Phi_\Delta$ in terms of $B_\Delta^{\mu\nu}$ up to a constant, it fails to do so for $\Delta = 2$. Notice that this is to be expected because $\Delta = 2$ corresponds to a pure gauge two-form, as discussed in section 3.3. Of course, this is only one possible way of explicitly solving the duality equation, which is invariant under gauge transformations on the $B_\Delta^{\mu\nu}$ side and under shifts by constant numbers on the $\Phi_\Delta$. However, this is a nice solution because it preserves the conformal primary nature of both objects, as already remarked.

## 6.2 Revisiting the scalar charge

Within the family of soft theorem-asymptotic symmetry relationships, the scalar case is at the root of potential conceptual puzzles, despite the fact that spinless fields are the easiest to handle technically. Indeed, how can we understand soft factorization theorems for scattering amplitudes as Ward identities in the absence of large local symmetries? In [27], Campiglia et al. studied soft scalar theorems in the field theoretical context where a massless scalar field $\Phi$ is coupled to a massive one via a Yukawa-type interaction[6]. They showed that the leading soft scalar factorization could be understood as arising from the conservation of certain asymptotic charges, whose soft part

$$\mathcal{Q}^{\text{soft}} = \oint_{\mathcal{I}_-^+} \kappa \,, \tag{6.13}$$

is expressed in terms of an antisymmetric tensor

$$\kappa^{\mu\nu} = j_\Lambda^\mu \xi^\nu - j_\Lambda^\nu \xi^\mu \,, \qquad j_\Lambda^\mu = \Lambda \partial^\mu \Phi - \Phi \partial^\mu \Lambda \,, \tag{6.14}$$

where $\xi = \xi^\mu \partial_\mu$ is the dilatation vector field and $\Lambda(x)$ a "symmetry parameter" that satisfies $\Box \Lambda = 0$. In Bondi coordinates, the relevant component of $\kappa$ is $\kappa_{ur}$ and, since $\xi = u\partial_u + r\partial_r$, we then have

$$\kappa_{ur} = (\Lambda \partial_u \Phi - \Phi \partial_u \Lambda)(-u) + (\Lambda \partial_r \Phi - \Phi \partial_r \Lambda)(u + r) \,. \tag{6.15}$$

The field $\Phi$ is taken to be a radiative configuration for which

$$\Phi \sim \frac{1}{r}\,\varphi(u, \vec{z}) + \cdots \,, \tag{6.16}$$

and such that

$$\partial_u \varphi(u, \vec{z}) \sim |u|^{-1-\epsilon} \tag{6.17}$$

for $|u| \to \infty$. A particular choice satisfying this is the real part in (5.25) of the $\Delta = 1$ conformal primary,

$$\operatorname{Re} \Phi_{\Delta=1}^\pm = \frac{\pi^2(1 + |\vec{z}|^2)}{r}\,\theta(-u)\,\delta^{(2)}(\vec{w} - \vec{z}) + \cdots \tag{6.18}$$

---

[6]See [28] for a study of scalar soft theorems for massless cubic interactions in even $D > 4$.

From this expression, we thus see that the radiative free data corresponds to a pure retarded time shift at future null infinity; one can thus interpret (6.18) as the "memory" imprint coming from the conformally soft ($\Delta = 1$) scalar primary, which was already introduced and analyzed in [19].

On the other hand, the asymptotic limit of $\Lambda$ near $\mathscr{I}^+$ is captured by [27]

$$\Lambda \sim \frac{1}{r} \lambda^{(1)} + \frac{1}{r} \log \frac{2|u|}{r} \lambda + \cdots \tag{6.19}$$

where both $\lambda^{(1)}$ and $\lambda$ are arbitrary functions of the angles but do not depend of $u$. This assumption is satisfied in particular by the imaginary part of the conformal primary wavefunction with $\Delta = 1$, for which we have, by (5.25),

$$\mp \operatorname{Im} \Phi_{\Delta=1}^{\pm} \sim \frac{1 + |\vec{z}|^2}{r|\vec{w} - \vec{z}|^2} + \frac{\pi(1 + |\vec{z}|^2)}{r} \log \left[ \frac{2r}{|u|(1 + |\vec{w}|^2)^2} \right] \delta^{(2)}(\vec{w} - \vec{z}) + \cdots . \tag{6.20}$$

The imaginary part of the scalar primary thus plays the role of what would be the scalar version of a "large gauge" transformation at $\mathscr{I}^+$.

Now, substituting (6.16)-(6.19) in (6.15), multiplying by $r^2$ and retaining only terms that do not tend to zero as $r \to \infty$, we have

$$r^2 \kappa_{ur} \sim -u \partial_u \varphi \left( \lambda^{(1)} + \log \frac{2u}{r} \lambda \right) + \varphi \lambda - \varphi \left( \lambda^{(1)} + \log \frac{2u}{r} \lambda \right) + \varphi \left( \lambda^{(1)} + \lambda + \log \frac{2u}{r} \lambda \right) . \tag{6.21}$$

Notice that the first term drops out when evaluating this quantity at $\mathcal{I}_{\pm}^+$ because of (6.17). Then, recalling that the unit sphere is parametrized by $q^\mu(\vec{z})/q^0(\vec{z})$, the scalar charge (6.13) reads [27]

$$\mathcal{Q}_\lambda^{\text{soft}} = \int_{\mathcal{I}^+} \frac{4 du d^2 \vec{z}}{(1 + |\vec{z}|^2)^2} \, 2\lambda(\vec{z}) \partial_u \varphi(u, \vec{z}) . \tag{6.22}$$

In particular, employing the two conformal primaries $\Lambda = \operatorname{Im} \Phi_{\Delta=1}^+$, $\Phi = \operatorname{Re} \Phi_{\Delta=1}^+$, one obtains

$$\mathcal{Q}_\lambda^{\text{soft}} = (2\pi)^3 \, \delta^{(2)}(\vec{w} - \vec{w}') . \tag{6.23}$$

Following [9,19,31,32], this soft charge can thus also be regarded as the canonical pairing between the (would-be) Goldstone and memory modes, whose role is played by the imaginary and real part of the conformally soft CPW, respectively.

Let us now discuss the asymptotic limit for the field operator itself. We know that the leading energetically soft theorem

$$\langle \text{out}|a(\omega q)\mathcal{S}|\text{in}\rangle \sim \sum_n \frac{g_n}{\omega \, p_n \cdot q} \langle \text{out}|\mathcal{S}|\text{in}\rangle , \quad \langle \text{out}|\mathcal{S} a^\dagger(\omega q)|\text{in}\rangle \sim -\sum_n \frac{g_n}{\omega \, p_n \cdot q} \langle \text{out}|\mathcal{S}|\text{in}\rangle \tag{6.24}$$

translates to the conformally soft one

$$\text{Res}\langle \text{out}|a_\Delta(q)\mathcal{S}|\text{in}\rangle\big|_{\Delta=1} \sim \sum_n \frac{g_n}{p_n \cdot q}\langle \text{out}|\mathcal{S}|\text{in}\rangle\,, \tag{6.25}$$

$$\text{Res}\langle \text{out}|\mathcal{S}a_\Delta^\dagger(q)|\text{in}\rangle\big|_{\Delta=1} \sim -\sum_n \frac{g_n}{p_n \cdot q}\langle \text{out}|\mathcal{S}|\text{in}\rangle\,. \tag{6.26}$$

Since we have shown that $\Phi_\Delta$ are smooth functions of $\Delta$, we can calculate $\langle \text{out}|\Phi(x)\mathcal{S}|\text{in}\rangle$, where $\Phi(x)$ is the scalar field operator, by using the explicit expansion (3.26) and using (6.25) to evaluate the $\Delta$ integral via residues, deforming the contour by closing it in the right half-plane. We assume that the only relevant pole comes from $\Delta = 1$. Consistently, we see from (5.24) that higher values $\Delta = 2, 3, \ldots$ corresponding to subleading soft behaviors are further suppressed in the large-$r$ or in the large-$|u|$ limit. Then,[7]

$$\langle \text{out}|\Phi(x)\mathcal{S}|\text{in}\rangle = \frac{1}{2(2\pi)^3}\int_{\mathbb{R}^d} d^2\vec{w}\,\Phi_1^+(x;\vec{w})\,\langle \text{out}|\mathcal{S}|\text{in}\rangle \sum_n \frac{g_n}{p_n \cdot q}\,. \tag{6.27}$$

Using (5.1)

$$\langle \text{out}|\Phi(x)\mathcal{S}|\text{in}\rangle = -\frac{2i}{2(2\pi)^3}\int_{\mathbb{R}^d} d^2\vec{w}\,f_1(x;\vec{w})\,\langle \text{out}|\mathcal{S}|\text{in}\rangle \sum_n \frac{g_n}{p_n \cdot q}\,. \tag{6.28}$$

Combining with the other "half" of the commutator,

$$\langle \text{out}|\mathcal{S}\Phi(x)|\text{in}\rangle = \frac{-1}{2(2\pi)^3}\int_{\mathbb{R}^d} d^2\vec{w}\,\Phi_1^-(x;\vec{w})\,\langle \text{out}|\mathcal{S}|\text{in}\rangle \sum_n \frac{g_n}{p_n \cdot q} \tag{6.29}$$

and

$$\langle \text{out}|\mathcal{S}\Phi(x)|\text{in}\rangle = \frac{-2i}{2(2\pi)^3}\int_{\mathbb{R}^d} d^2\vec{w}\,f_1^*(x;\vec{w})\,\langle \text{out}|\mathcal{S}|\text{in}\rangle \sum_n \frac{g_n}{p_n \cdot q}\,. \tag{6.30}$$

Therefore

$$\langle \text{out}|\left[\Phi(x),\mathcal{S}\right]|\text{in}\rangle = -\frac{2}{(2\pi)^3}\int_{\mathbb{R}^d} d^2\vec{w}\,\text{Im}\,f_1(x;\vec{w})\,\langle \text{out}|\mathcal{S}|\text{in}\rangle \sum_n \frac{g_n}{p_n \cdot q}\,. \tag{6.31}$$

Near $\mathscr{I}^+$, using (5.25) and recalling $q = \frac{1+\vec{z}^2}{2}N$,

$$\langle \text{out}|\left[\Phi(x),\mathcal{S}\right]|\text{in}\rangle = \frac{\theta(-u)}{4\pi r}\sum_n \frac{g_n}{p_n \cdot N}\,. \tag{6.32}$$

Of course we can smear this with $\lambda(\vec{x})$ in particular again with the $\delta$ function term in $f_1$, as we are instructed to do by (6.22).

---

[7]The minus sign due to the fact that the contour runs clockwise is compensated by the minus sign in the argument of $a_{2-\Delta}$.

This derivation parallels the one following from the plane wave representation:

$$\langle\text{out}|[\Phi(x),\mathcal{S}]|\text{in}\rangle = \frac{1}{(2\pi)^3}\int\frac{d^3\mathbf{p}}{2|\mathbf{p}|}\left(e^{ip\cdot x}\langle\text{out}|a(p)\mathcal{S}|\text{in}\rangle - e^{-ip\cdot x}\langle\text{out}|\mathcal{S}a^\dagger(p)|\text{in}\rangle\right) \qquad (6.33)$$

and in the large-$r$ limit [57]

$$\langle\text{out}|[\Phi(x),\mathcal{S}]|\text{in}\rangle = \frac{1}{4\pi r}\int_0^\infty\frac{d\omega}{2i\pi}\left(e^{-i\omega u}\langle\text{out}|a(\omega\hat{x})\mathcal{S}|\text{in}\rangle + e^{i\omega u}\langle\text{out}|\mathcal{S}a^\dagger(\omega\hat{x})|\text{in}\rangle\right). \qquad (6.34)$$

Since we are only interested in the large-$|u|$ limit, we can expand each matrix element using the soft theorem, so that

$$\langle\text{out}|[\Phi(x),\mathcal{S}]|\text{in}\rangle = \frac{1}{4\pi r}\int_{-\infty}^{+\infty}\frac{d\omega}{2i\pi\omega}e^{-i\omega u}\sum_n\frac{g_n}{p_n\cdot N} \qquad (6.35)$$

and one recovers

$$\langle\text{out}|[\Phi(x),\mathcal{S}]|\text{in}\rangle = \frac{\theta(-u)}{4\pi r}\sum_n\frac{g_n}{p_n\cdot N}. \qquad (6.36)$$

Let us finish by commenting on the dual two-form approach to the scalar soft charge (6.13). By the Hodge duality (6.7), for any scalar $\Phi_{1,2}$ and two-form $B_{1,2}$ pairwise dual to each other, the corresponding scalar products obey

$$(\Phi_1,\Phi_2) = (B_1,B_2) - i\int_{\partial\Sigma}\left(\Phi_1 B_2^* - B_1\Phi_2^*\right). \qquad (6.37)$$

So, in general, the standard inner products for scalars and two-forms dual to one another may differ by a boundary term. For our specific solution of the duality however, we find that the scalar products are exactly equal, so that the boundary term must be zero. Therefore, since the Hodge duality relates scalar and form CPWs of the same conformal dimension $\Delta$, one might have expected that inserting the $\Delta = 1$ two-form primary in the dual form inner product would lead to recovering the scalar soft symmetry charge, in the spirit of [29, 30]. This turns out however not to be the case, since $B_{\Delta=1}^{\mu\nu}$ does not reduce to a pure "dual large gauge" configuration. We thus see that this contrasts with the case of the soft photon and soft graviton charges, whose expression can be derived from the inner product between a generic (spin-one or spin-two) field perturbation and a pure gauge, conformally soft, CPW [9, 31]. As it was pointed out in [27], the spin zero soft charge rather resembles the magnetic version of the soft photon and soft graviton charges [67–72]; further connections between form celestial primaries and these new dual asymptotic charges would thus be worth exploring.

# 7 Discussion and outlook

Let us briefly summarize our main results. In this work, we explicitly constructed CPWs for $p$-form fields $B_{a_1\cdots a_p,\Delta}^{\mu_1\cdots\mu_p}$ with arbitrary form degree $p$ in generic spacetime dimension $D$. We derived

the expressions for their inner products and for the corresponding mode decomposition of the canonically quantized free field operators. For each $p$, the CPW $B^{\mu_1\cdots\mu_p}_{a_1\cdots a_p,\Delta=p}$ with conformal weight $\Delta = p$ is pure gauge in any $D$. We then constructed the associated families of shadow transforms $\widetilde{B}^{\mu_1\cdots\mu_p}_{a_1\cdots a_p,\Delta}$, working in the embedding formalism. Such shadow families also possess a pure-gauge waveform $\widetilde{B}^{\mu_1\cdots\mu_p}_{a_1\cdots a_p,\Delta=p}$ with $\Delta = p$ only in the special dimension $D = 2 + 2p$, corresponding to the "critical" dimension for the given form degree, which however coincides with the ordinary one $\widetilde{B}^{\mu_1\cdots\mu_p}_{a_1\cdots a_p,\Delta=p} = B^{\mu_1\cdots\mu_p}_{a_1\cdots a_p,\Delta=p}$. In order to discuss the limit at $\mathscr{I}^+$ of such wavefunctions, we investigated the limit $r \to \infty$ for fixed retarded time and fixed angles, providing a systematic strategy to perform such singular limits based on the method of regions. Finally, we revisited the asymptotic charges of scalar fields in $D = 4$ and their associated dual two-form CPW.

We leave to future work the discussion in the conformal primary basis of dual form memory effects of [73], as well as of further duality links between asymptotic charges associated to forms of different degrees, such as the one proposed in $D = 4$ in [29, 30]. The main appeal of such constructions is that the symmetry interpretation of the charges can be more transparent in one formulation than in the other. In particular, scalars do not have bona fide asymptotic symmetries, while two-forms do [29, 30]. The technical reason for the absence of a natural map between the soft charge associated to a leading soft scalar theorem and a symmetry charge involving a dual two-form CPW is that the latter is not pure gauge for $\Delta = 1$ and therefore its canonical pairing cannot be interpreted as the charge associated to a symmetry transformation. However, pairings between scalars and pure-gauge forms might occur when investigating subleading soft theorems [74, 75], a direction which is therefore worth exploring. In this respect, it would be natural to investigate conformally soft theorems providing analogs of know energetically soft theorems involving for instance two-forms and scalars. The latter, particularly in the case of the axion and the dilaton, have been reformulated in terms of the geometry of field space [76], and very recently a geometric formulation of conformally soft theorems was given [77, 78]. Finally, it can also be of interest to complement the study initiated in this paper by constructing CPWs for more "exotic" types of field including higher-spin ones [79–83]. Although interacting theories involving such fields are severely limited by a number of no-go results, including Weinberg's soft theorem which rules out their long range interactions, interest in theories involving massless higher-spin quanta is motivated by its connections to the high-energy limit of the string spectrum [84].

# Acknowledgments

We would like to thank Adrien Fiorucci, Gaston Giribet, Sabrina Pasterski, Andrea Puhm, Romain Ruzziconi, and Shahin Sheikh-Jabbari for discussions. LD acknowledges support from the Austrian Science Fund (FWF) START project Y 1447-N and from the INFN Iniziativa

Specifica ST&FI. The work of CH is supported by the Knut and Alice Wallenberg Foundation under grant KAW 2018.0116. Nordita is partially supported by Nordforsk.

# A    Distribution identities

Starting from the one-dimensional case, we note that, provided $\mathrm{Re}\,\lambda > 0$, integration by parts gives

$$\int_0^\infty \lambda t^{\lambda-1}\varphi(t)\,dt = -\int_0^\infty t^\lambda \varphi'(t)\,dt \sim \varphi(0) - \lambda \int_0^\infty \log(t)\,\varphi'(t)\,dt + \mathcal{O}(\lambda^2)\,, \tag{A.1}$$

for any test function $\varphi$. This implies, as $\lambda \to 0$ for $\mathrm{Re}\,\lambda > 0$,

$$\lambda t^{\lambda-1}\theta(t) \sim \delta(t) + \frac{\lambda}{|t|_\epsilon} + \mathcal{O}(\lambda^2)\,, \tag{A.2}$$

where $|t|_\epsilon^{-1}$ is the distribution defined by

$$\int \frac{1}{|t|_\epsilon}\varphi(t)\,dt = -\int_0^\infty \log(t)\,\varphi'(t)\,dt\,. \tag{A.3}$$

Noting that

$$\int \frac{1}{|t|_\epsilon}\varphi(t)\,dt = \lim_{\epsilon\to 0^+}\left[\varphi(0)\log(\epsilon) + \int_\epsilon^\infty \frac{dt}{t}\,\varphi(t)\right]\,, \tag{A.4}$$

we can also write

$$\frac{1}{|t|_\epsilon} = \lim_{\epsilon\to 0^+}\left[\delta(t)\log(\epsilon) + \frac{\theta(t-\epsilon)}{t}\right]\,. \tag{A.5}$$

The $d$-dimensional analog of (A.2) can be obtained considering

$$\int \lambda|\vec{w}-\vec{z}|^{\lambda-d}\varphi(\vec{z})\,d^d\vec{z} = \oint d\Omega_d(\hat{x})\int_0^\infty \lambda r^{\lambda-1}\varphi(\vec{w}+r\hat{x})\,dr \tag{A.6}$$

and applying (A.2) to the integral over $r$. One obtains

$$\lambda|\vec{w}-\vec{z}|^{\lambda-d} \sim \frac{2\pi^{\frac{d}{2}}}{\Gamma(\frac{d}{2})}\delta^{(d)}(\vec{w}-\vec{z}) + \frac{\lambda}{|\vec{w}-\vec{z}|_\epsilon^d} + \mathcal{O}(\lambda^2) \qquad (\lambda\to 0\,,\ \mathrm{Re}\,\lambda > 0)\,. \tag{A.7}$$

Once again a subtraction of the type (A.5) is needed to make sense of the ill-defined distribution $|\vec{w}-\vec{z}|^{-d}$ in $d$ dimensions,

$$\frac{1}{|\vec{w}-\vec{z}|_\epsilon^d} = \frac{2\pi^{\frac{d}{2}}}{\Gamma(\frac{d}{2})}\delta^{(d)}(\vec{w}-\vec{z})\log(\epsilon) + \frac{\theta\left(|\vec{w}-\vec{z}|-\epsilon\right)}{|\vec{w}-\vec{z}|^d}\,, \tag{A.8}$$

where the limit $\epsilon\to 0^+$ is left implicit.

For instance, using (A.7) and the identities

$$\partial_a |\vec{w} - \vec{z}|^\alpha = \alpha x_a |\vec{w} - \vec{z}|^{\alpha-1}\,, \tag{A.9}$$

$$\nabla^2 |\vec{w} - \vec{z}|^\alpha = \alpha(\alpha + d - 2)|\vec{w} - \vec{z}|^{\alpha-2}\,, \tag{A.10}$$

one gets

$$\nabla^2 |\vec{w} - \vec{z}|^{\lambda+2-d} = (\lambda + 2 - d)\,\lambda |\vec{w} - \vec{z}|^{\lambda-d} \tag{A.11}$$

and, sending $\lambda \to 0$, one retrieves in this way the expressions of the Green's functions

$$\nabla^2 |\vec{w} - \vec{z}|^{2-d} = \frac{2\pi^{\frac{d}{2}}}{\Gamma(\frac{d}{2})}\,\delta^{(d)}(\vec{w} - \vec{z})\,, \tag{A.12}$$

$$\nabla^2 \frac{\log |\vec{w} - \vec{z}|}{|\vec{w} - \vec{z}|^{d-2}} = \frac{2\pi^{\frac{d}{2}}}{\Gamma(\frac{d}{2})}\,\delta^{(d)}(\vec{w} - \vec{z}) + \frac{2-d}{|\vec{w} - \vec{z}|^d_\epsilon}\,. \tag{A.13}$$

When $d = 2$ the latter reduces to the formula

$$\nabla^2 \log |\vec{w} - \vec{z}| = 2\pi\delta^{(2)}(\vec{w} - \vec{z})\,. \tag{A.14}$$

Applying the Laplace operator $\nabla^2$ to (A.2) itself, one also gets

$$\lambda |\vec{w} - \vec{z}|^{\lambda-d-2} \sim \frac{\pi^{\frac{d}{2}}}{2\Gamma(\frac{d}{2}+1)}\left(1 + \frac{d+2}{2d}\lambda\right)\nabla^2\delta^{(d)}(\vec{w} - \vec{z}) + \frac{\lambda}{|\vec{w} - \vec{z}|^{d+2}_\epsilon} + \mathcal{O}(\lambda^2)\,, \tag{A.15}$$

where we have defined $|\vec{w} - \vec{z}|^{-d-2}_\epsilon$ appearing on the right-hand side via

$$\nabla^2 |\vec{w} - \vec{z}|^{-d}_\epsilon = 2d|\vec{w} - \vec{z}|^{-d-2}_\epsilon\,, \tag{A.16}$$

in order to comply with the formal behavior of $|\vec{w} - \vec{z}|^{-d}$ under (A.10).

Let us work out these quantities explicitly in $d = 2$, starting from the definition (A.8), which reads

$$\frac{1}{|\vec{w} - \vec{z}|^2_\epsilon} = 2\pi \log(\epsilon)\,\delta^{(2)}(\vec{w} - \vec{z}) + \frac{\theta(|\vec{w} - \vec{z}| - \epsilon)}{|\vec{w} - \vec{z}|^2}\,. \tag{A.17}$$

The first derivative involves in particular

$$\frac{1}{|\vec{z}|^2}\,\partial_a\theta(|\vec{z}| - \epsilon) = \frac{z_a}{|\vec{z}|^3}\,\delta(|\vec{z}| - \epsilon) \sim -\pi\partial_a\delta^{(2)}(\vec{z}) + \mathcal{O}(\epsilon)\,, \tag{A.18}$$

where in the last step we used, letting $\hat{n}(\theta) = (\cos\theta, \sin\theta)$,

$$\int \frac{z_a}{|\vec{z}|^3}\,\delta(|\vec{z}| - \epsilon)\varphi(\vec{z})d^2\vec{z} = \frac{1}{\epsilon}\int_0^{2\pi} n_a(\theta)\varphi\left(\epsilon\,\hat{n}(\theta)\right)d\theta \sim \pi\partial_a\varphi(0) + O(\epsilon)\,. \tag{A.19}$$

Using (A.18) in (A.17), we obtain

$$\partial_a\frac{1}{|\vec{w} - \vec{z}|^2_\epsilon} = 2\pi \log(\epsilon)\,\partial_a\delta^{(2)}(\vec{w} - \vec{z}) - \pi\,\partial_a\delta^{(2)}(\vec{w} - \vec{z}) + \theta(|\vec{w} - \vec{z}| - \epsilon)\frac{-2(z_a - w_a)}{|\vec{w} - \vec{z}|^4}\,. \tag{A.20}$$

The next derivative involves

$$\frac{-2z_a}{|\vec{z}|^4}\,\partial_a\theta(|\vec{z}|-\epsilon) = \frac{-2}{|\vec{z}|^3}\,\delta(|\vec{z}|-\epsilon) \sim -\frac{4\pi}{\epsilon^2}\,\delta^{(2)}(\vec{z}) - \pi\nabla^2\delta^{(2)}(\vec{z}) + \mathcal{O}(\epsilon)\,, \tag{A.21}$$

where we used

$$\frac{1}{|\vec{z}|^3}\,\delta(|\vec{z}|-\epsilon) = \frac{1}{\epsilon^2}\int\varphi(\epsilon\hat{n}(\theta))d\theta \sim \frac{2\pi}{\epsilon^2}\,\varphi(0) + \frac{\pi}{2}\,\nabla^2\varphi(0) + \mathcal{O}(\epsilon)\,. \tag{A.22}$$

We then obtain

$$\nabla^2\frac{1}{|\vec{w}-\vec{z}|^2_\epsilon} = -\frac{4\pi}{\epsilon^2}\,\delta^{(2)}(\vec{w}-\vec{z}) + 2\pi\log(\epsilon)\,\nabla^2\delta^{(2)}(\vec{w}-\vec{z}) - 2\pi\,\nabla^2\delta^{(2)}(\vec{w}-\vec{z}) + \frac{4\theta(|\vec{w}-\vec{z}|-\epsilon)}{|\vec{w}-\vec{z}|^4}\,. \tag{A.23}$$

Via (A.16), this defines $4|\vec{w}-\vec{z}|_\epsilon^{-4}$.

Let us also calculate

$$\nabla^2\frac{1+|\vec{z}|^2}{|\vec{w}-\vec{z}|^2_\epsilon} = \frac{4}{|\vec{w}-\vec{z}|^2_\epsilon} + 4z^a\partial_a\frac{1}{|\vec{w}-\vec{z}|^2_\epsilon} + (1+|\vec{z}|^2)\nabla^2\frac{1}{|\vec{w}-\vec{z}|^2_\epsilon}\,. \tag{A.24}$$

Using (A.20) and (A.23), we find

$$\nabla^2\frac{1+|\vec{z}|^2}{|\vec{w}-\vec{z}|^2_\epsilon} = -(1+|\vec{w}|^2)\frac{4\pi}{\epsilon^2}\,\delta^{(2)}(\vec{w}-\vec{z}) + 2\pi\log(\epsilon)\left[4+4z^a\partial_a+(1+|\vec{z}|^2)\nabla^2\right]\delta^{(2)}(\vec{w}-\vec{z})$$
$$- 4\pi\,z^a\partial_a\delta^{(2)}(\vec{w}-\vec{z}) - 2\pi(1+|\vec{z}|^2)\nabla^2\delta^{(2)}(\vec{w}-\vec{z}) + \theta(|\vec{w}-\vec{z}|-\epsilon)\frac{4(1+|\vec{w}|^2)}{|\vec{w}-\vec{z}|^4}\,, \tag{A.25}$$

and after integration by parts this reduces to

$$\nabla^2\frac{1+|\vec{z}|^2}{|\vec{w}-\vec{z}|^2_\epsilon} = \frac{4(1+|\vec{w}|^2)}{|\vec{w}-\vec{z}|^4_\epsilon} + 4\pi w^a\partial_a\delta^{(2)}(\vec{w}-\vec{z})\,. \tag{A.26}$$

# B    Asymptotic expansion cross-checks

The equation of motion $\Box f_\Delta = 0$ provides useful cross-checks on the asymptotic expansions worked out in section (5.1). Using (5.2) to go to retarded coordinates, we have (see e.g. [85])

$$\Box = \partial_\mu\partial^\mu = -\left(2\partial_r + \frac{d}{r}\right)\partial_u + \left(\partial_r^2 + \frac{d}{r}\partial_r + \frac{1}{r^2}\tilde{\nabla}^2\right)\,, \tag{B.1}$$

where $\tilde{\nabla}^2$ is the Laplacian on the sphere $\mathcal{S}^d$, which is related to the one on the Euclidean space $\mathbb{R}^d$, denoted $\nabla^2 = \partial_a\partial_a$, by

$$\tilde{\nabla}^2 = \left(\frac{1+|\vec{z}|^2}{2}\right)^d\partial_a\left(\left(\frac{1+|\vec{z}|^2}{2}\right)^{2-d}\partial_a\right)\,. \tag{B.2}$$

Writing the asymptotic expansion of a generic $f$ in the form

$$f \sim \sum_k \frac{f^{(k)}}{r^k} + \sum_k \frac{g^{(k)}}{r^k} \log(r) \,, \tag{B.3}$$

the equation of motion for $f$ translates into the recursion relations

$$(d - 2k)\partial_u g^{(k)} = [\tilde{\nabla}^2 + (k-1)(k-d)]g^{(k-1)} \,, \tag{B.4}$$

$$(d - 2k)\partial_u f^{(k)} + 2\partial_u g^{(k)} = [\tilde{\nabla}^2 + (k-1)(k-d)]f^{(k-1)} + (d+1-2k)g^{(k-1)} \,. \tag{B.5}$$

For simplicity, let us only verify the $d = 2$ expansions (5.25), (5.26), (5.27), for which

$$g_1^{(1)} = \frac{\pi(1 + |\vec{z}|^2)}{2} \delta^{(2)}(\vec{w} - \vec{z}) \,, \qquad g_1^{(2)} = -\frac{\pi u(1 + |\vec{w}|^2)(1 + |\vec{z}|^2)^2}{16} \nabla^2 \delta^{(2)}(\vec{w} - \vec{z}) \,,$$

$$f_1^{(1)} = \frac{1 + |\vec{z}|^2}{2|\vec{w} - \vec{z}|^2} + \frac{\pi(1 + |\vec{z}|^2)}{2} \log\left[\frac{2}{u(1 + |\vec{w}|^2)^2}\right] \delta^{(2)}(\vec{w} - \vec{z}) \,,$$

$$f_1^{(2)} = -\frac{u(1 + |\vec{w}|^2)(1 + |\vec{z}|^2)^2}{4|\vec{w} - \vec{z}|^4}$$

$$\qquad - \frac{\pi u(1 + |\vec{w}|^2)(1 + |\vec{z}|^2)^2}{16} \log\left[\frac{2e^2}{u(1 + |\vec{w}|^2)(1 + |\vec{z}|^2)}\right] \nabla^2 \delta^{(2)}(\vec{w} - \vec{z}) \,, \tag{B.6}$$

$$f_2^{(1)} = \frac{\pi}{2u} \delta^{(2)}(\vec{w} - \vec{z}) \,, \qquad g_2^{(2)} = \frac{\pi(1 + |\vec{z}|^2)^2}{16} \nabla^2 \delta^{(2)}(\vec{w} - \vec{z}) \,,$$

$$f_2^{(2)} = \frac{(1 + |\vec{z}|^2)^2}{4|\vec{w} - \vec{z}|^4} + \frac{\pi(1 + |\vec{z}|^2)^2}{16} \log\left[\frac{2e}{u(1 + |\vec{w}|^2)(1 + |\vec{z}|^2)}\right] \nabla^2 \delta^{(2)}(\vec{w} - \vec{z}) \,, \tag{B.7}$$

$$f_3^{(1)} = \frac{\pi \delta^{(2)}(\vec{w} - \vec{z})}{4u^2(1 + |\vec{w}|^2)} \,, \qquad f_3^{(2)} = \frac{\pi(1 + |\vec{z}|^2)^2}{32u(1 + |\vec{w}|^2)} \nabla^2 \delta^{(2)}(\vec{w} - \vec{z}) \,. \tag{B.8}$$

Using these expressions and (B.2) when $d = 2$, we see that: $f_1$ trivially solves (B.4), (B.5) for $k = 1$, while (B.4) for $k = 2$ provides a cross-check involving $g_1^{(1)}$ and $g_1^{(2)}$; $f_2$ trivially solves (B.4) for $k = 2$ and (B.5) for $k = 1$, while (B.5) for $k = 2$ provides a cross-check involving $f_2^{(1)}$ and the $u$-dependent part $f_2^{(2)}$; (B.5) for $k = 2$ provides a cross-check involving $f_3^{(1)}$ and $f_3^{(2)}$. To check (B.5) for $f_1$ when $k = 2$, we note that

$$\frac{-2\partial_u f^{(2)} + 2\partial_u g^{(2)}}{(1 + |\vec{w}|^2)(1 + |\vec{z}|^2)^2} = \frac{1}{2|\vec{w} - \vec{z}|^4} + \frac{\pi}{8} \log\left[\frac{2}{u(1 + |\vec{w}|^2)(1 + |\vec{z}|^2)}\right] \nabla^2 \delta^{(2)}(\vec{w} - \vec{z}) \tag{B.9}$$

and

$$\frac{\tilde{\nabla}^2 f^{(1)} - g^{(1)}}{(1 + |\vec{w}|^2)(1 + |\vec{z}|^2)^2} = \frac{1}{2|\vec{w} - \vec{z}|^4} + \frac{\pi w^a \partial_a \delta^{(2)}(\vec{w} - \vec{z})}{2(1 + |\vec{w}|^2)}$$

$$\qquad + \frac{\pi}{8} \log\left[\frac{2}{u(1 + |\vec{w}|^2)^2}\right] \nabla^2 \delta^{(2)}(\vec{w} - \vec{z}) - \frac{\pi \delta^{(2)}(\vec{w} - \vec{z})}{2(1 + |\vec{w}|^2)^2} \,, \tag{B.10}$$

where we used (see (A.26))

$$\nabla^2 \left( \frac{1 + |\vec{z}|^2}{|\vec{z} - \vec{w}|^2} \right) = \frac{4(1 + |\vec{w}|^2)}{|\vec{w} - \vec{z}|^4} + 4\pi w^a \partial_a \delta^{(2)}(\vec{w} - \vec{z}) \,. \tag{B.11}$$

The difference between (B.9) and (B.10) is thus

$$\frac{\pi}{8} \log \frac{1 + |\vec{w}|^2}{1 + |\vec{z}|^2} \nabla^2 \delta^{(2)}(\vec{w} - \vec{z}) - \frac{\pi \, w^a \partial_a \delta^{(2)}(\vec{w} - \vec{z})}{2(1 + |\vec{w}|^2)} + \frac{\pi \delta^{(2)}(\vec{w} - \vec{z})}{2(1 + |\vec{w}|^2)^2} \,. \tag{B.12}$$

This quantity indeed vanishes, as one can check integrating by parts.

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
