# Peer review of "$p$-Forms on the Celestial Sphere"

_SciPost Physics_

## Round 1 · Referee Report · Anonymous · 2023-3-20

Strengths

1. Gives an exceptionally clear treatment of the topic with a high level of detail regarding the construction of the higher-form CPW, their scalar products and shadow transforms.
2. Provides a detailed analysis of the asymptotic expansions of CPWs which is useful for understanding soft-scalar charges.

Report

In this work the authors construct conformal primary wave functions (CPW) for higher-form fields. These solutions are built from scalar CPWs and polarisation tensors. They further compute the scalar products of these solutions, provide a map between creation/annihilation operators and CPW states and construct the shadow transforms of the p-form CPW solutions. A key application of the work is to the study of soft physics, asymptotic symmetries and celestial holography. To this end the authors provide a very detailed analysis of the asymptotic expansion of CPWs including all distributional terms and a careful treatment of the cases where the dimensions are integers. Finally, the authors consider the scalar charge related to soft-scalar factorization in scattering providing a expression for the charge found by Campiglia et al which can be interpreted as a canonical pairing of the Goldstone and memory modes.

The paper is well written with very detailed descriptions of the calculations and will be very useful to researchers in the field.

Requested changes

There are a small number of typos, misprints etc
Page 10, equation (2.63) seems to have a sign error.
Page 2, equation (3.112) indices seem wrong
Page 47, equation (6.6) indices seem wrong
Page 52 "of know energetically soft theorems"

  • validity: top
  • significance: high
  • originality: high
  • clarity: high
  • formatting: excellent
  • grammar: excellent

Author:  Laura Donnay  on 2023-04-06  [id 3557]

(in reply to Report 1 on 2023-03-20)

We would like to sincerely thank the Referee for their careful reading of the manuscript and their kind words regarding our work.

We have implemented the requested minor changes in the revised version. More precisely, typos in eq. (3.112) and and (6.6) have been fixed. We confirm eq. (2.63) is correct (no sign error).

Many thanks again.

---

## Round 1 · Referee Report · Anonymous · 2023-3-30

Report

In this paper, the authors generalize the construction of conformal primary wavefunctions (cpw) to p-form fields. Using the fact that conformal transformations on the $d$-dimensional celestial sphere coincide with Lorentz symmetry in the embedding $(d+2)$-dimensional Minkowski space, the cpw are determined by their transformation properties as conformal primaries under these symmetries. Paralleling the construction of higher-spin conformal primary wavefunctions, $p$-form ones are constructed by dressing the scalar cpw with appropriate antisymmetric combinations of the polarization tensors. Inner products and mode decompositions are computed. Shadow transforms are studied and it is shown that the shadows of the pure gauge $\Delta = p$ cpw associated with $p$-form fields are pure gauge and proportional to the original cpw only in the critical embedding dimension $2 + 2p.$ The large-$r$ asymptotics of these solutions are studied, keeping track of logarithmic terms which nevertheless are shown to drop out for real solutions. The paper concludes with a discussion of asymptotic charges for scalars and the relation to 2-form symmetries.

Some comments:
-It may be useful for the reader if the authors briefly explained the origin of the equation of motion 2.63 for the two-form $B_{\mu\nu}$.
-The inner product 3.56 naively vanishes (due to the coefficient) for $\Delta \in \mathbb{Z}$, $\Delta \neq 0$ while it is finite for $\Delta = 0$. 3.54 which have a normalization differing by $\Gamma(\Delta)$ are similarly zero or divergent for $\Delta = 0,1,2.$ It would be useful if the authors commented briefly on the relation between these inner products and the ones defined in more recent papers on the discrete conformal primary basis (where the inner products are finite) and explain how the p-form story is affected (if at all).
- It might be nice to comment on the relation (if any) between the imaginary, logarithmic part of the large-$r$ expansions in 5.51, logarithmic soft theorems and the Weinberg Coulomb phases.

Some typos:
-In the title of section 3: wavefuntions $\rightarrow$ wavefunctions
-What is the role of the comma in 3.58?

The paper extends the construction of conformal primary wavefunctions to $p$-form fields and clarifies some issues related to their asymptotic expansions. The analysis may be important in top-down constructions of flat space holography. It is timely and well written, hence I recommend it for publication in SciPost.

  • validity: top
  • significance: high
  • originality: good
  • clarity: top
  • formatting: excellent
  • grammar: -

Author:  Laura Donnay  on 2023-04-06  [id 3558]

(in reply to Report 2 on 2023-03-30)

We would like to sincerely thank the Referee for their careful reading of the manuscript and their useful comments and suggestions.

  • We have added a footnote on page 10 regarding the equations of motion (2.63) for the two-form.
  • The conformal primary wavefunction inner products were indeed computed following similar techniques as Ref. [31], by means of the generalized delta function introduced there. We have added a comment below eq. (3.56) which point out to newly added references [46,47] on the recent proof of that CPWs span a discrete basis for integer dimensions.
  • We have added a comment below eq. (5.55) regarding the appearance of logarithms in the context of tail effects and corrections to soft theorems, as well as references [63,64].
  • Typos have been fixed (the comma in (3.58) was a typo).

---

## Editorial Decision

resubmitted